# LARGE DEPTH COMPLETION MODEL FROM SPARSE OBSERVATIONS

**Zhu Yu**[1]* **Zhengyi Zhao**[2] **Runmin Zhang**[1] **Lingteng Qiu**[2] **Kejie Qiu**[2]
**Yisheng He**[2] **Siyu Zhu**[3] **Zilong Dong**[2†] **Si-Yuan Cao**[4,5,6†] **Hui-Liang Shen**[1]

[1]Zhejiang University    [2]Tongyi Lab, Alibaba Group    [3]Fudan University
[4]Ningbo Innovation Center, Zhejiang University    [5]NingboTech University
[6]Jinhua Institute of Zhejiang University
{yu_zhu, cao_siyuan}@zju.edu.cn
 Project Page: https://pkqbajng.github.io/ldcm/

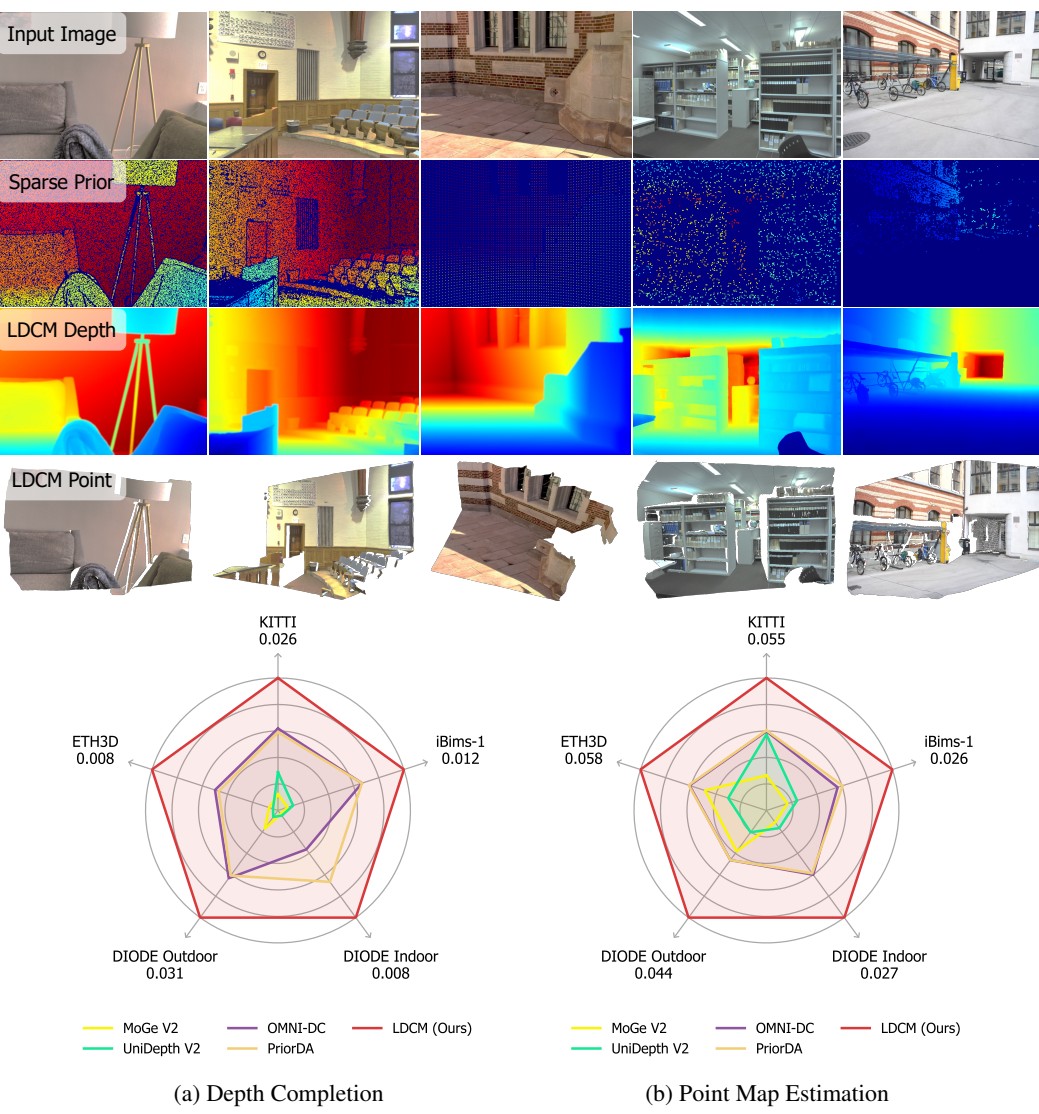

Figure 1: We present LDCM, a simple and effective model for depth completion. Without complex module design, LDCM achieves state-of-the-art performance in zero-shot depth completion and metric point map estimation. On the leaderboard, larger areas indicate lower relative error (REL). LDCM ranks first across diverse datasets.

*Internship at Tongyi Lab
†Corresponding author

## ABSTRACT

This work presents the Large Depth Completion Model (LDCM), a simple, effective, and robust framework for single-view metric depth estimation with sparse observations. Without relying on complex architectural designs, LDCM generates metric-accurate dense depth maps use a transformer. It outperforms existing approaches across diverse datasets and sparse observations. We achieve this from two key perspectives: (1) leveraging existing monocular foundation models to improve the quality of sparse depth inputs, and (2) reformulating training objectives to better capture geometric structure and metric consistency. Specifically, a Poisson-based depth initialization strategy is firstly introduced to generate a uniform coarse dense depth map from diverse sparse observations, providing a strong structural prior for the network. Regarding the training objective, we replace the conventional depth head with a point map head that regresses per-pixel 3D coordinates in camera space, enabling the model to directly learn the underlying 3D scene structure instead of performing pixel-wise depth map restoration. Moreover, this design eliminates the need for camera intrinsic parameters, allowing LDCM to naturally produce metric-scaled 3D point maps. Extensive experiments demonstrate that LDCM consistently outperforms state-of-the-art methods across multiple benchmarks and varying sparsity levels in both depth completion and point map estimation, showcasing its effectiveness and strong generalization to unseen data distributions.

## 1 INTRODUCTION

Dense depth maps are essential for applications in robotics Wang et al. (2024c), autonomous driving An et al. (2022), and augmented reality Krajancich et al. (2020). However, capturing dense and accurate depth data requires expensive active sensors such as LiDAR or structured light cameras, which are often limited by cost and hardware constraints. Thus, depth completion, which estimates a dense depth map from low-cost sparse depth observations and a corresponding RGB image, provides a cost-effective and efficient alternative.

While prior approaches Cheng et al. (2018; 2019); Yan et al. (2022; 2025a); Park et al. (2020); Yu et al. (2023); Zhou et al. (2023) perform well on in-domain datasets such as NYUv2 Silberman et al. (2012) and KITTI Uhrig et al. (2017), they often fail to generalize to unseen environments and irregular sparse depth maps (e.g., Structure-from-Motion points with non-uniform density and large missing regions), limiting their real-world applicability. Driven by the success of foundation models trained on large-scale datasets Yang et al. (2024a;b); Yin et al. (2023); Hu et al. (2024), recent works Zuo et al. (2024); Wang et al. (2023a; 2025g; 2024a; 2025a) have focused on architectural innovations and training with larger, more diverse data to improve robustness under domain shifts and varying sparsity. More recently, inspired by advances in natural language models Achiam et al. (2023); Yang et al. (2025), prompt-based approaches Lin et al. (2025); Viola et al. (2024); Liu et al. (2024); Park et al. (2024); Wang et al. (2025g) treat the sparse depth map as a conditioning signal for transformer-based Yang et al. (2024a;b) or diffusion-based Ke et al. (2024); Viola et al. (2024); Liu et al. (2024) depth foundation models, guiding the prediction toward metric-scale geometry. Despite their promising results, these methods fundamentally address depth completion as a depth restoration task, where the model learns to interpolate or denoise depth values conditioned on the sparse observation. This paradigm prioritizes local smoothness and texture-aware completion but lacks explicit 3D geometric reasoning, leading to unsatisfactory performance under severe domain shifts and highly irregular sparse depth maps.

In this work, we introduce the Large Depth Completion Model (LDCM), which produces dense, metric-accurate depth maps even from highly sparse and irregular observations. We achieve this by enhancing the input preprocessing pipeline and reformulating the training objective. To address the challenge of sparse and irregular depth maps, we leverage a monocular depth foundation model Yang et al. (2024a;b) to enrich the geometric prior. Specifically, we construct a dense gradient field by combining sparse depth map with relative depth cues predicted by the foundation model. We demonstrate that this hybrid gradient field serves as an proxy for solving a Poisson-based optimization problem, enabling the reconstruction of an initial coarse depth map that preserves fine

geometric structures and exhibits metric-consistent depth values. Regarding the training objective, we replace the conventional depth regression head with a point map regression head, inspired by recent advances in 3D reconstruction Wang et al. (2024b); Leroy et al. (2024); Wang et al. (2025b); Fang et al. (2025). This reformulation explicitly encourages the network to predict metric-scale 3D coordinates, rather than focusing on pixel-wise restoration. The final depth map is obtained by extracting the z-component of the predicted point map, leading to more geometrically faithful and globally consistent predictions. Moreover, benefiting from this design, LDCM naturally predicts 3D point maps without requiring camera intrinsics, facilitating robust deployment in uncalibrated environments.

We perform extensive experiments to evaluate LDCM across six diverse benchmarks. The results demonstrate that our model surpasses all previous state-of-the-art methods in both depth completion and point map estimation, achieving top rankings across all tasks and metrics, as displayed in Fig. 1. Our contribution can be summarized as follows:

- We propose the Large Depth Completion Model (LDCM), which replaces the conventional depth regression head with a point map regression head to directly predict metric-scale 3D coordinates from a monocular image and sparse observations. This formulation facilitates more effective learning of metric-consistent 3D structures, leading to superior performance in dense depth completion.

- We introduce a Poisson-based coarse depth completion strategy that leverages relative depth cues from a monocular depth foundation model and sparse observations. This strategy generates high-quality initial depth maps, providing a geometrically faithful structural prior for subsequent feature learning.

- We demonstrate through extensive experiments that LDCM outperforms previous state-of-the-art methods in both depth completion and metric point map estimation across diverse benchmarks and varying sparsity levels, showcasing its robust generalization to unseen data.

## 2 RELATED WORK

**Depth Completion.** Depth completion aims to infer a dense depth map from a monocular image and a sparse depth map, which can be readily obtained from sources such as Structure-from-Motion Schops et al. (2017) or low-cost depth cameras Silberman et al. (2012). Recent deep learning-based approaches have achieved significant progress by proposing numerous spatial propagation network variants Liu et al. (2017); Cheng et al. (2018; 2019); Park et al. (2020); Lin et al. (2022) or exploiting visual structural guidance from images for guided restoration. To better exploit the 3D geometric information in sparse inputs, several 2D-3D joint depth completion approaches have also been proposed Yu et al. (2023); Yan et al. (2024; 2025b); Zhou et al. (2023). Despite achieving impressive performance on single-domain datasets (e.g., NYUv2 Silberman et al. (2012) and KITTI Uhrig et al. (2017)), these methods often struggle with cross-domain generalization, particularly when deployed in unseen environments and varying sparse observations.

Inspired by the success of foundation models Kirillov et al. (2023); Oquab et al. (2023); Yang et al. (2024a;b); Yin et al. (2023); Hu et al. (2024); Wang et al. (2025a) trained on large-scale datasets, recent works Zuo et al. (2024); Wang et al. (2023a; 2024a; 2025g) have focused on architectural innovations and training with larger, more diverse datasets to improve generalization. More recently, drawing inspiration from large language models Achiam et al. (2023); Yang et al. (2025), prompt-based approaches Lin et al. (2025); Viola et al. (2024); Park et al. (2024); Jeong et al. (2025) have emerged that treat auxiliary priors as prompts to condition depth foundation models, effectively guiding predictions toward metric-scale outputs. PromptDA Lin et al. (2025) introduces a compact prompt fusion architecture specifically designed for the DPT head Ranftl et al. (2021), enabling the integration of low-resolution depth cues. TestPromptDC Jeong et al. (2025) presents a test-time prompt tuning method that adapts foundation models during inference without modifying their parameters, achieving sensor-specific depth scale adaptation while preserving foundational knowledge. MarigoldDC Viola et al. (2024) prompts the sparse depth to a diffusion-based Ke et al. (2024) foundation model. However, these methods fundamentally address depth completion as a depth restoration task, where the model learns to interpolate or denoise depth values conditioned on

sparse inputs. The performance remains unsatisfactory under severe domain shifts and highly irregular sparse depth maps. In this work, we introduce a Poisson-based depth initialization module to effectively maximize the potential of depth foundation models to generate a coarse dense depth map, which serves as a strong structural prior for the following geometric feature learning. Besides, we reformulate the training objective as point maps, providing a more structurally faithful supervision for the network.

**Monocular Depth Estimation.** A variety of monocular depth estimation foundation models Yang et al. (2024a;b); Piccinelli et al. (2024; 2025); Yin et al. (2023); Ke et al. (2024); Wang et al. (2025d;c) have been proposed. These models learn rich, generalizable priors from large-scale data and serve as strong backbones for downstream tasks such as stereo matching Wen et al. (2025); Jiang et al. (2025a); Cheng et al. (2025), depth super-resolution Yan et al. (2025c), depth completion Park et al. (2024); Lin et al. (2025); Liu et al. (2024); Viola et al. (2024); Wang et al. (2025g), and autonomous driving Yu et al. (2024); Li et al. (2025); Yu et al. (2025); Li et al. (2023a); An et al. (2022). For instance, FoundationStereo Wen et al. (2025) introduces a side-tuning feature adapter that leverages monocular priors to bridge the sim-to-real domain gap. DuCos Yan et al. (2025c) treats foundation model outputs as structural priors for depth super-resolution (DSR) and seamlessly integrates them into a Lagrangian duality framework. PriorDA Wang et al. (2025g) employs a local weighted linear regression (LWLR) module Xu et al. (2022) to align the scale of relative depth with sparse observations, where the result is then refined by a structure-aware network to produce dense depth map. However, this local alignment strategy often fails under highly sparse observations. In contrast, we propose a novel Poisson-based initialization strategy to better exploit the potential of foundation models by enforcing gradient consistency constraints, yielding a significantly more geometrically coherent coarse depth map.

**Geometry Estimation Foundation Models.** Point map Wang et al. (2024b; 2025b;e); Fang et al. (2025); Gao et al. (2025); Jang et al. (2025) representation has demonstrated strong potential for holistic scene understanding. Unlike depth maps, which indeedly encode 2.5D geometry tied to camera intrinsics, point maps explicitly model 3D structure. Several approaches Yin et al. (2021); Piccinelli et al. (2024; 2025) decouple this task into depth prediction and camera parameter estimation. In contrast, DUSt3R Wang et al. (2024b) bypasses explicit camera modeling by directly regressing a scale-invariant point map in an end-to-end fashion, with its successor Mast3R Leroy et al. (2024) enabling metric-scale reconstruction. VGGT Wang et al. (2025b) introduces a feedforward neural network capable of 3D reconstruction from one, a few, or even hundreds of input views of a scene. AnySplat Jiang et al. (2025b) extends VGGT Wang et al. (2025b) to support novel view synthesis from uncalibrated image collections. To facilitate single-view geometry learning, MoGe Wang et al. (2025e;f) predicts an affine-invariant point map and recovers metric scale using a global scaling factor derived from contextual cues. More recently, several approaches Liu et al. (2025); Keetha et al. (2025); Jang et al. (2025) have introduced additional priors to enhance geometry estimation. Notably, Pow3R Jang et al. (2025) extends the DUSt3R Wang et al. (2024b) paradigm by incorporating complementary modalities; however, it remains limited to relative geometry. In this work, we introduce point map representations for depth completion, enabling the model to directly learn the underlying 3D scene structure and produce metric quantities. Our concurrent work, MapAnything Keetha et al. (2025), also estimates metric 3D geometry from images and additional priors.

## 3 METHOD

### 3.1 OVERALL FRAMEWORK

The framework of the proposed LDCM is illustrated in Fig. 2. Given an RGB image $\mathbf{I} \in \mathbb{R}^{H \times W \times 3}$ and a sparse depth map $\mathbf{S} \in \mathbb{R}^{H \times W}$, LDCM predicts a metric point map $\mathbf{P} \in \mathbb{R}^{H \times W \times 3}$ in camera space, from which the dense depth map is derived by extracting the z-channel component. The framework consists of two main stages. In the first stage, we harness the power of monocular depth foundation model to generate an initial coarse depth map $\mathbf{C} \in \mathbb{R}^{H \times W}$ via Poisson reconstruction. In the second stage, a ViT-based Dosovitskiy et al. (2020) depth completion network takes the image $\mathbf{I}$ and the coarse depth $\mathbf{C}$ as input to predict the final metric 3D point map $\mathbf{P}$. The details of each stage are elaborated in the following sections.

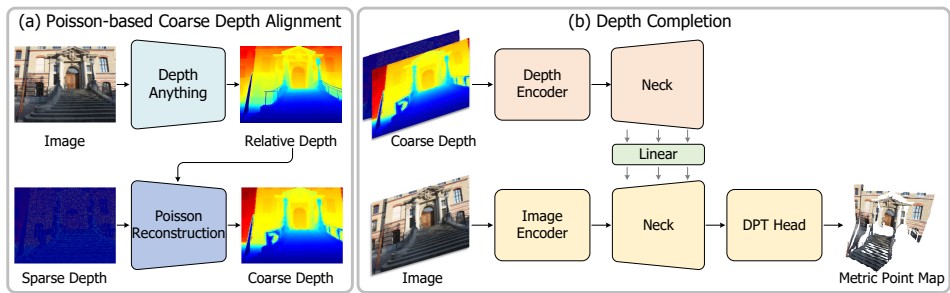

Figure 2: Schematics and detailed architecture of LDCM. Given a single image and sparse depth map, LDCM begins with a Poisson-based coarse depth alignment strategy. This process leverages a pretrained depth foundation model to generate an initial coarse depth map through gradient-domain optimization. This coarse depth, together with the input image, is then fed into the following point map prediction network to regress a dense, metric-scale 3D point map.

## 3.2 COARSE DEPTH ALIGNMENT

Different types of sparse depth priors exhibit distinct spatial distributions, ranging from random points and Structure-from-Motion keypoints to LiDAR-like structured sparsity, posing significant challenges for generalization. A straightforward approach involves direct interpolation of the sparse depth map Liu et al. (2024); however, it often introduces severe artifacts due to the absence of strong geometric priors. With the advent of depth foundation models Ranftl et al. (2020; 2021), which capture scene-level structure from large-scale training, leveraging them to provide robust geometric guidance has emerged as a promising direction.

To integrate sparse observations with foundation model predictions, we evaluate several coarse alignment strategies, including global affine alignment, local weighted linear regression (LWLR), and Poisson-based optimization. While the former two offer simple parametric alignment, they exhibit critical limitations. Global affine alignment assumes a uniform scale and shift across the entire image, making it unable to recover per-pixel metric values. LWLR improves spatial adaptivity by fitting local models, but its performance is highly sensitive to the distribution and density of sparse depth maps. In contrast, Poisson-based optimization formulates the alignment as a gradient-field reconstruction problem, demonstrating superior geometric coherence and metric accuracy across diverse sparse observations. Therefore, we adopt Poisson reconstruction in the first stage of LDCM to generate the coarse depth map $\mathbf{C}$.

Specifically, given a sparse depth input $\mathbf{S}$ and relative depth cues $\mathbf{D}_r$ from a foundation model, we aim to generate a coarse dense depth map $\mathbf{C}$ that aligns with the geometric structure of $\mathbf{D}_r$, while preserving the observed values in $\mathbf{S}$. The problem can be formulated as minimizing the following optimzation function:

$$\mathbf{C} = \arg\min_{\mathbf{D}} \left( \sum_i \|\nabla \log \mathbf{D}_i - \mathbf{G}_i\|^2 + \lambda \sum_{i \in \Omega} (\mathbf{D}_i - \mathbf{S}_i)^2 \right), \tag{1}$$

where $\mathbf{G}$ is a target log-gradient field that encodes structural fidelity and metric consistency, $\Omega$ denotes the set of valid sparse depth points, and $\lambda$ balances two terms. A naive choice is $\mathbf{G} = \nabla \log \mathbf{D}_r$, but this ignores the unknown scale and shift of relative depth and may lead to misaligned gradients in metric space. Instead, we construct a more informed target by incorporating metric priors from sparse observations. Let $(\alpha, \beta)$ be the global affine transformation that best aligns $\mathbf{D}_r$ with $\mathbf{S}$:

$$(\alpha, \beta) = \arg\min_{\alpha', \beta'} \sum_{i \in \Omega} (\mathbf{S}_i - \alpha' \cdot (\mathbf{D}_r)_i - \beta')^2, \tag{2}$$

and define $\gamma = \beta/\alpha$. We then set:

$$\mathbf{G} = \nabla \log(\mathbf{D}_r + \gamma). \tag{3}$$

This choice is motivated by the fact that during training, the relative depth ground truth $\mathbf{D}_r$ is derived from the metric ground truth $\mathbf{D}^*$ via an affine transformation: $\mathbf{D}_r = (\mathbf{D}^* - \beta)/\alpha$. While this ideal

relationship may not strictly hold for the predicted $\mathbf{D}_r$ during inference, introducing a shift $\gamma$ helps align its gradient structure with the metric space. Empirically, $\nabla \log(\mathbf{D}_r + \gamma)$ serves as a robust proxy for the target log-gradient field, preserving fine geometric details while being anchored to the metric scale through sparse inputs. Thus, the final formulation becomes:

$$\mathbf{C} = \arg\min_{\mathbf{D}} \left( \sum_i \|\nabla \log \mathbf{D}_i - \nabla \log(\mathbf{D}_r + \gamma)_i\|^2 + \lambda \sum_{i \in \Omega} (\mathbf{D}_i - \mathbf{S}_i)^2 \right), \qquad (4)$$

which can be solved through the conjugate gradient method Hestenes et al. (1952). In this formulation, each sparse point anchors the global energy, and due to the nature of gradient-domain reconstruction, its influence propagates across the entire image via the structural constraints encoded in the gradient field.

### 3.3 DEPTH COMPLETION NETWORK

The architecture is illustrated in Fig. 2(b). We employ dual encoders to extract features from the coarse depth map $\mathbf{C}$ and the RGB image, respectively. Features are fused using the prompt fusion block Lin et al. (2025). For the final output, instead of regressing a depth map, we replace the standard depth regression head with a point map head that directly predicts per-pixel 3D coordinates $\mathbf{P}$. This enables the model to learn the underlying 3D scene structure holistically, rather than performing pixel-wise depth restoration. Ablation studies demonstrate that this design leads to better accuracy. Moreover, thanks to this end-to-end formulation, the model naturally produces metric 3D point maps, facilitating robust deployment in uncalibrated environments.

### 3.4 TRAINING

**Training Losses.** We train the LDCM using three complementary losses on the predicted 3D point map $\mathbf{P}$, with the ground-truth metric point map denoted as $\hat{\mathbf{P}}$.

$$\mathcal{L} = \mathcal{L}_{\text{global}} + \lambda_{\text{local}} \mathcal{L}_{\text{local}} + \lambda_{\text{normal}} \mathcal{L}_{\text{normal}}, \qquad (5)$$

where the individual terms are defined as follows. The global point map loss enforces overall structural consistency:

$$\mathcal{L}_{\text{global}} = \sum_{i \in \mathcal{M}} \frac{1}{\hat{\mathbf{D}}_i} \|\mathbf{P}_i - \hat{\mathbf{P}}_i\|_1, \qquad (6)$$

where $\mathcal{M}$ denotes the region of valid ground-truth. The local point map loss captures fine-grained geometry by operating in 3D neighborhoods. Following Wang et al. (2025e), we sample anchor points and define spherical regions $\mathcal{S}_j$ in 3D space:

$$\mathcal{L}_{\text{local}} = \sum_{j \in \mathcal{H}_a} \sum_{i \in \mathcal{S}_j} \frac{1}{\hat{\mathbf{D}}_i} \|\mathbf{P}_i - \hat{\mathbf{P}}_i\|_1. \qquad (7)$$

This encourages local coherence independent of image perspective. The normal loss promotes surface smoothness and alignment:

$$\mathcal{L}_{\text{normal}} = \sum_{i \in \mathcal{M}} \arccos\left( \frac{\mathbf{N}_i^\top \hat{\mathbf{N}}_i}{\|\mathbf{N}_i\|\|\hat{\mathbf{N}}_i\|} \right), \qquad (8)$$

where $\mathbf{N}_i$ and $\hat{\mathbf{N}}_i$ are surface normals estimated from $\mathbf{P}$ and $\hat{\mathbf{P}}$, respectively.

**Implementation Details.** We train the LDCM on 11 public RGB-D datasets Roberts et al. (2021); Wang et al. (2020; 2019); Zheng et al. (2023); Gómez et al. (2025); Wrenninge & Unger (2018); Li et al. (2023b); LightwheelAI & contributors (2024); Huang et al. (2018); Ros et al. (2016); Yeshwanth et al. (2023), approximately 2.7 million samples. The combined data covers diverse indoor and outdoor scenes; further details are provided in the *suppl. material*.

LDCM uses a ViT-B Dosovitskiy et al. (2020) pretrained with DINOv2 Oquab et al. (2023) as the image encoder. For coarse depth alignment, we use DepthAnythingV2-S Yang et al. (2024b) as the foundation model. Training runs for 200 K iterations using the AdamW optimizer Loshchilov &

Table 1: **Quantitative comparison of depth completion methods on benchmark datasets.** All methods are evaluated under zero-shot settings. Methods marked with † produce relative depth, and metric depth is recovered by optimizing global scale and shift via least squares regression using the *same* sparse depth prior. Methods marked with ‡ use dataset-specific configurations for indoor and outdoor scenes, respectively. The best and second-best results are highlighted.

| Method | KITTI | | | | | iBims-1 | | | | | DIODE Indoor | | | | |
|---|---|---|---|---|---|---|---|---|---|---|---|---|---|---|---|
| | RMSE↓ | MAE↓ | REL↓ | $\delta_1$↑ | Rk.↓ | RMSE↓ | MAE↓ | REL↓ | $\delta_1$↑ | Rk.↓ | RMSE↓ | MAE↓ | REL↓ | $\delta_1$↑ | Rk.↓ |
| DepthPro | 4.149 | 2.763 | 0.178 | 0.731 | 13.432 | 0.605 | 0.503 | 0.156 | 0.829 | 13.750 | 0.837 | 0.702 | 0.193 | 0.668 | 14.114 |
| UniDepth V1 | 3.335 | 2.010 | 0.118 | 0.938 | 8.636 | 1.166 | 1.082 | 0.370 | 0.236 | 16.000 | 0.939 | 0.840 | 0.158 | 0.779 | 13.523 |
| UniDepth V2 | 3.150 | 1.598 | 0.090 | 0.960 | 6.500 | 0.446 | 0.321 | 0.100 | 0.935 | 11.932 | 0.811 | 0.678 | 0.165 | 0.681 | 13.023 |
| DepthAnythingV2† | 4.007 | 1.890 | 0.092 | 0.916 | 9.091 | 0.349 | 0.179 | 0.043 | 0.975 | 8.295 | 0.386 | 0.189 | 0.045 | 0.976 | 7.295 |
| VGGT† | 4.219 | 2.518 | 0.158 | 0.783 | 12.909 | 0.348 | 0.194 | 0.053 | 0.957 | 10.318 | 0.425 | 0.294 | 0.096 | 0.920 | 10.773 |
| MoGe V1† | 3.050 | 1.821 | 0.125 | 0.887 | 8.568 | 0.238 | 0.120 | 0.035 | 0.981 | 6.045 | 0.272 | 0.175 | 0.064 | 0.950 | 7.386 |
| MoGe V2 | 4.617 | 3.366 | 0.213 | 0.458 | 15.182 | 0.633 | 0.540 | 0.156 | 0.707 | 14.500 | 1.064 | 0.938 | 0.235 | 0.433 | 15.841 |
| G2-MonoDepth‡ | 2.638 | 0.964 | 0.054 | 0.949 | 5.295 | 0.227 | 0.094 | 0.028 | 0.973 | 5.409 | 0.298 | 0.198 | 0.067 | 0.879 | 6.341 |
| OMNI-DC | 2.302 | 0.760 | 0.042 | 0.963 | 3.045 | 0.192 | 0.063 | 0.018 | 0.982 | 2.932 | 0.141 | 0.064 | 0.022 | 0.968 | 2.932 |
| PriorDA | 2.364 | 0.861 | 0.044 | 0.971 | 4.159 | 0.176 | 0.065 | 0.018 | 0.990 | 3.477 | 0.093 | 0.037 | 0.012 | 0.994 | 3.023 |
| SPNet‡ | 2.365 | 0.757 | 0.041 | 0.966 | 3.000 | 0.189 | 0.059 | 0.016 | 0.987 | 2.659 | 0.157 | 0.078 | 0.028 | 0.954 | 3.273 |
| PromptDA | 3.040 | 1.261 | 0.067 | 0.946 | 6.545 | 0.249 | 0.116 | 0.033 | 0.975 | 6.091 | 0.203 | 0.115 | 0.037 | 0.965 | 6.068 |
| WorldMirror† | 4.439 | 2.432 | 0.142 | 0.824 | 11.818 | 0.352 | 0.192 | 0.051 | 0.963 | 9.205 | 0.386 | 0.243 | 0.084 | 0.941 | 9.364 |
| MapAnything | 12.974 | 6.784 | 0.350 | 0.588 | 15.750 | 0.968 | 0.374 | 0.104 | 0.909 | 13.295 | 0.909 | 0.458 | 0.104 | 0.899 | 11.000 |
| Pow3R† | 3.515 | 2.096 | 0.141 | 0.832 | 10.750 | 0.338 | 0.183 | 0.049 | 0.965 | 9.091 | 0.353 | 0.240 | 0.078 | 0.943 | 9.000 |
| LDCM (Ours) | 1.911 | 0.537 | 0.026 | 0.983 | 1.068 | 0.161 | 0.044 | 0.012 | 0.991 | 1.659 | 0.084 | 0.025 | 0.008 | 0.993 | 1.545 |

| Method | DIODE Outdoor | | | | | ETH3D | | | | | Average | | | | |
|---|---|---|---|---|---|---|---|---|---|---|---|---|---|---|---|
| | RMSE↓ | MAE↓ | REL↓ | $\delta_1$↑ | Rk.↓ | RMSE↓ | MAE↓ | REL↓ | $\delta_1$↑ | Rk.↓ | RMSE↓ | MAE↓ | REL↓ | $\delta_1$↑ | Rk.↓ |
| DepthPro | 9.539 | 7.635 | 0.403 | 0.177 | 14.636 | 3.199 | 2.562 | 0.302 | 0.477 | 15.023 | 3.666 | 2.833 | 0.246 | 0.576 | 14.191 |
| UniDepth V1 | 5.782 | 3.841 | 0.189 | 0.661 | 11.795 | 3.482 | 3.170 | 0.579 | 0.116 | 15.728 | 2.941 | 2.189 | 0.283 | 0.546 | 13.136 |
| UniDepth V2 | 11.145 | 8.936 | 0.515 | 0.526 | 15.250 | 1.630 | 1.169 | 0.200 | 0.726 | 13.387 | 3.436 | 2.540 | 0.214 | 0.766 | 12.018 |
| DepthAnythingV2† | 5.940 | 2.777 | 0.124 | 0.869 | 8.659 | 2.091 | 0.424 | 0.049 | 0.979 | 9.477 | 2.555 | 1.092 | 0.071 | 0.943 | 8.563 |
| VGGT† | 4.898 | 2.893 | 0.237 | 0.772 | 10.591 | 0.540 | 0.317 | 0.060 | 0.949 | 9.103 | 2.086 | 1.243 | 0.121 | 0.876 | 10.739 |
| MoGe V1† | 10.576 | 8.340 | 0.406 | 0.599 | 14.250 | 1.651 | 0.550 | 0.082 | 0.943 | 8.750 | 3.157 | 2.201 | 0.142 | 0.872 | 9.000 |
| MoGe V2 | 4.807 | 3.352 | 0.182 | 0.680 | 10.477 | 0.847 | 0.619 | 0.114 | 0.839 | 11.784 | 2.394 | 1.763 | 0.180 | 0.623 | 13.557 |
| G2-MonoDepth‡ | 2.393 | 0.875 | 0.062 | 0.938 | 4.682 | 0.428 | 0.177 | 0.034 | 0.969 | 5.603 | 1.197 | 0.462 | 0.049 | 0.942 | 5.466 |
| OMNI-DC | 2.322 | 0.726 | 0.049 | 0.955 | 3.341 | 0.290 | 0.100 | 0.016 | 0.987 | 2.932 | 1.049 | 0.343 | 0.029 | 0.971 | 3.036 |
| PriorDA | 2.310 | 0.858 | 0.051 | 0.957 | 3.932 | 0.274 | 0.105 | 0.017 | 0.990 | 3.443 | 1.043 | 0.385 | 0.028 | 0.980 | 3.607 |
| SPNet‡ | 2.111 | 0.658 | 0.048 | 0.959 | 2.114 | 0.419 | 0.119 | 0.019 | 0.986 | 3.625 | 1.048 | 0.334 | 0.030 | 0.970 | 2.934 |
| PromptDA | 3.604 | 1.561 | 0.087 | 0.912 | 6.182 | 0.644 | 0.276 | 0.041 | 0.967 | 7.102 | 1.548 | 0.666 | 0.053 | 0.953 | 6.398 |
| WorldMirror† | 4.464 | 2.317 | 0.151 | 0.828 | 8.045 | 0.524 | 0.302 | 0.051 | 0.962 | 7.761 | 2.033 | 1.097 | 0.096 | 0.904 | 9.239 |
| MapAnything | 7.675 | 3.891 | 0.219 | 0.731 | 11.318 | 1.952 | 0.711 | 0.108 | 0.904 | 12.523 | 4.896 | 2.444 | 0.177 | 0.806 | 12.777 |
| Pow3R† | 3.682 | 2.068 | 0.169 | 0.840 | 7.568 | 0.480 | 0.273 | 0.048 | 0.964 | 7.545 | 1.674 | 0.972 | 0.097 | 0.909 | 8.791 |
| LDCM (Ours) | 1.969 | 0.529 | 0.031 | 0.970 | 1.568 | 0.187 | 0.048 | 0.008 | 0.997 | 1.148 | 0.862 | 0.237 | 0.017 | 0.987 | 1.398 |

Hutter (2017) with a cosine learning rate schedule and linear warmup over the first 5% of iterations. The peak learning rates are $1 \times 10^{-5}$ for the encoder and $1 \times 10^{-4}$ for all other layers. We use a global batch size of 128, with mini-batches sampling an approximately equal number of images from each dataset. During training, images are resized such that their aspect ratios range uniformly from $1 : 2$ to $2 : 1$, and total pixel counts fall between $250\,\mathrm{K}$ and $500\,\mathrm{K}$. Data augmentation includes random cropping, color jittering, Gaussian blur, JPEG compression-decompression, and perspective-aware cropping to align the principal point with the image center. Sparse depth inputs are synthetically generated by subsampling dense ground-truth depth maps with varying patterns, following the protocol of OMNI-DC Zuo et al. (2024). The training is conducted on 16 H20 GPUs and takes approximately six days to complete.

## 4 EXPERIMENTS

### 4.1 QUANTITATIVE EVALUATIONS

We evaluate the zero-shot performance of LDCM and compare it with several state-of-the-art approaches for depth completion Wang et al. (2023a); Zuo et al. (2024); Wang et al. (2024a); Lin et al. (2025); Wang et al. (2025g), monocular depth estimation Yang et al. (2024a;b); Wang et al. (2025b); Bochkovskiy et al. (2025), and monocular point map estimation Piccinelli et al. (2024; 2025); Wang et al. (2025e;f); Liu et al. (2025); Jang et al. (2025); Keetha et al. (2025). Additional details on the compared approaches and evaluation protocols are provided in the *suppl. material*. As demonstrated in the experiments, LDCM achieves superior performance across multiple benchmarks.

**Depth completion.** We evaluate depth completion on KITTI Uhrig et al. (2017), ETH3D Schops et al. (2017), iBims-1 Koch et al. (2018), and DIODE Vasiljevic et al. (2019), covering both indoor and outdoor scenarios. To assess robustness under diverse sparse sampling patterns, we synthesize sparse depth inputs using the following strategies:

- **Noisy random sampling**: uniformly sampled points at varying densities (e.g., 1%, 3%, 5%, 10%), with mild noise simulation;

- **Keypoint-based sampling**: depth values extracted at SIFT or ORB keypoints;

- **LiDAR-simulated sampling**: synthetic LiDAR scans with varying numbers of vertical beams (e.g., 64, 32, 16 lines).

On KITTI, the simulation is applied to raw single-frame LiDAR measurements Zuo et al. (2024); Wang et al. (2023b). For all other datasets, they are generated from dense ground-truth depth maps. We evaluate the predicted depth maps using standard metrics: Root Mean Squared Error (RMSE), Mean Absolute Error (MAE), Relative Error (REL), and the accuracy threshold $\delta_1$. For methods that produce relative depth maps Wang et al. (2025b); Yang et al. (2024b); Wang et al. (2025e), we recover the global scale and shift via least squares regression using the sparse depth prior. Table 1 reports the average RMSE, MAE, REL, and $\delta_1$ across all synthetic patterns per dataset, along with the mean ranking over competing methods. As shown in the table, LDCM achieves state-of-the-art performance. Notably, it maintains high accuracy even under extreme sparsity, demonstrating strong robustness and generalization across diverse sparse input configurations.

Table 2: **Quantitative comparison of point map estimation methods on benchmark datasets.** All methods are evaluated under zero-shot settings. The best and second-best results are highlighted.

| Method | KITTI | | | | | iBims-1 | | | | | DIODE Indoor | | | | |
|---|---|---|---|---|---|---|---|---|---|---|---|---|---|---|---|
| | $MAE^p$ ↓ | $RMSE^p$ ↓ | $REL^p$ ↓ | $\delta_1^p$ ↑ | Rk.↓ | $MAE^p$ ↓ | $RMSE^p$ ↓ | $REL^p$ ↓ | $\delta_1^p$ ↑ | Rk.↓ | $MAE^p$ ↓ | $RMSE^p$ ↓ | $REL^p$ ↓ | $\delta_1^p$ ↑ | Rk.↓ |
| UniDepth V1 | 2.207 | 3.540 | 0.120 | 0.954 | 6.773 | 1.154 | 1.239 | 0.370 | 0.239 | 9.000 | 0.911 | 1.017 | 0.159 | 0.779 | 7.318 |
| UniDepth V2 | 1.813 | 3.540 | 0.096 | 0.961 | 5.409 | 0.365 | 0.489 | 0.107 | 0.932 | 6.909 | 0.730 | 0.872 | 0.164 | 0.694 | 7.273 |
| MoGe V2 | 3.536 | 4.899 | 0.208 | 0.484 | 9.000 | 0.574 | 0.667 | 0.156 | 0.740 | 8.000 | 1.048 | 1.185 | 0.242 | 0.410 | 8.955 |
| G2-MonoDepth | 1.669 | 3.118 | 0.098 | 0.946 | 4.841 | 0.186 | 0.287 | 0.052 | 0.972 | 4.750 | 0.305 | 0.401 | 0.087 | 0.875 | 4.841 |
| OMNI-DC | 1.542 | 2.828 | 0.092 | 0.960 | 3.409 | 0.164 | 0.256 | 0.046 | 0.980 | 3.341 | 0.174 | 0.241 | 0.045 | 0.967 | 2.977 |
| PriorDA | 1.573 | 2.836 | 0.091 | 0.965 | 4.341 | 0.159 | 0.240 | 0.043 | 0.989 | 3.500 | 0.140 | 0.190 | 0.034 | 0.994 | 2.909 |
| SPNet | 1.507 | 2.881 | 0.089 | 0.964 | 3.068 | 0.152 | 0.239 | 0.042 | 0.988 | 2.455 | 0.172 | 0.236 | 0.046 | 0.963 | 2.636 |
| PromptDA | 1.933 | 3.612 | 0.110 | 0.938 | 6.659 | 0.199 | 0.309 | 0.054 | 0.975 | 5.545 | 0.204 | 0.301 | 0.056 | 0.963 | 5.523 |
| ' LDCM (Ours) | 1.027 | 2.308 | 0.055 | 0.982 | 1.045 | 0.092 | 0.194 | 0.026 | 0.992 | 1.000 | 0.127 | 0.179 | 0.027 | 0.992 | 1.159 |

| Method | DIODE Outdoor | | | | | ETH3D | | | | | Average | | | | |
|---|---|---|---|---|---|---|---|---|---|---|---|---|---|---|---|
| | $MAE^p$ ↓ | $RMSE^p$ ↓ | $REL^p$ ↓ | $\delta_1^p$ ↑ | Rk.↓ | $MAE^p$ ↓ | $RMSE^p$ ↓ | $REL^p$ ↓ | $\delta_1^p$ ↑ | Rk.↓ | $MAE^p$ ↓ | $RMSE^p$ ↓ | $REL^p$ ↓ | $\delta_1^p$ ↑ | Rk.↓ |
| UniDepth V1 | 4.653 | 5.100 | 0.461 | 0.145 | 9.000 | 3.541 | 3.875 | 0.551 | 0.106 | 9.000 | 2.493 | 2.954 | 0.332 | 0.445 | 8.218 |
| UniDepth V2 | 1.879 | 2.844 | 0.216 | 0.712 | 8.000 | 1.252 | 1.785 | 0.191 | 0.769 | 8.000 | 1.208 | 1.906 | 0.155 | 0.814 | 7.118 |
| MoGe V2 | 0.931 | 1.206 | 0.115 | 0.890 | 5.977 | 0.716 | 0.913 | 0.119 | 0.865 | 6.409 | 1.361 | 1.774 | 0.168 | 0.678 | 7.668 |
| G2-MonoDepth | 0.794 | 1.129 | 0.109 | 0.891 | 4.864 | 0.603 | 0.826 | 0.105 | 0.911 | 5.160 | 0.711 | 1.152 | 0.090 | 0.919 | 4.891 |
| OMNI-DC | 0.714 | 0.946 | 0.095 | 0.915 | 2.795 | 0.550 | 0.710 | 0.095 | 0.929 | 3.284 | 0.629 | 0.996 | 0.075 | 0.950 | 3.161 |
| PriorDA | 0.698 | 0.908 | 0.095 | 0.919 | 3.295 | 0.538 | 0.682 | 0.094 | 0.936 | 3.352 | 0.622 | 0.971 | 0.071 | 0.961 | 3.479 |
| SPNet | 0.733 | 1.243 | 0.100 | 0.914 | 3.932 | 0.557 | 0.859 | 0.096 | 0.931 | 3.796 | 0.624 | 1.092 | 0.075 | 0.952 | 3.177 |
| PromptDA | 0.824 | 1.422 | 0.100 | 0.911 | 5.591 | 0.592 | 0.950 | 0.093 | 0.932 | 4.159 | 0.750 | 1.319 | 0.083 | 0.944 | 5.495 |
| LDCM (Ours) | 0.427 | 0.580 | 0.044 | 0.995 | 1.000 | 0.347 | 0.456 | 0.058 | 0.996 | 1.000 | 0.404 | 0.743 | 0.042 | 0.991 | 1.041 |

**Point map estimation.** We adopt the same benchmarks used for depth completion to evaluate monocular point map estimation. The predicted point maps are evaluated using point-wise metrics: $RMSE^p$, $MAE^p$, $REL^p$ and $\delta_1^p$. Table 2 reports the average performance across all synthetic patterns per dataset for each metric. For depth completion methods, we use the camera intrinsics from UniDepth V2 Piccinelli et al. (2025) to back-project the completed depth maps into 3D point maps. As shown in the table, LDCM consistently outperforms all competing methods, achieving the best results across all datasets and metrics.

**Affine-invariant point map estimation.** We adopt the same benchmarks to evaluate monocular affine-invariant point map estimation. Following MoGe Wang et al. (2025e), we resolve the scale and shift of the predicted point map using the proposed ROE solver to align it with the ground truth. Table 3 reports the average performance in terms of $REL^p$ and $\delta_1^p$. As shown in the table, our method achieves superior performance compared to baseline approaches and outperforms state-of-the-art relative geometry estimation methods, including VGGT Wang et al. (2025b) and WorldMirror Liu et al. (2025). This demonstrates that our model preserves—rather than compromises—the accuracy of relative geometry estimation.

## 4.2 ABLATION STUDY

We conduct ablation studies to evaluate the effectiveness of the Poisson-based coarse depth alignment strategy and the training objectives. For simplicity, we adopt LiDAR-simulated sparse patterns (64, 32, 16, and 8 lines) on outdoor datasets, and keypoint-based sampling on indoor datasets.

Table 3: **Quantitative comparison of affine-invariant point map estimation methods on benchmark datasets.** All methods are evaluated under zero-shot settings. The best and second-best results are highlighted.

| Method | KITTI | | | iBims-1 | | | DIODE Indoor | | |
|---|---|---|---|---|---|---|---|---|---|
| | $REL^p \downarrow$ | $\delta_1^p \uparrow$ | Rk.$\downarrow$ | $REL^p \downarrow$ | $\delta_1^p \uparrow$ | Rk.$\downarrow$ | $REL^p \downarrow$ | $\delta_1^p \uparrow$ | Rk.$\downarrow$ |
| VGGT | 0.147 | 0.823 | 4.500 | 0.048 | 0.967 | 3.909 | 0.107 | 0.926 | 4.636 |
| MoGe V2 | 0.056 | 0.968 | 1.909 | 0.046 | 0.972 | 2.455 | 0.052 | 0.972 | 1.955 |
| WorldMirror | 0.108 | 0.886 | 3.136 | 0.044 | 0.965 | 2.864 | 0.073 | 0.953 | 3.091 |
| MapAnything | 0.366 | 0.344 | 6.000 | 0.233 | 0.611 | 6.000 | 0.172 | 0.758 | 6.000 |
| Pow3R | 0.152 | 0.850 | 4.318 | 0.064 | 0.965 | 4.318 | 0.108 | 0.947 | 4.273 |
| LDCM (Ours) | 0.039 | 0.983 | 1.091 | 0.017 | 0.992 | 1.000 | 0.014 | 0.995 | 1.000 |

| Method | DIODE Outdoor | | | ETH3D | | | Average | | |
|---|---|---|---|---|---|---|---|---|---|
| | $REL^p \downarrow$ | $\delta_1^p \uparrow$ | Rk.$\downarrow$ | $REL^p \downarrow$ | $\delta_1^p \uparrow$ | Rk.$\downarrow$ | $REL^p \downarrow$ | $\delta_1^p \uparrow$ | Rk.$\downarrow$ |
| VGGT | 0.215 | 0.700 | 5.000 | 0.053 | 0.978 | 3.591 | 0.114 | 0.879 | 4.327 |
| MoGe V2 | 0.124 | 0.841 | 2.000 | 0.044 | 0.980 | 2.637 | 0.064 | 0.947 | 2.191 |
| WorldMirror | 0.155 | 0.788 | 3.045 | 0.049 | 0.976 | 3.023 | 0.086 | 0.914 | 3.032 |
| MapAnything | 0.302 | 0.501 | 6.000 | 0.265 | 0.549 | 6.000 | 0.268 | 0.553 | 6.000 |
| Pow3R | 0.197 | 0.750 | 3.955 | 0.074 | 0.982 | 3.796 | 0.119 | 0.899 | 4.132 |
| LDCM (Ours) | 0.077 | 0.949 | 1.000 | 0.039 | 0.994 | 1.728 | 0.037 | 0.983 | 1.164 |

Table 4: ablation study on the coarse depth alignment strategy. We report the relative error (REL) for coarse depth and final prediction. The best and second-best results are highlighted.

| Configuration | Corse Depth (REL $\downarrow$) | | | | | Estimated Depth (REL $\downarrow$) | | | | |
|---|---|---|---|---|---|---|---|---|---|---|
| | KITTI | iBims-1 | DIODE | ETH3D | Average | KITTI | iBims-1 | DIODE | ETH3D | Average |
| Sparse | - | - | - | - | - | 0.021 | 0.029 | 0.040 | 0.026 | 0.029 |
| Global alignment | 0.095 | 0.075 | 0.102 | 0.078 | 0.087 | 0.020 | 0.019 | 0.035 | 0.023 | 0.024 |
| LWLR | 0.078 | 0.108 | 0.108 | 0.061 | 0.088 | 0.019 | 0.022 | 0.036 | 0.021 | 0.025 |
| Poisson w/o global alignment | 0.069 | 0.208 | 0.174 | 0.138 | 0.147 | - | - | - | - | - |
| Poisson | 0.033 | 0.073 | 0.088 | 0.044 | 0.059 | 0.019 | 0.018 | 0.033 | 0.019 | 0.022 |

**Coarse Depth Alignment Strategy.** We ablate various coarse depth alignment strategies for robust geometric guidance. First, we assess the accuracy of the generated coarse depth maps. As shown on the left side of Table 4, Poisson-based alignment achieves the best performance, demonstrating its effectiveness. Notably, global alignment is essential—its omission leads to a significant performance drop. By comparison, LWLR performs worse than even simple global alignment under extreme sparsity, highlighting its sensitivity to sparse and irregular inputs. A qualitative ablation example is provided in Fig. 3, where the Poisson-based method not only achieves the highest accuracy but also best preserves geometric structure. On the right side of Table 4, we use these coarse depth maps as inputs to our completion model; again, the Poisson-based variant yields the best results.

Table 5: Ablation study on the output representation. We report the relative error (REL) for depth completion and $REL^p$ for point map estimation. The best and second-best results are highlighted.

| Configuration | Depth Completion (REL $\downarrow$) | | | | | Point Map Estimation ($REL^p \downarrow$) | | | | |
|---|---|---|---|---|---|---|---|---|---|---|
| | KITTI | iBims-1 | DIODE | ETH3D | Average | KITTI | iBims-1 | DIODE | ETH3D | Average |
| SI-Log Depth | 0.023 | 0.023 | 0.037 | 0.021 | 0.026 | - | - | - | - | - |
| SI-Log Depth + Ray map | 0.022 | 0.022 | 0.038 | 0.021 | 0.026 | 0.073 | 0.050 | 0.084 | 0.097 | 0.067 |
| Point Map | 0.019 | 0.018 | 0.033 | 0.019 | 0.022 | 0.047 | 0.032 | 0.070 | 0.059 | 0.045 |

**Output Representation.** We ablate the output representation by replacing the point map with either a conventional depth map or the concatenation of depth and dense ray maps (depth + ray map). As shown in Table 5, both alternatives lead to performance degradation, demonstrating that the point map provides more effective 3D structural guidance than depth-based representations.

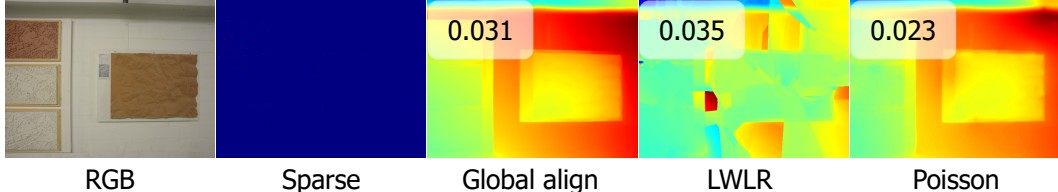

Figure 3: Qualitative comparison between three coarse alignment stragies. We report the relative error for each result.

### 4.3 DEPTH COMPLETION RESULTS ON STANDARD BENCHMARKS

Table 6: **Quantitative comparison of depth completion methods on real-pattern benchmark datasets.** All methods are evaluated under zero-shot settings. Methods marked with † produce relative depth, and metric depth is recovered by optimizing global scale and shift via least squares regression using the same sparse depth prior. Methods marked with ‡ use dataset-specific configurations for indoor and outdoor scenes, respectively. The best and second-best results are highlighted.

| Method | NYUv2 | | | | | VOID | | | | | ETH3D | | | | |
|---|---|---|---|---|---|---|---|---|---|---|---|---|---|---|---|
| | RMSE↓ | MAE↓ | REL↓ | $\delta_1$↑ | Rk.↓ | RMSE↓ | MAE↓ | REL↓ | $\delta_1$↑ | Rk.↓ | RMSE↓ | MAE↓ | REL↓ | $\delta_1$↑ | Rk.↓ |
| DepthPro | 0.332 | 0.253 | 0.096 | 0.929 | 15.000 | 0.759 | 0.396 | 0.189 | 0.726 | 14.833 | 3.199 | 2.562 | 0.302 | 0.477 | 14.750 |
| UniDepth V1 | 0.213 | 0.148 | 0.056 | 0.981 | 10.375 | 0.651 | 0.267 | 0.107 | 0.902 | 12.083 | 3.482 | 3.170 | 0.579 | 0.116 | 15.625 |
| UniDepth V2 | 0.293 | 0.218 | 0.085 | 0.948 | 14.000 | 0.651 | 0.269 | 0.115 | 0.900 | 13.000 | 1.630 | 1.169 | 0.200 | 0.726 | 13.000 |
| DepthAnythingV2† | 0.220 | 0.128 | 0.045 | 0.977 | 11.250 | 0.605 | 0.214 | 0.063 | 0.958 | 8.250 | 1.915 | 0.493 | 0.063 | 0.963 | 6.500 |
| VGGT† | 0.168 | 0.087 | 0.033 | 0.985 | 7.000 | 0.572 | 0.196 | 0.064 | 0.952 | 6.750 | 0.650 | 0.432 | 0.095 | 0.906 | 6.375 |
| MoGe V1† | 0.180 | 0.093 | 0.037 | 0.979 | 8.750 | 0.577 | 0.200 | 0.064 | 0.952 | 7.500 | 2.877 | 0.450 | 0.108 | 0.924 | 6.500 |
| MoGe V2 | 0.261 | 0.186 | 0.070 | 0.963 | 13.000 | 0.779 | 0.421 | 0.202 | 0.557 | 15.833 | 0.847 | 0.619 | 0.114 | 0.839 | 9.375 |
| G2-MonoDepth‡ | 0.166 | 0.071 | 0.026 | 0.985 | 7.125 | 0.607 | 0.195 | 0.055 | 0.942 | 7.500 | 1.425 | 0.525 | 0.136 | 0.886 | 10.375 |
| OMNI-DC | 0.147 | 0.053 | 0.020 | 0.987 | 4.375 | 0.574 | 0.168 | 0.040 | 0.962 | 4.167 | 0.822 | 0.317 | 0.079 | 0.925 | 4.625 |
| PriorDA | 0.122 | 0.047 | 0.017 | 0.993 | 2.750 | 0.571 | 0.171 | 0.039 | 0.968 | 3.333 | 0.671 | 0.260 | 0.061 | 0.962 | 2.500 |
| SPNet‡ | 0.127 | 0.047 | 0.017 | 0.992 | 2.500 | 0.578 | 0.178 | 0.054 | 0.959 | 5.250 | 1.299 | 0.372 | 0.092 | 0.943 | 6.625 |
| PromptDA | 0.162 | 0.079 | 0.028 | 0.989 | 6.000 | 0.565 | 0.182 | 0.049 | 0.965 | 4.000 | 0.911 | 0.483 | 0.090 | 0.896 | 6.875 |
| WorldMirror† | 0.217 | 0.121 | 0.042 | 0.979 | 10.125 | 0.596 | 0.208 | 0.067 | 0.946 | 9.833 | 0.898 | 0.668 | 0.153 | 0.836 | 9.250 |
| MapAnything | 0.724 | 0.327 | 0.132 | 0.885 | 16.000 | 0.782 | 0.282 | 0.110 | 0.900 | 13.750 | 2.283 | 0.874 | 0.150 | 0.863 | 12.375 |
| Pow3R† | 0.155 | 0.081 | 0.031 | 0.988 | 5.625 | 0.571 | 0.196 | 0.067 | 0.949 | 7.333 | 0.881 | 0.656 | 0.154 | 0.833 | 9.875 |
| LDCM (Ours) | 0.113 | 0.037 | 0.013 | 0.994 | 1.000 | 0.536 | 0.145 | 0.028 | 0.977 | 1.000 | 0.445 | 0.154 | 0.035 | 0.978 | 1.250 |

To further evaluate zero-shot depth completion under real-world sparse patterns, we follow prior work in evaluating methods on the NYUv2 Silberman et al. (2012), VOID Wong et al. (2020), and ETH3D Schops et al. (2017) datasets. For NYUv2, we adopt the sampling protocol from OMNI-DC Zuo et al. (2024), extracting 500 and 100 sparse depth points per image, respectively. For VOID, we use the provided sparse depth maps derived from a visual-inertial odometry system, which come in three sparsity levels: 1500, 500, and 150 points per frame. For ETH3D, we project the sparse 3D points from COLMAP SfM reconstructions into the image plane to generate sparse depth maps. Table 6 reports the quantitative results on each dataset. As shown in the table, LDCM significantly outperforms all comparison methods, ranking first on all the datasets.

## 5 CONCLUSION

We have presented the Large Depth Completion Model (LDCM), a simple yet powerful framework for metric depth estimation from sparse observations. LDCM is both effective and robust, leveraging a Poisson-based alignment strategy to maximize the potential of existing monocular foundation models by preprocessing input sparse observations into strong geometric priors for subsequent feature learning. Furthermore, LDCM replaces the conventional depth map representation with a point map representation, enabling direct learning of the underlying 3D structure rather than per-pixel depth restoration. Our method achieves superior zero-shot performance across multiple benchmarks, demonstrating robustness under varying sparse observation patterns. Moreover, the point map design allows LDCM to naturally output metric-scaled 3D geometry without requiring camera intrinsics, facilitating reliable deployment in uncalibrated environments. We believe LDCM marks a significant advancement in depth completion and can serve as a robust foundational model for downstream 3D vision tasks.

## ACKNOWLEDGMENTS

This work was supported in part by the National Natural Science Foundation of China under grant 62301484, in part by the Jinhua Science and Technology Bureau Project under grant 2026-1-022, in part by the Young Talent Fund of Zhejiang Association for Science and Technology under grant ZJSKXQT2026135, and in part by the Ningbo Natural Science Foundation of China under grant 2024J454.

## ETHICS STATEMENT

Our study focuses on depth completion, a core problem in the field of computer vision. The experimental evaluation is based exclusively on public datasets that have been curated without inclusion of any personally identifiable or sensitive content. We assert that this research has been carried out in accordance with the code of ethics.

## REPRODUCIBILITY STATEMENT

To facilitate verification and extension of our work, we include the implementation code in the supplementary materials. Furthermore, we provide key training and evaluation procedures in the paper, and will make the complete code and trained models publicly available after publication to support full experimental reproducibility.

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

APPENDIX

# A DATASETS

## A.1 TRAINING DATASETS

We collected 11 open-source RGB-D datasets to train LDCM, comprising 10 synthetic and 1 real-world dataset. An overview of the training datasets is provided in Table 7, spanning four distinct domains: indoor, outdoor, in-the-wild, and driving scenarios. The combined training set contains approximately 2.6 million images. The number of RGB-D pairs in each dataset may slightly differ from the originally released versions, as we manually excluded some invalid frames.

Table 7: An overview of the training datasets.

| Dataset | Domain | Statistic | Type |
|---|---|---|---|
| Hypersim Roberts et al. (2021) | Indoor | 75K | Synthetic |
| TartanAir Wang et al. (2020) | In-the-wild | 306K | Synthetic |
| IRS Wang et al. (2019) | Indoor | 101K | Synthetic |
| PointOdyssey Zheng et al. (2023) | Indoor | 303K | Synthetic |
| UrbanSyn Gómez et al. (2025) | Outdoor/Driving | 7K | Synthetic |
| Synscapes Wrenninge & Unger (2018) | Outdoor/Driving | 25K | Synthetic |
| MatrixCity Li et al. (2023b) | Outdoor/Driving | 424K | Synthetic |
| LightwheelOcc LightwheelAI & contributors (2024) | Outdoor/Driving | 204K | Synthetic |
| MVS-Synth Huang et al. (2018) | Outdoor/Driving | 12K | Synthetic |
| Synthia Ros et al. (2016) | Outdoor/Driving | 140K | Synthetic |
| ScanNet++ Yeshwanth et al. (2023) | Indoor | 1M | Real |
| Total | - | 2.6M | - |

## A.2 EVALUATION DATASETS

We use six datasets that are excluded from the training set for to compare the performance between LDCM and previous state-of-the-art methods. Below, we provide details for each dataset.

**NYUv2 Dataset.** The NYUv2 dataset Silberman et al. (2012) is an indoor dataset captured using a Microsoft Kinect sensor, containing RGB and depth sequences from 464 indoor scenes. The official test split contains 654 samples. Following Marigold Ke et al. (2024), we crop the images to a resolution of $426 \times 560$ for consistent input dimensions.

**KITTI Dataset.** The KITTI Depth dataset Geiger et al. (2012); Uhrig et al. (2017) is a large-scale outdoor dataset collected from a moving vehicle. The official validation split consists of 1,000 samples. Depth maps are acquired using an HDL-64 LiDAR sensor, with raw depth maps containing fewer than $6\%$ valid pixels. The provided ground truth is generated by fusing multiple consecutive LiDAR scans, resulting in a denser depth map with approximately $14\%$ valid pixels. For depth completion, input images are center-cropped to the bottom region of $252 \times 1216$ to exclude the sky and regions with unreliable depth due to the limited vertical field of view of the LiDAR.

**DIODE Dataset.** The DIODE dataset Vasiljevic et al. (2019) contains thousands of high-resolution RGB images with accurate, dense, and long-range depth measurements, captured using a FARO Focus S350 laser scanner. The official validation split includes 3 indoor and 3 outdoor scenes, comprising 325 and 446 samples, respectively. To reduce noise at occlusion boundaries, we filter out depth values where the maximum relative difference to any neighboring pixel exceeds $5\%$ (indoor) and $15\%$ (outdoor). Input images are resized to $480 \times 640$.

**iBims-1 Dataset.** The iBims-1 dataset Koch et al. (2018) is an indoor benchmark captured in diverse environments, providing high-resolution RGB images and highly accurate depth maps derived from laser scans. The official evaluation split contains 100 samples, with images at a native resolution of $480 \times 640$.

**VOID Dataset.** The VOID dataset Wong et al. (2020) is an indoor dataset captured using the Intel RealSense D435i camera. The official validation split consists of 800 samples, each paired with sparse depth maps at three sparsity levels (approximately 1500, 500, and 150 valid pixels) and RGB images at a resolution of $480 \times 640$. These varying sparsity levels allow for robust evaluation under different input conditions.

**ETH3D Dataset.** The ETH3D dataset Schops et al. (2017) consists of multi-view stereo images and dense depth maps captured using a high-precision laser scanner and DSLR cameras, covering diverse viewpoints and scene types. The official validation set contains 13 scenes with a total of 454 image pairs. The original image resolution is $4032 \times 6048$. Input images are resized to $480 \times 640$.

## B    EVALUATION DETAILS

### B.1    COMPARISON METHODS

We compare LDCM against a comprehensive set of state-of-the-art approaches: Depth-Pro[1] Bochkovskiy et al. (2025), UniDepth V1 & V2[2] Piccinelli et al. (2024; 2025), Depth Anything V2[3] Yang et al. (2024b), VGGT[4] Wang et al. (2025b), MoGe V1 & V2[5] Wang et al. (2025e;f), G2-MonoDepth[6] Wang et al. (2023a), OMNI-DC[7] Zuo et al. (2024), PriorDA[8] Wang et al. (2025g), SPNet[9] Wang et al. (2024a), PromptDA[10] Lin et al. (2025), Marigold-DC[11] Viola et al. (2024), DepthLab[12] Liu et al. (2024), Pow3R[13] Jang et al. (2025), MapAnything[14] Keetha et al. (2025), WorldMirror[15] Liu et al. (2025), spanning the key tasks of monocular depth estimation, monocular geometry estimation, depth completion. All methods are evaluated using their publicly available implementations and pre-trained checkpoints. Notably, G2-MonoDepth Wang et al. (2023a) and SPNet Wang et al. (2024a) employ different configurations for indoor and outdoor scenarios, while LDCM and the remaining methods do not use scenario-specific hyperparameters. PromptDA Lin et al. (2025) is specifically designed to leverage dense, low-resolution priors; therefore, we apply Poisson surface reconstruction to the input sparse depth map to obtain a dense prior before inference. Pow3R Jang et al. (2025) and WorldMirror Liu et al. (2025) produce relative geometry, even when sparse depth priors are provided.

### B.2    EVALUATION PROTOCOL

To clarity the notations in this section:

- $\mathbf{P}$ and $\hat{\mathbf{P}}$ are the predicted and ground truth points, respectively.
- $\mathbf{D}$ and $\hat{\mathbf{D}}$ are the predicted and ground truth depths, which are the z-coordinate of corresponding points.
- $\mathcal{M}$ is the mask of valid ground truth.

**Depth Completion.** In the manuscript, we use four standard metrics for depth completion evaluation, including RMSE, MAE, REL, $\delta_1$. Formally, they are defined as follows:

---

[1] https://github.com/apple/ml-depth-pro.
[2] https://github.com/lpiccinelli-eth/UniDepth.
[3] https://github.com/DepthAnything/Depth-Anything-V2.
[4] https://github.com/facebookresearch/vggt.
[5] https://github.com/microsoft/MoGe.
[6] https://github.com/Wang-xjtu/G2-MonoDepth.
[7] https://github.com/princeton-vl/OMNI-DC.
[8] https://github.com/SpatialVision/Prior-Depth-Anything.
[9] https://github.com/Wang-xjtu/SPNet.
[10] https://github.com/DepthAnything/PromptDA.
[11] https://github.com/prs-eth/Marigold-DC.
[12] https://github.com/ant-research/DepthLab.
[13] https://github.com/naver/pow3r.
[14] https://github.com/facebookresearch/map-anything.
[15] https://github.com/Tencent-Hunyuan/HunyuanWorld-Mirror.

- Root mean square error (RMSE) (RMSE):

$$\sqrt{\frac{1}{|\mathcal{M}|} \sum_{i \in \mathcal{M}} (\hat{\mathbf{D}}_i - \mathbf{D}_i)^2} \tag{9}$$

- Mean absolute error (MAE):

$$\frac{1}{|\mathcal{M}|} \sum_{i \in \mathcal{M}} \left| \hat{\mathbf{D}}_i - \mathbf{D}_i \right| \tag{10}$$

- Mean relative error (REL):

$$\frac{1}{|\mathcal{M}|} \sum_{i \in \mathcal{M}} \frac{\left| \hat{\mathbf{D}}_i - \mathbf{D}_i \right|}{\hat{\mathbf{D}}_i} \tag{11}$$

- Thresholded accuracy ($\delta_1$):

$$\frac{1}{|\mathcal{M}|} \sum_{i \in \mathcal{M}} \max \left( \frac{\hat{\mathbf{D}}_i}{\mathbf{D}_i}, \frac{\mathbf{D}_i}{\hat{\mathbf{D}}_i} \right) < 1.25 \tag{12}$$

For models that produce relative depth maps $\mathbf{D}_r$, we first follow Equation 2 to compute $(\alpha, \beta)$, and then the metric depth maps are recoverd by:

$$\mathbf{D} = \alpha \cdot \mathbf{D}_r + \beta. \tag{13}$$

**Point Map Estimation.** For evaluating the reconstructed 3D point map, we adopt analogous metrics based on Euclidean distances between predicted and ground truth points. The metrics include $\mathrm{RMSE}^p$, $\mathrm{MAE}^p$, $\mathrm{REL}^p$, and $\delta_1^p$, defined as:

- Point-wise Root Mean Square Error ($\mathrm{RMSE}^p$):

$$\sqrt{\frac{1}{|\mathcal{M}|} \sum_{i \in \mathcal{M}} \left\| \hat{\mathbf{P}}_i - \mathbf{P}_i \right\|^2} \tag{14}$$

- Point-wise Mean Absolute Error ($\mathrm{MAE}^p$):

$$\frac{1}{|\mathcal{M}|} \sum_{i \in \mathcal{M}} \left\| \hat{\mathbf{P}}_i - \mathbf{P}_i \right\| \tag{15}$$

- Point-wise Mean Relative Error ($\mathrm{REL}^p$):

$$\frac{1}{|\mathcal{M}|} \sum_{i \in \mathcal{M}} \frac{\left\| \hat{\mathbf{P}}_i - \mathbf{P}_i \right\|}{\|\mathbf{P}_i\|} \tag{16}$$

- Point-wise Thresholded Accuracy ($\delta_1^p$):

$$\frac{1}{|\mathcal{M}|} \sum_{i \in \mathcal{M}} \left\| \hat{\mathbf{P}}_i - \mathbf{P}_i \right\| < 0.25 \cdot \min \left( \|\mathbf{P}_i\|, \|\hat{\mathbf{P}}_i\| \right) \tag{17}$$

**Affine-invariant Point Map Estimation.** To evaluate the affine-invariant point, we first compute the scale $\alpha_p$ and shift $\beta_p$ using the following equation, which recovers the affine transformation applied to the predicted point map. This equation can be solved efficiently using the ROE solver proposed by MoGe Wang et al. (2025e).

$$(\alpha_p, \beta_p) = \arg \min_{\alpha_p, \beta_p} \sum_{i \in \mathcal{M}} \left( \hat{\mathbf{P}}_i - \alpha_p \cdot \mathbf{P}_i - \beta_p \right)^2, \tag{18}$$

## C   MORE QUANTITATIVE RESULTS

From Table 12 to Table 29, we provide detailed quantitative results under different types of sparse observations.

Table 8: **Quantitative comparison of depth completion with diffusion-based methods on benchmark datasets.** The **best** results are in **bold**.

| Method | VOID-1500-Points | | | | VOID-500-Points | | | | VOID-150-Points | | | |
|---|---|---|---|---|---|---|---|---|---|---|---|---|
| | RMSE↓ | MAE↓ | REL↓ | $\delta_1$ ↑ | RMSE↓ | MAE↓ | REL↓ | $\delta_1$ ↑ | RMSE↓ | MAE↓ | REL↓ | $\delta_1$ ↑ |
| DepthLab | 0.577 | 0.162 | 0.034 | 0.969 | 0.572 | 0.183 | 0.053 | 0.941 | 0.688 | 0.249 | 0.083 | 0.901 |
| Marigold-DC | 0.553 | 0.154 | 0.031 | 0.975 | 0.536 | 0.162 | 0.043 | 0.965 | 0.626 | 0.199 | 0.053 | 0.955 |
| LDCM (Ours) | **0.528** | **0.135** | **0.021** | **0.981** | **0.501** | **0.134** | **0.027** | **0.978** | **0.580** | **0.167** | **0.035** | **0.972** |
| Method | NYUv2-500-Points | | | | NYUv2-100-Points | | | | KITTI-64-Lines | | | |
| | RMSE↓ | MAE↓ | REL↓ | $\delta_1$ ↑ | RMSE↓ | MAE↓ | REL↓ | $\delta_1$ ↑ | RMSE↓ | MAE↓ | REL↓ | $\delta_1$ ↑ |
| DepthLab | 0.118 | 0.041 | 0.015 | 0.993 | 0.213 | 0.100 | 0.037 | 0.976 | 2.032 | 0.828 | 0.061 | 0.962 |
| Marigold-DC | 0.116 | 0.040 | 0.014 | 0.993 | 0.157 | 0.061 | 0.022 | 0.988 | 1.931 | 0.818 | 0.054 | 0.971 |
| LDCM (Ours) | **0.094** | **0.028** | **0.009** | **0.996** | **0.131** | **0.045** | **0.016** | **0.992** | **1.240** | **0.292** | **0.016** | **0.993** |
| Method | KITTI-32-Lines | | | | KITTI-16-Lines | | | | AVERAGE | | | |
| | RMSE↓ | MAE↓ | REL↓ | $\delta_1$ ↑ | RMSE↓ | MAE↓ | REL↓ | $\delta_1$ ↑ | RMSE↓ | MAE↓ | REL↓ | $\delta_1$ ↑ |
| DepthLab | 2.250 | 0.893 | 0.064 | 0.959 | 2.748 | 0.932 | 0.066 | 0.953 | 1.150 | 0.424 | 0.052 | 0.957 |
| Marigold-DC | 2.155 | 0.875 | 0.057 | 0.968 | 2.546 | 0.981 | 0.062 | 0.963 | 1.078 | 0.411 | 0.042 | 0.972 |
| LDCM (Ours) | **1.416** | **0.332** | **0.018** | **0.991** | **1.603** | **0.393** | **0.020** | **0.990** | **0.762** | **0.191** | **0.020** | **0.987** |

## D  MORE COMPARISON RESULTS WITH DIFFUSION-BASED METHODS

Here, we present additional comparisons with diffusion-based models—Marigold-DC Viola et al. (2024) and DepthLab Liu et al. (2024). Due to their prohibitively long inference times, we evaluate these methods primarily on three benchmarks with varying levels of sparse input: NYUv2 Silberman et al. (2012) (500 and 100 points), VOID Wong et al. (2020) (150, 500, and 1500 points), and KITTI Geiger et al. (2012) (64, 32, and 16 scan lines). As shown in Table 8, LDCM consistently outperforms both Marigold-DC and DepthLab across all settings.

Table 9: Ablation study on the training data. We report the relative error (REL) for depth completion.

| Configuration | Depth Completion (REL ↓) | | | | |
|---|---|---|---|---|---|
| | KITTI | iBims-1 | DIODE | ETH3D | Average |
| w/ more real data | 0.020 | **0.017** | 0.035 | **0.018** | **0.022** |
| Ours | **0.019** | 0.018 | **0.033** | 0.019 | **0.022** |

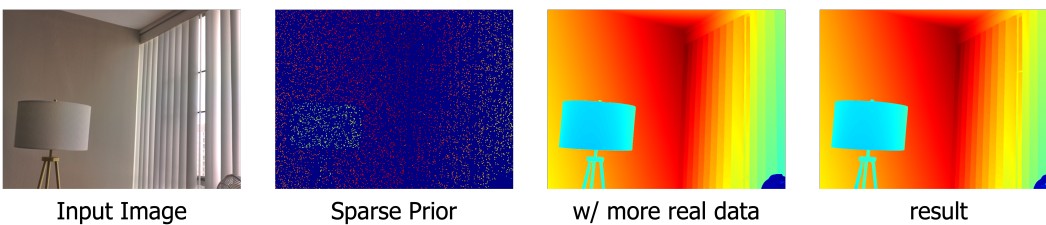

| Input Image | Sparse Prior | w/ more real data | result |

Figure 4: Qualitative comparison between the results from models using different training datasets.

## E  ABLATION ON THE TRAINING DATA

**Training Data.** We perform an ablation study on the training data used to train the LDCM. In addition to the original datasets, we introduce an extra dataset: DrivingStereo Yang et al. (2019). The quantitative results are presented in Table 9. As shown, the inclusion of this additional data does not significantly affect metric performance. However, as illustrated in Fig. 4, incorporating more real-world data leads to visually less sharp predictions, likely due to imperfect supervision signals in the added dataset.

Table 10: **Quantitative comparison of depth completion methods on benchmark datasets.** All methods are evaluated under zero-shot settings. Methods marked with † produce relative depth, and metric depth is recovered by optimizing global scale and shift via least squares regression using the *same* sparse depth prior. The best and second-best results are highlighted.

| Method | KITTI | | | | | iBims-1 | | | | | DIODE Indoor | | | | |
|---|---|---|---|---|---|---|---|---|---|---|---|---|---|---|---|
| | RMSE↓ | MAE↓ | REL↓ | $\delta_1$ ↑ | Rk.↓ | RMSE↓ | MAE↓ | REL↓ | $\delta_1$ ↑ | Rk.↓ | RMSE↓ | MAE↓ | REL↓ | $\delta_1$ ↑ | Rk.↓ |
| DepthAnythingV2† | 4.007 | 1.890 | 0.092 | 0.916 | 5.318 | 0.349 | 0.179 | 0.043 | 0.975 | 5.614 | 0.386 | 0.189 | 0.045 | 0.976 | 4.909 |
| DepthAnythingV2 w/ Poisson | 2.448 | 0.953 | 0.051 | 0.959 | 2.955 | 0.231 | 0.098 | 0.027 | 0.976 | 3.136 | 0.195 | 0.091 | 0.026 | 0.967 | 2.795 |
| VGGT† | 4.219 | 2.518 | 0.158 | 0.783 | 6.955 | 0.348 | 0.194 | 0.053 | 0.957 | 6.727 | 0.425 | 0.294 | 0.096 | 0.920 | 6.750 |
| VGGT w/ Poisson | 2.627 | 1.112 | 0.065 | 0.937 | 4.205 | 0.241 | 0.104 | 0.028 | 0.975 | 4.318 | 0.217 | 0.111 | 0.037 | 0.957 | 4.341 |
| MoGe V1† | 3.050 | 1.821 | 0.125 | 0.887 | 5.341 | 0.238 | 0.120 | 0.035 | 0.981 | 4.159 | 0.272 | 0.175 | 0.064 | 0.950 | 5.250 |
| MoGe V1 w/ Poisson | 2.179 | 0.865 | 0.050 | 0.959 | 2.136 | 0.214 | 0.089 | 0.025 | 0.977 | 2.409 | 0.177 | 0.085 | 0.028 | 0.965 | 2.614 |
| LDCM | 1.911 | 0.537 | 0.026 | 0.983 | 1.023 | 0.161 | 0.044 | 0.012 | 0.991 | 1.227 | 0.084 | 0.025 | 0.008 | 0.993 | 1.000 |

| Method | DIODE Outdoor | | | | | ETH3D | | | | | Average | | | | |
|---|---|---|---|---|---|---|---|---|---|---|---|---|---|---|---|
| | RMSE↓ | MAE↓ | REL↓ | $\delta_1$ ↑ | Rk.↓ | RMSE↓ | MAE↓ | REL↓ | $\delta_1$ ↑ | Rk.↓ | RMSE↓ | MAE↓ | REL↓ | $\delta_1$ ↑ | Rk.↓ |
| DepthAnythingV2† | 5.940 | 2.777 | 0.124 | 0.869 | 5.114 | 2.091 | 0.424 | 0.049 | 0.979 | 5.864 | 2.555 | 1.092 | 0.071 | 0.943 | 5.364 |
| DepthAnythingV2 w/ Poisson | 3.285 | 1.182 | 0.064 | 0.941 | 3.455 | 0.662 | 0.168 | 0.025 | 0.983 | 3.966 | 1.364 | 0.498 | 0.039 | 0.965 | 3.261 |
| VGGT† | 4.898 | 2.893 | 0.237 | 0.772 | 5.705 | 0.540 | 0.317 | 0.060 | 0.949 | 5.818 | 2.086 | 1.243 | 0.121 | 0.876 | 6.391 |
| VGGT w/ Poisson | 2.910 | 1.262 | 0.081 | 0.917 | 3.750 | 0.339 | 0.140 | 0.024 | 0.980 | 3.262 | 1.267 | 0.546 | 0.047 | 0.953 | 3.975 |
| MoGe V1† | 10.576 | 8.340 | 0.406 | 0.599 | 6.932 | 1.651 | 0.550 | 0.082 | 0.943 | 5.455 | 3.157 | 2.201 | 0.142 | 0.872 | 5.427 |
| MoGe V1 w/ Poisson | 2.340 | 0.910 | 0.053 | 0.950 | 2.000 | 0.319 | 0.118 | 0.021 | 0.986 | 2.262 | 1.046 | 0.413 | 0.035 | 0.967 | 2.284 |
| LDCM | 1.969 | 0.529 | 0.031 | 0.970 | 1.000 | 0.187 | 0.048 | 0.008 | 0.997 | 1.000 | 0.862 | 0.237 | 0.017 | 0.987 | 1.050 |

# F APPLYING POISSON-BASED ALIGNMENT STRATEGY TO MONOCULAR ESTIMATORS

In Table 10, we apply the Poisson alignment strategy to relative geometry estimators to obtain dense depth maps. As shown in the table, this strategy effectively improves the metric accuracy of these approaches, demonstrating its effectiveness. Moreover, our LDCM maintains state-of-the-art performance.

# G INFERENCE TIME

We report the per-stage inference times of our method, measured at a resolution of 480×640 on an NVIDIA L20 GPU. Our pipeline comprises four stages: Depth Anything Small (0.006 s), global alignment (0.006 s), Poisson-based alignment (0.020 s), and the subsequent refinement model (0.040 s), resulting in a total runtime of 0.072 s. For comparison, LWLR runs in 0.010 s under the same conditions. A detailed comparison with the inference times of several competing methods is provided in Table 11.

Table 11: Inference time (in seconds) of different methods at $480 \times 640$ resolution on an NVIDIA L20 GPU, with all inference performed in FP32 precision.

| Method | OMNI-DC | PriorDA | DepthPro | VGGT | MoGe V2 | DepthAnythingV2 | LDCM (Ours) |
|---|---|---|---|---|---|---|---|
| Inference Time (s) | 0.128 | 0.064 | 0.554 | 0.196 | 0.220 | 0.019 | 0.072 |

# H MORE QUALITATIVE RESULTS

Fig. 6 and Fig. 7 presents a qualitative comparison between LDCM and state-of-the-art methods. Notably, LDCM produces sharper geometric structures and more accurate depth distributions, particularly in regions with complex geometry or extreme sparsity. The predictions from LDCM exhibit significantly clearer boundaries and finer details, demonstrating the effectiveness of our coarse-to-fine framework and structural prior integration. In Fig. 8, we provide more visualization results for depth map and point map.

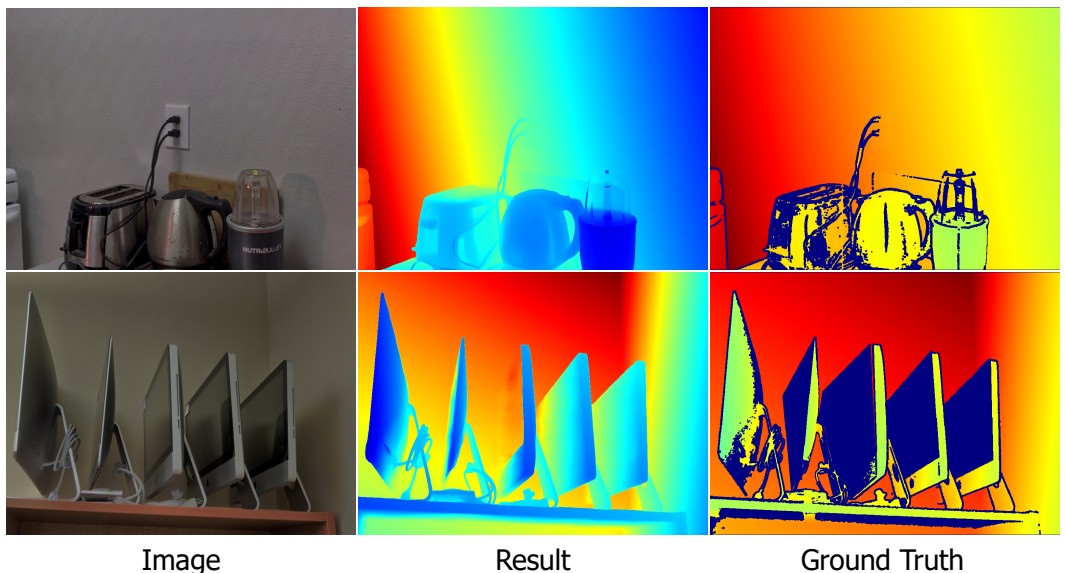

Image    Result    Ground Truth

Figure 5: Example of two failure cases.

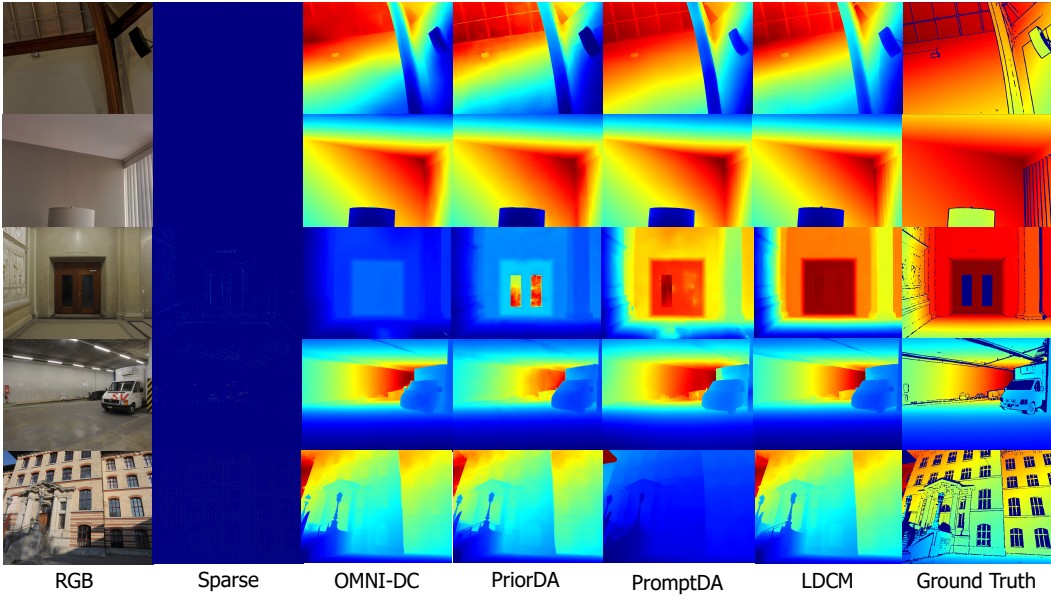

RGB Sparse OMNI-DC PriorDA PromptDA LDCM Ground Truth

Figure 6: Visualization comparion with state-of-the-art methods.

## I   NOISE ANALYSIS

Fig. 9 presents an example with noisy input. When the sparse prior contains noise, the Poisson alignment strategy is adversely affected. However, the subsequent network effectively mitigates this issue and produces a high-quality result.

## J   LIMTATION AND FUTURE WORK

Although LDCM achieves superior performance, accurately reconstructing transparent objects and reflective surfaces remains challenging, as illustrated by two failure cases in Fig. 5. This limitation

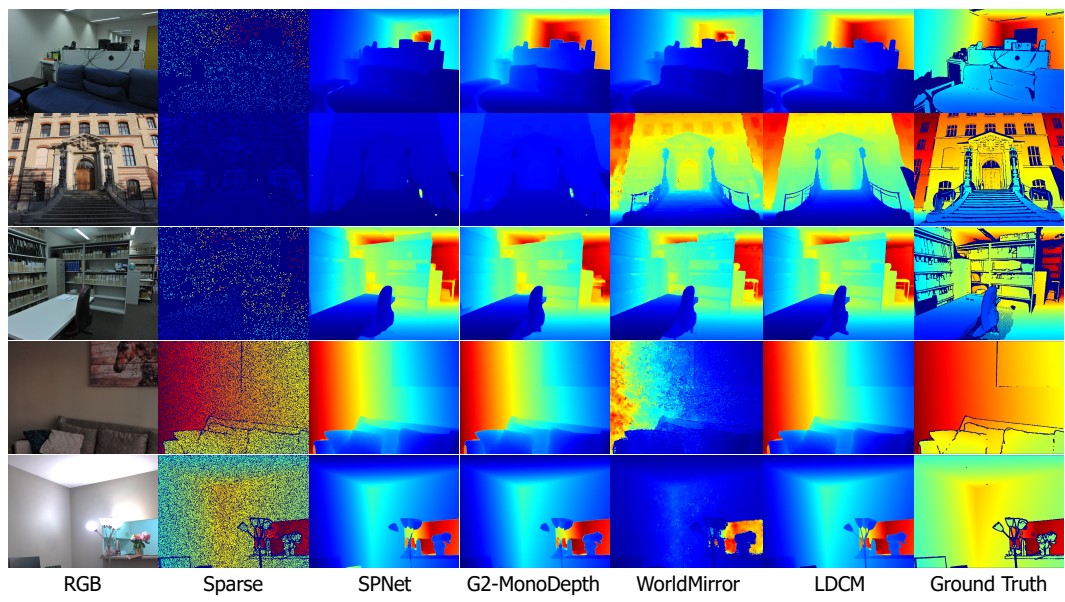

RGB    Sparse    SPNet    G2-MonoDepth    WorldMirror    LDCM    Ground Truth

Figure 7: Visualization comparion with state-of-the-art methods.

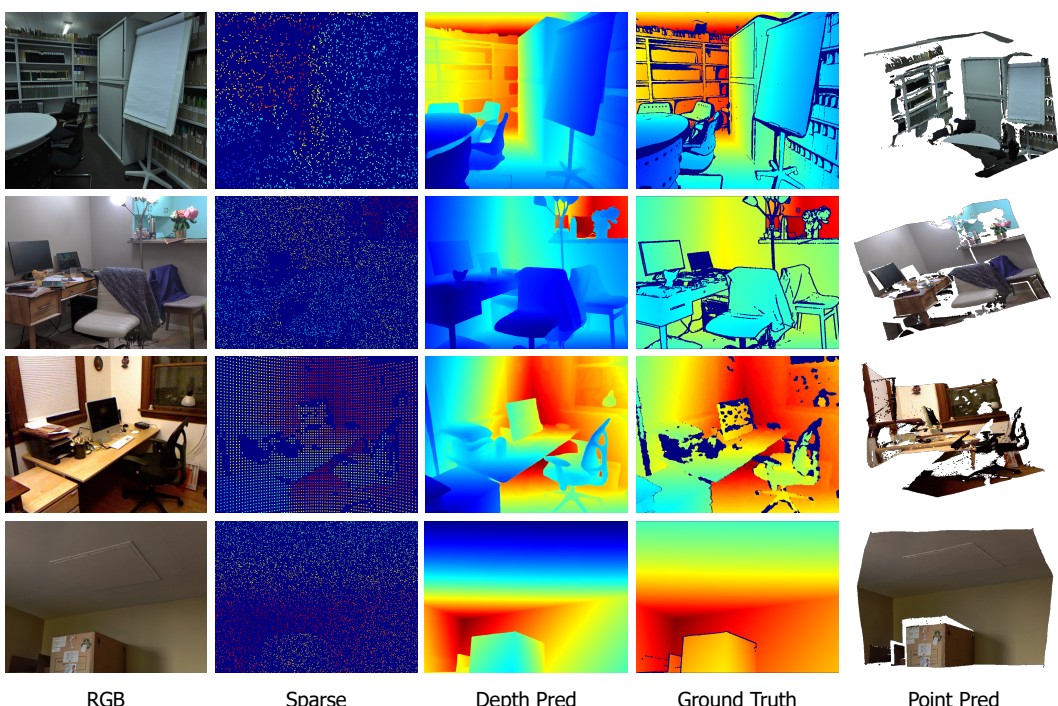

RGB    Sparse    Depth Pred    Ground Truth    Point Pred

Figure 8: More visualization results for depth map and point map.

stems from the lack of large-scale datasets containing such materials, which are difficult to capture and annotate. In the future, we plan to investigate synthetic data simulation to augment training and improve robustness on these challenging scenarios. Additionally, while monocular video reconstruction is a promising application, achieving temporal consistency poses substantial challenges. Extending LDCM to process video sequences for consistent 3D geometry estimation over time is an important direction for future exploration.

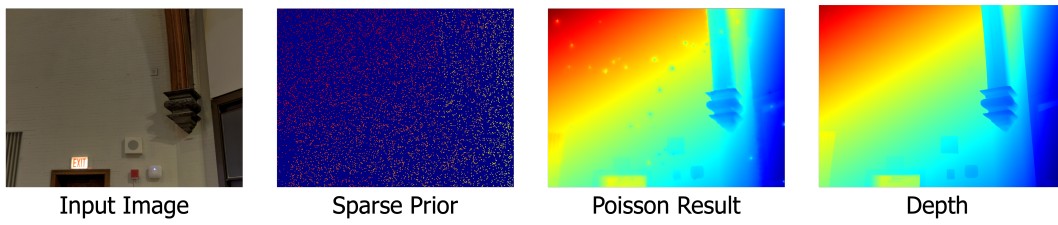

| Input Image | Sparse Prior | Poisson Result | Depth |

Figure 9: An example for noise input

## K STATEMENT ON THE USE OF LLMS

Large language models (LLMs) were used only for linguistic refinement, such as improving grammar and phrasing. They played no role in shaping research concepts, designing experiments, or interpreting data. The authors authored all content, verified its accuracy and originality, and assume full responsibility for the manuscript.

Table 12: **Quantitative comparison of depth completion with baseline methods on the KITTI dataset** Geiger et al. (2012); Uhrig et al. (2017). Methods marked with † produce relative depth maps, where the metric depth is recovered by optimizing global scale and shift via least squares regression using the sparse depth prior. Methods marked with ‡ use scenario-specific configurations for indoor and outdoor scenes, respectively. The best and the second best results are highlighted.

| method | Lidar-64-Lines | | | | | Lidar-32-Lines | | | | | Lidar-16-Lines | | | | |
|---|---|---|---|---|---|---|---|---|---|---|---|---|---|---|---|
| | RMSE↓ | MAE↓ | REL↓ | δ₁↑ | Rk.↓ | RMSE↓ | MAE↓ | REL↓ | δ₁↑ | Rk.↓ | RMSE↓ | MAE↓ | REL↓ | δ₁↑ | Rk.↓ |
| DepthPro | 4.149 | 2.763 | 0.178 | 0.731 | 14.000 | 4.149 | 2.763 | 0.178 | 0.731 | 14.000 | 4.149 | 2.763 | 0.178 | 0.731 | 13.750 |
| UniDepth V1 | 3.335 | 2.010 | 0.118 | 0.938 | 9.000 | 3.335 | 2.010 | 0.118 | 0.938 | 9.500 | 3.335 | 2.010 | 0.118 | 0.938 | 9.500 |
| UniDepth V2 | 3.150 | 1.598 | 0.090 | 0.960 | 7.500 | 3.150 | 1.598 | 0.090 | 0.960 | 7.250 | 3.150 | 1.598 | 0.090 | 0.960 | 7.500 |
| DepthAnythingV2† | 3.902 | 1.826 | 0.088 | 0.923 | 9.250 | 3.903 | 1.824 | 0.088 | 0.923 | 8.750 | 3.902 | 1.824 | 0.088 | 0.923 | 9.000 |
| VGGT† | 4.122 | 2.417 | 0.148 | 0.804 | 12.000 | 4.126 | 2.415 | 0.147 | 0.806 | 13.000 | 4.134 | 2.431 | 0.149 | 0.802 | 12.500 |
| MoGe V1† | 3.822 | 2.450 | 0.159 | 0.869 | 11.500 | 3.299 | 1.997 | 0.131 | 0.899 | 10.250 | 2.977 | 1.720 | 0.113 | 0.915 | 8.500 |
| MoGe V2 | 4.617 | 3.366 | 0.213 | 0.458 | 15.250 | 4.617 | 3.366 | 0.213 | 0.458 | 15.250 | 4.617 | 3.366 | 0.213 | 0.458 | 15.000 |
| G2-MonoDepth‡ | 1.609 | 0.378 | 0.024 | 0.986 | 4.000 | 1.801 | 0.454 | 0.028 | 0.984 | 4.000 | 2.187 | 0.652 | 0.036 | 0.981 | 4.250 |
| OMNI-DC | 1.184 | 0.274 | 0.015 | 0.993 | 1.000 | 1.424 | 0.354 | 0.019 | 0.990 | 2.000 | 1.710 | 0.460 | 0.024 | 0.986 | 2.000 |
| PriorDA | 1.776 | 0.561 | 0.029 | 0.985 | 5.000 | 1.912 | 0.645 | 0.034 | 0.983 | 5.000 | 2.124 | 0.773 | 0.041 | 0.979 | 4.750 |
| SPNet‡ | 1.547 | 0.369 | 0.023 | 0.987 | 3.000 | 1.774 | 0.418 | 0.026 | 0.985 | 3.000 | 2.069 | 0.531 | 0.031 | 0.982 | 3.000 |
| PromptDA | 2.409 | 0.857 | 0.043 | 0.973 | 6.000 | 2.490 | 0.886 | 0.043 | 0.972 | 6.000 | 2.682 | 0.993 | 0.050 | 0.966 | 6.000 |
| WorldMirror† | 3.740 | 2.245 | 0.159 | 0.793 | 11.500 | 4.076 | 2.078 | 0.121 | 0.883 | 11.250 | 5.017 | 2.543 | 0.134 | 0.832 | 13.000 |
| MapAnything | 12.431 | 6.241 | 0.318 | 0.621 | 15.750 | 12.621 | 6.357 | 0.322 | 0.616 | 15.750 | 12.856 | 6.551 | 0.331 | 0.606 | 15.750 |
| Pow3R† | 3.027 | 1.797 | 0.122 | 0.865 | 9.000 | 3.254 | 1.920 | 0.128 | 0.853 | 10.000 | 3.352 | 1.944 | 0.127 | 0.855 | 10.500 |
| LDCM (Ours) | 1.240 | 0.292 | 0.016 | 0.993 | 1.750 | 1.416 | 0.332 | 0.018 | 0.991 | 1.000 | 1.603 | 0.393 | 0.020 | 0.990 | 1.000 |

| method | Lidar-8-Lines | | | | | Lidar-4-Lines | | | | | 10% | | | | |
|---|---|---|---|---|---|---|---|---|---|---|---|---|---|---|---|
| | RMSE↓ | MAE↓ | REL↓ | δ₁↑ | Rk.↓ | RMSE↓ | MAE↓ | REL↓ | δ₁↑ | Rk.↓ | RMSE↓ | MAE↓ | REL↓ | δ₁↑ | Rk.↓ |
| DepthPro | 4.149 | 2.763 | 0.178 | 0.731 | 13.250 | 4.149 | 2.763 | 0.178 | 0.731 | 13.250 | 4.149 | 2.763 | 0.178 | 0.731 | 14.000 |
| UniDepth V1 | 3.335 | 2.010 | 0.118 | 0.938 | 9.500 | 3.335 | 2.010 | 0.118 | 0.938 | 8.500 | 3.335 | 2.010 | 0.118 | 0.938 | 9.250 |
| UniDepth V2 | 3.150 | 1.598 | 0.090 | 0.960 | 7.500 | 3.150 | 1.598 | 0.090 | 0.960 | 5.000 | 3.150 | 1.598 | 0.090 | 0.960 | 7.500 |
| DepthAnythingV2† | 3.927 | 1.838 | 0.089 | 0.922 | 9.000 | 4.070 | 1.904 | 0.091 | 0.917 | 9.500 | 3.919 | 1.833 | 0.088 | 0.923 | 9.250 |
| VGGT† | 4.158 | 2.410 | 0.147 | 0.809 | 12.250 | 4.284 | 2.400 | 0.147 | 0.811 | 12.250 | 4.129 | 2.421 | 0.148 | 0.803 | 13.000 |
| MoGe V1† | 2.828 | 1.587 | 0.105 | 0.918 | 8.000 | 2.796 | 1.508 | 0.098 | 0.926 | 6.250 | 2.932 | 1.688 | 0.112 | 0.914 | 8.500 |
| MoGe V2 | 4.617 | 3.366 | 0.213 | 0.458 | 15.000 | 4.617 | 3.366 | 0.213 | 0.458 | 15.000 | 4.617 | 3.366 | 0.213 | 0.458 | 15.250 |
| G2-MonoDepth‡ | 2.658 | 0.877 | 0.046 | 0.972 | 4.500 | 3.847 | 1.572 | 0.077 | 0.930 | 7.500 | 1.849 | 0.490 | 0.028 | 0.985 | 3.750 |
| OMNI-DC | 2.116 | 0.629 | 0.031 | 0.981 | 2.000 | 3.305 | 1.114 | 0.050 | 0.958 | 3.000 | 1.759 | 0.443 | 0.025 | 0.985 | 2.250 |
| PriorDA | 2.404 | 0.936 | 0.049 | 0.973 | 4.500 | 3.300 | 1.320 | 0.063 | 0.954 | 4.250 | 1.980 | 0.640 | 0.032 | 0.982 | 5.000 |
| SPNet‡ | 2.379 | 0.716 | 0.037 | 0.978 | 3.000 | 3.516 | 1.219 | 0.055 | 0.958 | 4.000 | 1.776 | 0.459 | 0.026 | 0.986 | 2.750 |
| PromptDA | 2.908 | 1.098 | 0.054 | 0.960 | 6.250 | 3.686 | 1.526 | 0.078 | 0.939 | 6.750 | 2.591 | 0.964 | 0.048 | 0.966 | 6.000 |
| WorldMirror† | 5.776 | 2.941 | 0.148 | 0.794 | 13.750 | 6.412 | 3.235 | 0.159 | 0.765 | 13.750 | 3.848 | 2.192 | 0.133 | 0.859 | 11.500 |
| MapAnything | 13.057 | 6.717 | 0.339 | 0.598 | 15.750 | 13.166 | 6.883 | 0.351 | 0.579 | 15.750 | 12.926 | 6.684 | 0.344 | 0.593 | 15.750 |
| Pow3R† | 3.448 | 1.977 | 0.128 | 0.854 | 10.500 | 3.648 | 1.961 | 0.123 | 0.864 | 10.000 | 3.491 | 2.043 | 0.132 | 0.855 | 11.000 |
| LDCM (Ours) | 1.878 | 0.494 | 0.023 | 0.987 | 1.000 | 2.592 | 0.767 | 0.031 | 0.978 | 1.000 | 1.565 | 0.373 | 0.019 | 0.990 | 1.000 |

| method | 5% | | | | | 3% | | | | | 1% | | | | |
|---|---|---|---|---|---|---|---|---|---|---|---|---|---|---|---|
| | RMSE↓ | MAE↓ | REL↓ | δ₁↑ | Rk.↓ | RMSE↓ | MAE↓ | REL↓ | δ₁↑ | Rk.↓ | RMSE↓ | MAE↓ | REL↓ | δ₁↑ | Rk.↓ |
| DepthPro | 4.149 | 2.763 | 0.178 | 0.731 | 14.000 | 4.149 | 2.763 | 0.178 | 0.731 | 14.000 | 4.149 | 2.763 | 0.178 | 0.731 | 13.750 |
| UniDepth V1 | 3.335 | 2.010 | 0.118 | 0.938 | 9.250 | 3.335 | 2.010 | 0.118 | 0.938 | 9.250 | 3.335 | 2.010 | 0.118 | 0.938 | 9.250 |
| UniDepth V2 | 3.150 | 1.598 | 0.090 | 0.960 | 7.750 | 3.150 | 1.598 | 0.090 | 0.960 | 7.500 | 3.150 | 1.598 | 0.090 | 0.960 | 7.000 |
| DepthAnythingV2† | 3.924 | 1.835 | 0.088 | 0.922 | 9.250 | 3.940 | 1.843 | 0.089 | 0.922 | 9.500 | 3.996 | 1.866 | 0.090 | 0.920 | 9.500 |
| VGGT† | 4.136 | 2.428 | 0.149 | 0.802 | 13.000 | 4.146 | 2.435 | 0.150 | 0.801 | 13.000 | 4.185 | 2.458 | 0.151 | 0.797 | 13.250 |
| MoGe V1† | 2.803 | 1.569 | 0.104 | 0.922 | 8.000 | 2.748 | 1.517 | 0.100 | 0.925 | 7.750 | 2.757 | 1.512 | 0.099 | 0.922 | 7.500 |
| MoGe V2 | 4.617 | 3.366 | 0.213 | 0.458 | 15.250 | 4.617 | 3.366 | 0.213 | 0.458 | 15.250 | 4.617 | 3.366 | 0.213 | 0.458 | 15.250 |
| G2-MonoDepth‡ | 2.035 | 0.573 | 0.031 | 0.983 | 3.750 | 2.244 | 0.672 | 0.035 | 0.980 | 4.000 | 2.930 | 1.040 | 0.051 | 0.966 | 5.250 |
| OMNI-DC | 1.951 | 0.516 | 0.028 | 0.983 | 3.000 | 2.124 | 0.589 | 0.031 | 0.980 | 3.000 | 2.677 | 0.840 | 0.042 | 0.969 | 3.500 |
| PriorDA | 2.099 | 0.690 | 0.034 | 0.980 | 5.000 | 2.210 | 0.738 | 0.036 | 0.978 | 4.750 | 2.524 | 0.880 | 0.042 | 0.972 | 4.500 |
| SPNet‡ | 1.897 | 0.513 | 0.027 | 0.986 | 2.000 | 2.053 | 0.577 | 0.029 | 0.984 | 2.000 | 2.534 | 0.773 | 0.037 | 0.977 | 2.250 |
| PromptDA | 2.762 | 1.070 | 0.055 | 0.961 | 6.000 | 2.874 | 1.117 | 0.056 | 0.959 | 6.500 | 3.245 | 1.341 | 0.068 | 0.946 | 6.750 |
| WorldMirror† | 3.820 | 2.123 | 0.125 | 0.868 | 11.250 | 3.835 | 2.128 | 0.125 | 0.867 | 11.250 | 3.903 | 2.160 | 0.127 | 0.860 | 11.250 |
| MapAnything | 12.980 | 6.726 | 0.345 | 0.594 | 15.750 | 13.106 | 6.986 | 0.368 | 0.573 | 15.750 | 13.334 | 7.326 | 0.388 | 0.550 | 15.750 |
| Pow3R† | 3.472 | 2.020 | 0.131 | 0.850 | 11.250 | 3.472 | 2.010 | 0.130 | 0.853 | 11.000 | 3.544 | 2.047 | 0.132 | 0.850 | 11.250 |
| LDCM (Ours) | 1.691 | 0.420 | 0.021 | 0.989 | 1.000 | 1.821 | 0.468 | 0.022 | 0.988 | 1.000 | 2.160 | 0.610 | 0.027 | 0.983 | 1.000 |

| method | SIFT | | | | | ORB | | | | | Average | | | | |
|---|---|---|---|---|---|---|---|---|---|---|---|---|---|---|---|
| | RMSE↓ | MAE↓ | REL↓ | δ₁↑ | Rk.↓ | RMSE↓ | MAE↓ | REL↓ | δ₁↑ | Rk.↓ | RMSE↓ | MAE↓ | REL↓ | δ₁↑ | Rk.↓ |
| DepthPro | 4.149 | 2.763 | 0.178 | 0.731 | 12.000 | 4.149 | 2.763 | 0.178 | 0.731 | 11.750 | 4.149 | 2.763 | 0.178 | 0.731 | 13.432 |
| UniDepth V1 | 3.335 | 2.010 | 0.118 | 0.938 | 5.750 | 3.335 | 2.010 | 0.118 | 0.938 | 6.250 | 3.335 | 2.010 | 0.118 | 0.938 | 8.636 |
| UniDepth V2 | 3.150 | 1.598 | 0.090 | 0.960 | 3.250 | 3.150 | 1.598 | 0.090 | 0.960 | 3.750 | 3.150 | 1.598 | 0.090 | 0.960 | 6.500 |
| DepthAnythingV2† | 4.295 | 2.080 | 0.102 | 0.896 | 8.000 | 4.299 | 2.116 | 0.108 | 0.886 | 9.000 | 4.007 | 1.890 | 0.092 | 0.916 | 9.091 |
| VGGT† | 4.516 | 2.949 | 0.199 | 0.691 | 14.000 | 4.473 | 2.933 | 0.204 | 0.682 | 13.750 | 4.219 | 2.518 | 0.158 | 0.783 | 12.909 |
| MoGe V1† | 3.274 | 2.194 | 0.170 | 0.792 | 8.500 | 3.315 | 2.289 | 0.184 | 0.760 | 9.500 | 3.050 | 1.821 | 0.125 | 0.887 | 8.568 |
| MoGe V2 | 4.617 | 3.366 | 0.213 | 0.458 | 15.250 | 4.617 | 3.366 | 0.213 | 0.458 | 15.250 | 4.617 | 3.366 | 0.213 | 0.458 | 15.182 |
| G2-MonoDepth‡ | 4.238 | 2.212 | 0.133 | 0.800 | 10.000 | 3.617 | 1.680 | 0.101 | 0.869 | 7.250 | 2.638 | 0.964 | 0.054 | 0.949 | 5.295 |
| OMNI-DC | 3.630 | 1.632 | 0.099 | 0.884 | 6.250 | 3.443 | 1.514 | 0.097 | 0.889 | 5.500 | 2.302 | 0.760 | 0.042 | 0.963 | 3.045 |
| PriorDA | 2.904 | 1.174 | 0.061 | 0.948 | 2.250 | 2.769 | 1.118 | 0.061 | 0.949 | 2.250 | 2.364 | 0.861 | 0.044 | 0.971 | 4.159 |
| SPNet‡ | 3.358 | 1.482 | 0.085 | 0.891 | 4.500 | 3.107 | 1.265 | 0.075 | 0.911 | 3.500 | 2.365 | 0.757 | 0.041 | 0.966 | 3.000 |
| PromptDA | 3.882 | 1.990 | 0.120 | 0.887 | 7.250 | 3.909 | 2.027 | 0.126 | 0.872 | 8.500 | 3.040 | 1.261 | 0.067 | 0.946 | 6.545 |
| WorldMirror† | 4.166 | 2.558 | 0.165 | 0.769 | 10.750 | 4.239 | 2.551 | 0.166 | 0.770 | 10.750 | 4.439 | 2.432 | 0.142 | 0.824 | 11.818 |
| MapAnything | 13.101 | 7.111 | 0.374 | 0.561 | 15.750 | 13.133 | 7.046 | 0.372 | 0.577 | 15.750 | 12.974 | 6.784 | 0.350 | 0.588 | 15.750 |
| Pow3R† | 3.981 | 2.646 | 0.197 | 0.735 | 11.500 | 3.973 | 2.694 | 0.206 | 0.719 | 12.250 | 3.515 | 2.096 | 0.141 | 0.832 | 10.750 |
| LDCM (Ours) | 2.579 | 0.910 | 0.043 | 0.963 | 1.000 | 2.478 | 0.846 | 0.042 | 0.962 | 1.000 | 1.911 | 0.537 | 0.026 | 0.983 | 1.068 |

Table 13: **Quantitative comparison of depth completion with baseline methods on the indoor scenes of the DIODE dataset** Vasiljevic et al. (2019)**.** Methods marked with † produce relative depth maps, where the metric depth is recovered by optimizing global scale and shift via least squares regression using the sparse depth prior. Methods marked with ‡ use scenario-specific configurations for indoor and outdoor scenes, respectively. The best and the second best results are highlighted.

| Method | 10% Noise RMSE↓ | MAE↓ | REL↓ | $\delta_1$↑ | Rk.↓ | 5% RMSE↓ | MAE↓ | REL↓ | $\delta_1$↑ | Rk.↓ | 3% RMSE↓ | MAE↓ | REL↓ | $\delta_1$↑ | Rk.↓ |
|---|---|---|---|---|---|---|---|---|---|---|---|---|---|---|---|
| DepthPro | 0.837 | 0.702 | 0.193 | 0.668 | 14.500 | 0.837 | 0.702 | 0.193 | 0.668 | 14.500 | 0.837 | 0.702 | 0.193 | 0.668 | 14.500 |
| UniDepth V1 | 0.939 | 0.840 | 0.158 | 0.779 | 14.000 | 0.939 | 0.840 | 0.158 | 0.779 | 14.000 | 0.939 | 0.840 | 0.158 | 0.779 | 14.000 |
| UniDepth V2 | 0.811 | 0.678 | 0.165 | 0.681 | 13.500 | 0.811 | 0.678 | 0.165 | 0.681 | 13.500 | 0.811 | 0.678 | 0.165 | 0.681 | 13.500 |
| DepthAnythingV2† | 0.383 | 0.185 | 0.041 | 0.979 | 8.500 | 0.387 | 0.181 | 0.040 | 0.979 | 8.750 | 0.388 | 0.181 | 0.040 | 0.979 | 8.750 |
| VGGT† | 0.392 | 0.262 | 0.078 | 0.928 | 11.750 | 0.391 | 0.261 | 0.078 | 0.929 | 11.750 | 0.391 | 0.261 | 0.078 | 0.929 | 11.750 |
| MoGe V1† | 0.243 | 0.144 | 0.045 | 0.956 | 8.750 | 0.239 | 0.140 | 0.043 | 0.958 | 8.250 | 0.239 | 0.140 | 0.043 | 0.958 | 8.000 |
| MoGe V2 | 1.064 | 0.938 | 0.235 | 0.433 | 16.000 | 1.064 | 0.938 | 0.235 | 0.433 | 16.000 | 1.064 | 0.938 | 0.235 | 0.433 | 16.000 |
| G2-MonoDepth‡ | 0.020 | 0.004 | 0.001 | 1.000 | 1.500 | 0.021 | 0.005 | 0.001 | 1.000 | 2.250 | 0.026 | 0.005 | 0.001 | 1.000 | 2.250 |
| OMNI-DC | 0.066 | 0.009 | 0.002 | 0.999 | 4.000 | 0.022 | 0.002 | 0.000 | 1.000 | 1.500 | 0.026 | 0.002 | 0.000 | 1.000 | 1.250 |
| PriorDA | 0.077 | 0.029 | 0.006 | 0.999 | 4.750 | 0.050 | 0.016 | 0.004 | 1.000 | 4.000 | 0.051 | 0.016 | 0.004 | 0.999 | 5.000 |
| SPNet‡ | 0.019 | 0.003 | 0.001 | 1.000 | 1.000 | 0.018 | 0.003 | 0.001 | 1.000 | 1.500 | 0.021 | 0.003 | 0.001 | 1.000 | 1.500 |
| PromptDA | 0.099 | 0.049 | 0.014 | 0.998 | 6.000 | 0.092 | 0.047 | 0.013 | 0.997 | 6.000 | 0.095 | 0.047 | 0.013 | 0.998 | 6.000 |
| WorldMirror† | 0.323 | 0.195 | 0.062 | 0.958 | 10.250 | 0.380 | 0.231 | 0.069 | 0.946 | 10.500 | 0.376 | 0.220 | 0.066 | 0.948 | 10.500 |
| MapAnything | 0.535 | 0.129 | 0.030 | 0.977 | 8.500 | 0.508 | 0.133 | 0.027 | 0.984 | 8.250 | 0.552 | 0.156 | 0.031 | 0.982 | 8.500 |
| Pow3R† | 0.260 | 0.186 | 0.056 | 0.976 | 9.250 | 0.296 | 0.207 | 0.063 | 0.954 | 9.500 | 0.310 | 0.211 | 0.063 | 0.951 | 9.500 |
| LDCM (Ours) | 0.023 | 0.005 | 0.001 | 1.000 | 2.000 | 0.025 | 0.004 | 0.001 | 1.000 | 2.500 | 0.028 | 0.004 | 0.001 | 1.000 | 2.500 |

| Method | 1% RMSE↓ | MAE↓ | REL↓ | $\delta_1$↑ | Rk.↓ | 500 RMSE↓ | MAE↓ | REL↓ | $\delta_1$↑ | Rk.↓ | 100 RMSE↓ | MAE↓ | REL↓ | $\delta_1$↑ | Rk.↓ |
|---|---|---|---|---|---|---|---|---|---|---|---|---|---|---|---|
| DepthPro | 0.837 | 0.702 | 0.193 | 0.668 | 14.500 | 0.837 | 0.702 | 0.193 | 0.668 | 14.250 | 0.837 | 0.702 | 0.193 | 0.668 | 13.750 |
| UniDepth V1 | 0.939 | 0.840 | 0.158 | 0.779 | 14.000 | 0.939 | 0.840 | 0.158 | 0.779 | 13.750 | 0.939 | 0.840 | 0.158 | 0.779 | 13.250 |
| UniDepth V2 | 0.811 | 0.678 | 0.165 | 0.681 | 13.500 | 0.811 | 0.678 | 0.165 | 0.681 | 13.250 | 0.811 | 0.678 | 0.165 | 0.681 | 12.500 |
| DepthAnythingV2† | 0.391 | 0.182 | 0.040 | 0.979 | 8.000 | 0.410 | 0.189 | 0.041 | 0.978 | 8.500 | 0.428 | 0.193 | 0.041 | 0.978 | 7.500 |
| VGGT† | 0.391 | 0.261 | 0.078 | 0.928 | 11.500 | 0.392 | 0.262 | 0.078 | 0.928 | 10.750 | 0.399 | 0.261 | 0.078 | 0.931 | 9.750 |
| MoGe V1† | 0.239 | 0.140 | 0.043 | 0.958 | 7.750 | 0.240 | 0.141 | 0.044 | 0.956 | 7.750 | 0.244 | 0.141 | 0.043 | 0.959 | 6.250 |
| MoGe V2 | 1.064 | 0.938 | 0.235 | 0.433 | 16.000 | 1.064 | 0.938 | 0.235 | 0.433 | 15.750 | 1.064 | 0.938 | 0.235 | 0.433 | 15.750 |
| G2-MonoDepth‡ | 0.043 | 0.009 | 0.002 | 1.000 | 3.250 | 0.136 | 0.045 | 0.011 | 0.997 | 5.000 | 0.732 | 0.550 | 0.179 | 0.622 | 12.750 |
| OMNI-DC | 0.036 | 0.003 | 0.001 | 1.000 | 1.250 | 0.088 | 0.016 | 0.004 | 0.998 | 2.500 | 0.185 | 0.059 | 0.014 | 0.992 | 3.000 |
| PriorDA | 0.057 | 0.016 | 0.004 | 0.999 | 5.000 | 0.081 | 0.022 | 0.005 | 0.999 | 2.750 | 0.124 | 0.039 | 0.009 | 0.996 | 1.500 |
| SPNet‡ | 0.034 | 0.005 | 0.001 | 1.000 | 1.250 | 0.101 | 0.020 | 0.004 | 0.998 | 3.000 | 0.200 | 0.067 | 0.016 | 0.992 | 3.750 |
| PromptDA | 0.105 | 0.051 | 0.014 | 0.997 | 6.000 | 0.166 | 0.077 | 0.022 | 0.991 | 6.000 | 0.246 | 0.115 | 0.031 | 0.982 | 5.250 |
| WorldMirror† | 0.343 | 0.193 | 0.057 | 0.954 | 9.750 | 0.326 | 0.183 | 0.054 | 0.955 | 8.750 | 0.330 | 0.180 | 0.052 | 0.957 | 7.500 |
| MapAnything | 0.635 | 0.224 | 0.045 | 0.974 | 10.000 | 1.149 | 0.609 | 0.135 | 0.879 | 13.000 | 1.248 | 0.771 | 0.153 | 0.791 | 13.000 |
| Pow3R† | 0.321 | 0.207 | 0.061 | 0.956 | 9.750 | 0.337 | 0.210 | 0.060 | 0.958 | 9.250 | 0.349 | 0.215 | 0.062 | 0.953 | 8.750 |
| LDCM (Ours) | 0.038 | 0.005 | 0.001 | 1.000 | 1.750 | 0.079 | 0.012 | 0.002 | 0.999 | 1.000 | 0.154 | 0.034 | 0.006 | 0.996 | 1.250 |

| Method | SIFT RMSE↓ | MAE↓ | REL↓ | $\delta_1$↑ | Rk.↓ | ORB RMSE↓ | MAE↓ | REL↓ | $\delta_1$↑ | Rk.↓ | Virtual-Lidar-32-Lines RMSE↓ | MAE↓ | REL↓ | $\delta_1$↑ | Rk.↓ |
|---|---|---|---|---|---|---|---|---|---|---|---|---|---|---|---|
| DepthPro | 0.837 | 0.702 | 0.193 | 0.668 | 13.500 | 0.837 | 0.702 | 0.193 | 0.668 | 13.000 | 0.837 | 0.702 | 0.193 | 0.668 | 14.250 |
| UniDepth V1 | 0.939 | 0.840 | 0.158 | 0.779 | 12.250 | 0.939 | 0.840 | 0.158 | 0.779 | 12.250 | 0.939 | 0.840 | 0.158 | 0.779 | 13.750 |
| UniDepth V2 | 0.811 | 0.678 | 0.165 | 0.681 | 12.000 | 0.811 | 0.678 | 0.165 | 0.681 | 11.750 | 0.811 | 0.678 | 0.165 | 0.681 | 13.250 |
| DepthAnythingV2† | 0.365 | 0.221 | 0.072 | 0.961 | 3.250 | 0.343 | 0.205 | 0.060 | 0.961 | 3.000 | 0.388 | 0.181 | 0.040 | 0.979 | 8.000 |
| VGGT† | 0.571 | 0.447 | 0.183 | 0.881 | 9.750 | 0.577 | 0.438 | 0.174 | 0.877 | 8.500 | 0.391 | 0.261 | 0.078 | 0.928 | 11.000 |
| MoGe V1† | 0.411 | 0.329 | 0.161 | 0.919 | 6.750 | 0.413 | 0.328 | 0.155 | 0.913 | 5.250 | 0.240 | 0.140 | 0.043 | 0.957 | 5.250 |
| MoGe V2 | 1.064 | 0.938 | 0.235 | 0.433 | 15.500 | 1.064 | 0.938 | 0.235 | 0.433 | 15.750 | 1.064 | 0.938 | 0.235 | 0.433 | 16.000 |
| G2-MonoDepth‡ | 0.894 | 0.670 | 0.241 | 0.554 | 13.750 | 0.996 | 0.755 | 0.261 | 0.513 | 14.750 | 0.081 | 0.020 | 0.005 | 0.999 | 4.000 |
| OMNI-DC | 0.353 | 0.223 | 0.083 | 0.870 | 5.000 | 0.492 | 0.330 | 0.119 | 0.800 | 5.750 | 0.056 | 0.007 | 0.002 | 0.999 | 1.500 |
| PriorDA | 0.145 | 0.081 | 0.032 | 0.975 | 1.500 | 0.197 | 0.115 | 0.049 | 0.967 | 1.500 | 0.063 | 0.018 | 0.004 | 0.999 | 3.000 |
| SPNet‡ | 0.473 | 0.318 | 0.125 | 0.768 | 7.750 | 0.552 | 0.375 | 0.140 | 0.746 | 7.500 | 0.071 | 0.011 | 0.002 | 0.999 | 2.500 |
| PromptDA | 0.365 | 0.252 | 0.095 | 0.877 | 5.500 | 0.608 | 0.419 | 0.149 | 0.795 | 8.500 | 0.124 | 0.058 | 0.018 | 0.995 | 6.000 |
| WorldMirror† | 0.532 | 0.411 | 0.187 | 0.897 | 9.000 | 0.648 | 0.502 | 0.214 | 0.874 | 9.000 | 0.330 | 0.185 | 0.054 | 0.954 | 9.000 |
| MapAnything | 1.105 | 0.670 | 0.157 | 0.812 | 11.250 | 1.032 | 0.578 | 0.140 | 0.854 | 9.750 | 1.057 | 0.560 | 0.147 | 0.879 | 12.750 |
| Pow3R† | 0.499 | 0.388 | 0.156 | 0.884 | 7.250 | 0.499 | 0.386 | 0.154 | 0.872 | 7.000 | 0.331 | 0.208 | 0.060 | 0.956 | 9.500 |
| LDCM (Ours) | 0.149 | 0.072 | 0.027 | 0.973 | 1.500 | 0.208 | 0.104 | 0.037 | 0.963 | 1.500 | 0.053 | 0.007 | 0.001 | 0.999 | 1.000 |

| Method | Virtual-Lidar-16-Lines RMSE↓ | MAE↓ | REL↓ | $\delta_1$↑ | Rk.↓ | Virtual-Lidar-8-Lines RMSE↓ | MAE↓ | REL↓ | $\delta_1$↑ | Rk.↓ | Average RMSE↓ | MAE↓ | REL↓ | $\delta_1$↑ | Rk.↓ |
|---|---|---|---|---|---|---|---|---|---|---|---|---|---|---|---|
| DepthPro | 0.837 | 0.702 | 0.193 | 0.668 | 14.250 | 0.837 | 0.702 | 0.193 | 0.668 | 14.250 | 0.837 | 0.702 | 0.193 | 0.668 | 14.114 |
| UniDepth V1 | 0.939 | 0.840 | 0.158 | 0.779 | 13.750 | 0.939 | 0.840 | 0.158 | 0.779 | 13.750 | 0.939 | 0.840 | 0.158 | 0.779 | 13.523 |
| UniDepth V2 | 0.811 | 0.678 | 0.165 | 0.681 | 13.250 | 0.811 | 0.678 | 0.165 | 0.681 | 13.250 | 0.811 | 0.678 | 0.165 | 0.681 | 13.023 |
| DepthAnythingV2† | 0.382 | 0.180 | 0.040 | 0.979 | 8.000 | 0.382 | 0.179 | 0.039 | 0.979 | 8.000 | 0.386 | 0.189 | 0.045 | 0.976 | 7.295 |
| VGGT† | 0.392 | 0.261 | 0.078 | 0.928 | 11.000 | 0.393 | 0.264 | 0.079 | 0.928 | 11.000 | 0.425 | 0.294 | 0.096 | 0.920 | 10.773 |
| MoGe V1† | 0.240 | 0.140 | 0.043 | 0.957 | 7.500 | 0.241 | 0.142 | 0.044 | 0.954 | 7.500 | 0.272 | 0.175 | 0.064 | 0.950 | 7.386 |
| MoGe V2 | 1.064 | 0.938 | 0.235 | 0.433 | 15.750 | 1.064 | 0.938 | 0.235 | 0.433 | 15.750 | 1.064 | 0.938 | 0.235 | 0.433 | 15.841 |
| G2-MonoDepth‡ | 0.121 | 0.033 | 0.008 | 0.997 | 5.000 | 0.211 | 0.086 | 0.023 | 0.988 | 5.250 | 0.298 | 0.198 | 0.067 | 0.879 | 6.341 |
| OMNI-DC | 0.080 | 0.014 | 0.003 | 0.998 | 2.500 | 0.144 | 0.042 | 0.010 | 0.995 | 4.000 | 0.141 | 0.064 | 0.022 | 0.968 | 2.932 |
| PriorDA | 0.078 | 0.022 | 0.005 | 0.999 | 2.750 | 0.104 | 0.034 | 0.008 | 0.998 | 1.500 | 0.093 | 0.037 | 0.012 | 0.994 | 3.023 |
| SPNet‡ | 0.094 | 0.017 | 0.004 | 0.998 | 3.250 | 0.143 | 0.038 | 0.009 | 0.996 | 3.000 | 0.157 | 0.078 | 0.028 | 0.954 | 3.273 |
| PromptDA | 0.146 | 0.066 | 0.018 | 0.994 | 6.000 | 0.190 | 0.087 | 0.024 | 0.988 | 5.500 | 0.203 | 0.115 | 0.037 | 0.965 | 6.068 |
| WorldMirror† | 0.327 | 0.182 | 0.053 | 0.956 | 8.750 | 0.331 | 0.188 | 0.056 | 0.951 | 9.000 | 0.386 | 0.243 | 0.084 | 0.941 | 9.364 |
| MapAnything | 1.077 | 0.606 | 0.149 | 0.870 | 13.000 | 1.099 | 0.600 | 0.126 | 0.890 | 13.000 | 0.909 | 0.458 | 0.104 | 0.899 | 11.000 |
| Pow3R† | 0.337 | 0.211 | 0.061 | 0.955 | 9.750 | 0.341 | 0.216 | 0.063 | 0.953 | 9.500 | 0.353 | 0.240 | 0.078 | 0.943 | 9.000 |
| LDCM (Ours) | 0.067 | 0.011 | 0.002 | 0.999 | 1.000 | 0.104 | 0.022 | 0.005 | 0.998 | 1.000 | 0.084 | 0.025 | 0.008 | 0.993 | 1.545 |

Table 14: **Quantitative comparison of depth completion with baseline methods on the outdoor scenes of the DIODE dataset** Vasiljevic et al. (2019). Methods marked with † produce relative depth maps, where the metric depth is recovered by optimizing global scale and shift via least squares regression using the sparse depth prior. Methods marked with ‡ use scenario-specific configurations for indoor and outdoor scenes, respectively. The best and the second best results are highlighted.

| Method | Virtual-Lidar-64-Lines | | | | | Virtual-Lidar-32-Lines | | | | | Virtual-Lidar-16-Lines | | | | |
|---|---|---|---|---|---|---|---|---|---|---|---|---|---|---|---|
| | RMSE↓ | MAE↓ | REL↓ | $\delta_1$↑ | Rk.↓ | RMSE↓ | MAE↓ | REL↓ | $\delta_1$↑ | Rk.↓ | RMSE↓ | MAE↓ | REL↓ | $\delta_1$↑ | Rk.↓ |
| DepthPro | 9.539 | 7.635 | 0.403 | 0.177 | 14.750 | 9.539 | 7.635 | 0.403 | 0.177 | 15.250 | 9.539 | 7.635 | 0.403 | 0.177 | 15.000 |
| UniDepth V1 | 5.782 | 3.841 | 0.189 | 0.661 | 12.000 | 5.782 | 3.841 | 0.189 | 0.661 | 11.750 | 5.782 | 3.841 | 0.189 | 0.661 | 11.750 |
| UniDepth V2 | 11.145 | 8.936 | 0.515 | 0.526 | 15.750 | 11.145 | 8.936 | 0.515 | 0.526 | 15.750 | 11.145 | 8.936 | 0.515 | 0.526 | 15.750 |
| DepthAnythingV2† | 5.786 | 2.626 | 0.118 | 0.882 | 8.750 | 5.829 | 2.649 | 0.118 | 0.880 | 8.750 | 5.881 | 2.670 | 0.121 | 0.879 | 8.750 |
| VGGT† | 4.739 | 2.748 | 0.227 | 0.779 | 10.750 | 4.743 | 2.761 | 0.230 | 0.779 | 10.250 | 4.765 | 2.758 | 0.228 | 0.780 | 10.250 |
| MoGe V1† | 10.329 | 8.351 | 0.396 | 0.603 | 14.500 | 9.455 | 7.366 | 0.354 | 0.648 | 14.000 | 9.034 | 6.809 | 0.317 | 0.685 | 12.750 |
| MoGe V2 | 4.807 | 3.352 | 0.182 | 0.680 | 10.750 | 4.807 | 3.352 | 0.182 | 0.680 | 10.500 | 4.807 | 3.352 | 0.182 | 0.680 | 10.750 |
| G2-MonoDepth‡ | 1.938 | 0.489 | 0.039 | 0.975 | 4.000 | 2.275 | 0.629 | 0.052 | 0.967 | 4.500 | 2.719 | 0.882 | 0.069 | 0.950 | 5.000 |
| OMNI-DC | 1.899 | 0.424 | 0.033 | 0.977 | 3.000 | 2.196 | 0.532 | 0.042 | 0.970 | 3.250 | 2.659 | 0.738 | 0.056 | 0.960 | 3.750 |
| PriorDA | 1.970 | 0.674 | 0.041 | 0.969 | 5.000 | 2.097 | 0.716 | 0.044 | 0.966 | 4.000 | 2.278 | 0.796 | 0.050 | 0.961 | 2.500 |
| SPNet‡ | 1.809 | 0.419 | 0.032 | 0.978 | 1.750 | 2.100 | 0.518 | 0.039 | 0.972 | 2.250 | 2.536 | 0.715 | 0.054 | 0.962 | 2.500 |
| PromptDA | 3.142 | 1.239 | 0.071 | 0.939 | 6.000 | 3.316 | 1.336 | 0.077 | 0.931 | 6.000 | 3.579 | 1.470 | 0.082 | 0.925 | 6.000 |
| WorldMirror† | 4.103 | 2.166 | 0.147 | 0.836 | 7.750 | 4.224 | 2.248 | 0.151 | 0.832 | 8.250 | 4.214 | 2.214 | 0.147 | 0.835 | 8.000 |
| MapAnything | 7.393 | 3.444 | 0.203 | 0.822 | 11.750 | 8.873 | 4.927 | 0.299 | 0.673 | 12.750 | 9.705 | 5.717 | 0.318 | 0.560 | 14.000 |
| Pow3R† | 3.859 | 2.169 | 0.179 | 0.834 | 8.250 | 3.910 | 2.193 | 0.179 | 0.835 | 7.750 | 3.957 | 2.193 | 0.178 | 0.835 | 7.750 |
| LDCM (Ours) | 1.795 | 0.404 | 0.024 | 0.978 | 1.000 | 2.010 | 0.476 | 0.029 | 0.974 | 1.000 | 2.280 | 0.603 | 0.036 | 0.967 | 1.250 |

| Method | Virtual-Lidar-8-Lines | | | | | Virtual-Lidar-4-Lines | | | | | 10% Noise | | | | |
|---|---|---|---|---|---|---|---|---|---|---|---|---|---|---|---|
| | RMSE↓ | MAE↓ | REL↓ | $\delta_1$↑ | Rk.↓ | RMSE↓ | MAE↓ | REL↓ | $\delta_1$↑ | Rk.↓ | RMSE↓ | MAE↓ | REL↓ | $\delta_1$↑ | Rk.↓ |
| DepthPro | 9.539 | 7.635 | 0.403 | 0.177 | 15.000 | 9.539 | 7.635 | 0.403 | 0.177 | 14.500 | 9.539 | 7.635 | 0.403 | 0.177 | 14.500 |
| UniDepth V1 | 5.782 | 3.841 | 0.189 | 0.661 | 11.750 | 5.782 | 3.841 | 0.189 | 0.661 | 11.000 | 5.782 | 3.841 | 0.189 | 0.661 | 12.500 |
| UniDepth V2 | 11.145 | 8.936 | 0.515 | 0.526 | 15.500 | 11.145 | 8.936 | 0.515 | 0.526 | 15.500 | 11.145 | 8.936 | 0.515 | 0.526 | 15.000 |
| DepthAnythingV2† | 5.911 | 2.746 | 0.123 | 0.875 | 8.750 | 6.421 | 2.974 | 0.138 | 0.855 | 9.250 | 5.778 | 2.655 | 0.119 | 0.881 | 9.500 |
| VGGT† | 4.836 | 2.792 | 0.227 | 0.780 | 10.500 | 5.463 | 3.182 | 0.263 | 0.773 | 10.000 | 4.741 | 2.753 | 0.225 | 0.789 | 11.000 |
| MoGe V1† | 7.903 | 5.351 | 0.270 | 0.744 | 12.500 | 7.833 | 4.104 | 0.241 | 0.798 | 11.500 | 12.694 | 10.609 | 0.512 | 0.488 | 15.500 |
| MoGe V2 | 4.807 | 3.352 | 0.182 | 0.680 | 10.500 | 4.807 | 3.352 | 0.182 | 0.680 | 9.250 | 4.807 | 3.352 | 0.182 | 0.680 | 11.250 |
| G2-MonoDepth‡ | 3.611 | 1.467 | 0.099 | 0.909 | 5.250 | 5.910 | 3.230 | 0.204 | 0.698 | 10.750 | 0.876 | 0.206 | 0.016 | 0.992 | 2.250 |
| OMNI-DC | 3.536 | 1.228 | 0.075 | 0.935 | 3.750 | 5.361 | 2.572 | 0.158 | 0.801 | 6.000 | 1.058 | 0.222 | 0.017 | 0.991 | 3.750 |
| PriorDA | 2.744 | 1.019 | 0.059 | 0.949 | 2.000 | 4.068 | 1.731 | 0.106 | 0.877 | 2.000 | 1.957 | 0.769 | 0.041 | 0.971 | 5.000 |
| SPNet‡ | 3.286 | 1.148 | 0.079 | 0.939 | 3.250 | 4.652 | 2.188 | 0.168 | 0.813 | 5.000 | 0.850 | 0.192 | 0.014 | 0.993 | 1.000 |
| PromptDA | 4.106 | 1.780 | 0.096 | 0.908 | 6.000 | 5.690 | 3.010 | 0.167 | 0.788 | 8.000 | 2.933 | 1.207 | 0.068 | 0.939 | 6.000 |
| WorldMirror† | 4.360 | 2.294 | 0.150 | 0.830 | 8.250 | 4.724 | 2.444 | 0.151 | 0.829 | 4.250 | 3.049 | 1.525 | 0.112 | 0.891 | 7.250 |
| MapAnything | 10.595 | 6.288 | 0.308 | 0.520 | 14.500 | 10.993 | 6.991 | 0.346 | 0.343 | 14.500 | 5.845 | 2.077 | 0.122 | 0.884 | 9.750 |
| Pow3R† | 4.045 | 2.239 | 0.174 | 0.835 | 7.500 | 4.616 | 2.587 | 0.197 | 0.821 | 6.000 | 2.976 | 1.723 | 0.144 | 0.861 | 8.750 |
| LDCM (Ours) | 2.679 | 0.823 | 0.045 | 0.954 | 1.000 | 3.418 | 1.300 | 0.072 | 0.913 | 1.000 | 1.052 | 0.222 | 0.014 | 0.992 | 2.250 |

| Method | 5% | | | | | 3% | | | | | 1% | | | | |
|---|---|---|---|---|---|---|---|---|---|---|---|---|---|---|---|
| | RMSE↓ | MAE↓ | REL↓ | $\delta_1$↑ | Rk.↓ | RMSE↓ | MAE↓ | REL↓ | $\delta_1$↑ | Rk.↓ | RMSE↓ | MAE↓ | REL↓ | $\delta_1$↑ | Rk.↓ |
| DepthPro | 9.539 | 7.635 | 0.403 | 0.177 | 14.500 | 9.539 | 7.635 | 0.403 | 0.177 | 14.500 | 9.539 | 7.635 | 0.403 | 0.177 | 13.500 |
| UniDepth V1 | 5.782 | 3.841 | 0.189 | 0.661 | 12.250 | 5.782 | 3.841 | 0.189 | 0.661 | 12.250 | 5.782 | 3.841 | 0.189 | 0.661 | 11.250 |
| UniDepth V2 | 11.145 | 8.936 | 0.515 | 0.526 | 15.000 | 11.145 | 8.936 | 0.515 | 0.526 | 15.000 | 11.145 | 8.936 | 0.515 | 0.526 | 14.250 |
| DepthAnythingV2† | 5.891 | 2.667 | 0.119 | 0.881 | 9.750 | 5.896 | 2.666 | 0.119 | 0.881 | 8.250 | 5.924 | 2.680 | 0.119 | 0.881 | 8.250 |
| VGGT† | 4.732 | 2.756 | 0.228 | 0.781 | 11.000 | 4.732 | 2.759 | 0.229 | 0.781 | 11.000 | 4.734 | 2.758 | 0.229 | 0.781 | 10.000 |
| MoGe V1† | 12.548 | 10.462 | 0.502 | 0.496 | 15.500 | 12.131 | 10.042 | 0.480 | 0.519 | 15.500 | 11.346 | 9.323 | 0.441 | 0.558 | 14.250 |
| MoGe V2 | 4.807 | 3.352 | 0.182 | 0.680 | 11.250 | 4.807 | 3.352 | 0.182 | 0.680 | 11.000 | 4.807 | 3.352 | 0.182 | 0.680 | 10.250 |
| G2-MonoDepth‡ | 1.076 | 0.247 | 0.020 | 0.989 | 2.750 | 1.245 | 0.285 | 0.023 | 0.986 | 3.250 | 1.680 | 0.402 | 0.032 | 0.980 | 4.000 |
| OMNI-DC | 1.110 | 0.224 | 0.018 | 0.989 | 2.250 | 1.270 | 0.260 | 0.020 | 0.987 | 2.250 | 1.649 | 0.356 | 0.027 | 0.981 | 2.250 |
| PriorDA | 1.785 | 0.645 | 0.037 | 0.974 | 5.000 | 1.805 | 0.645 | 0.038 | 0.973 | 5.000 | 1.895 | 0.666 | 0.040 | 0.970 | 5.000 |
| SPNet‡ | 1.029 | 0.227 | 0.017 | 0.990 | 1.500 | 1.187 | 0.258 | 0.020 | 0.988 | 1.250 | 1.566 | 0.349 | 0.027 | 0.982 | 1.250 |
| PromptDA | 2.941 | 1.210 | 0.070 | 0.937 | 6.000 | 2.981 | 1.206 | 0.070 | 0.938 | 6.000 | 3.090 | 1.233 | 0.073 | 0.935 | 6.000 |
| WorldMirror† | 3.510 | 1.836 | 0.127 | 0.860 | 8.250 | 3.844 | 2.047 | 0.141 | 0.840 | 8.250 | 6.659 | 2.731 | 0.159 | 0.860 | 9.250 |
| MapAnything | 5.868 | 2.073 | 0.116 | 0.895 | 8.750 | 6.099 | 2.226 | 0.125 | 0.888 | 9.250 | 3.796 | 2.152 | 0.180 | 0.833 | 8.000 |
| Pow3R† | 3.365 | 1.997 | 0.174 | 0.823 | 8.750 | 3.586 | 2.105 | 0.183 | 0.822 | 9.000 | 1.645 | 0.360 | 0.022 | 0.981 | 2.000 |
| LDCM (Ours) | 1.209 | 0.255 | 0.016 | 0.989 | 2.750 | 1.348 | 0.285 | 0.018 | 0.987 | 2.500 | 1.645 | 0.360 | 0.022 | 0.981 | 2.000 |

| Method | SIFT | | | | | ORB | | | | | Average | | | | |
|---|---|---|---|---|---|---|---|---|---|---|---|---|---|---|---|
| | RMSE↓ | MAE↓ | REL↓ | $\delta_1$↑ | Rk.↓ | RMSE↓ | MAE↓ | REL↓ | $\delta_1$↑ | Rk.↓ | RMSE↓ | MAE↓ | REL↓ | $\delta_1$↑ | Rk.↓ |
| DepthPro | 9.539 | 7.635 | 0.403 | 0.177 | 14.500 | 9.539 | 7.635 | 0.403 | 0.177 | 14.500 | 9.539 | 7.635 | 0.403 | 0.177 | 14.636 |
| UniDepth V1 | 5.782 | 3.841 | 0.189 | 0.661 | 11.750 | 5.782 | 3.841 | 0.189 | 0.661 | 11.500 | 5.782 | 3.841 | 0.189 | 0.661 | 11.795 |
| UniDepth V2 | 11.145 | 8.936 | 0.515 | 0.526 | 15.250 | 11.145 | 8.936 | 0.515 | 0.526 | 15.000 | 11.145 | 8.936 | 0.515 | 0.526 | 15.250 |
| DepthAnythingV2† | 5.948 | 3.043 | 0.133 | 0.836 | 8.750 | 6.079 | 3.170 | 0.136 | 0.827 | 8.750 | 5.940 | 2.777 | 0.124 | 0.869 | 8.659 |
| VGGT† | 5.102 | 3.192 | 0.256 | 0.739 | 11.000 | 5.296 | 3.368 | 0.267 | 0.728 | 10.750 | 4.898 | 2.893 | 0.237 | 0.772 | 10.591 |
| MoGe V1† | 11.377 | 9.519 | 0.472 | 0.535 | 15.250 | 11.684 | 9.807 | 0.485 | 0.518 | 15.500 | 10.576 | 8.340 | 0.406 | 0.599 | 14.250 |
| MoGe V2 | 4.807 | 3.352 | 0.182 | 0.680 | 10.250 | 4.807 | 3.352 | 0.182 | 0.680 | 9.500 | 4.807 | 3.352 | 0.182 | 0.680 | 10.477 |
| G2-MonoDepth‡ | 2.335 | 0.800 | 0.060 | 0.945 | 4.750 | 2.659 | 0.984 | 0.068 | 0.927 | 5.000 | 2.393 | 0.875 | 0.062 | 0.938 | 4.682 |
| OMNI-DC | 2.201 | 0.608 | 0.040 | 0.966 | 3.000 | 2.606 | 0.824 | 0.049 | 0.951 | 3.500 | 2.322 | 0.726 | 0.049 | 0.955 | 3.341 |
| PriorDA | 2.220 | 0.803 | 0.046 | 0.964 | 4.250 | 2.587 | 0.979 | 0.054 | 0.952 | 3.500 | 2.310 | 0.858 | 0.051 | 0.957 | 3.932 |
| SPNet‡ | 1.983 | 0.549 | 0.038 | 0.970 | 1.750 | 2.223 | 0.678 | 0.044 | 0.961 | 1.750 | 2.111 | 0.658 | 0.048 | 0.959 | 2.114 |
| PromptDA | 3.705 | 1.572 | 0.083 | 0.915 | 6.000 | 4.159 | 1.911 | 0.102 | 0.882 | 6.000 | 3.604 | 1.561 | 0.087 | 0.912 | 6.182 |
| WorldMirror† | 4.426 | 2.477 | 0.166 | 0.802 | 7.750 | 5.995 | 3.501 | 0.209 | 0.696 | 11.250 | 4.464 | 2.317 | 0.151 | 0.828 | 8.045 |
| MapAnything | 7.718 | 3.593 | 0.208 | 0.797 | 11.500 | 7.540 | 3.316 | 0.183 | 0.821 | 9.750 | 7.675 | 3.891 | 0.219 | 0.731 | 11.318 |
| Pow3R† | 4.236 | 2.551 | 0.214 | 0.802 | 8.750 | 4.308 | 2.634 | 0.213 | 0.792 | 8.750 | 3.682 | 2.068 | 0.169 | 0.840 | 7.568 |
| LDCM (Ours) | 1.984 | 0.491 | 0.028 | 0.974 | 1.250 | 2.234 | 0.603 | 0.034 | 0.965 | 1.250 | 1.969 | 0.529 | 0.031 | 0.970 | 1.568 |

Table 15: **Quantitative comparison of depth completion with baseline methods on the iBims-1 dataset** Koch et al. (2018). Methods marked with † produce relative depth maps, where the metric depth is recovered by optimizing global scale and shift via least squares regression using the sparse depth prior. Methods marked with ‡ use scenario-specific configurations for indoor and outdoor scenes, respectively. The best and the second best results are highlighted.

| Method | 10% Noise | | | | | 5% | | | | | 3% | | | | |
|---|---|---|---|---|---|---|---|---|---|---|---|---|---|---|---|
| | RMSE↓ | MAE↓ | REL↓ | $\delta_1$↑ | Rk.↓ | RMSE↓ | MAE↓ | REL↓ | $\delta_1$↑ | Rk.↓ | RMSE↓ | MAE↓ | REL↓ | $\delta_1$↑ | Rk.↓ |
| DepthPro | 0.605 | 0.503 | 0.156 | 0.829 | 13.750 | 0.605 | 0.503 | 0.156 | 0.829 | 13.750 | 0.605 | 0.503 | 0.156 | 0.829 | 13.750 |
| UniDepth V1 | 1.166 | 1.082 | 0.370 | 0.236 | 16.000 | 1.166 | 1.082 | 0.370 | 0.236 | 16.000 | 1.166 | 1.082 | 0.370 | 0.236 | 16.000 |
| UniDepth V2 | 0.446 | 0.321 | 0.100 | 0.935 | 12.500 | 0.446 | 0.321 | 0.100 | 0.935 | 12.500 | 0.446 | 0.321 | 0.100 | 0.935 | 12.500 |
| DepthAnythingV2† | 0.332 | 0.169 | 0.041 | 0.980 | 8.750 | 0.347 | 0.172 | 0.041 | 0.978 | 9.250 | 0.348 | 0.172 | 0.041 | 0.978 | 9.250 |
| VGGT† | 0.339 | 0.178 | 0.048 | 0.965 | 10.750 | 0.337 | 0.176 | 0.047 | 0.964 | 10.500 | 0.337 | 0.176 | 0.047 | 0.964 | 10.750 |
| MoGe V1† | 0.234 | 0.117 | 0.034 | 0.985 | 7.000 | 0.231 | 0.110 | 0.031 | 0.985 | 7.000 | 0.231 | 0.110 | 0.031 | 0.985 | 7.000 |
| MoGe V2 | 0.633 | 0.540 | 0.156 | 0.707 | 14.500 | 0.633 | 0.540 | 0.156 | 0.707 | 14.500 | 0.633 | 0.540 | 0.156 | 0.707 | 14.500 |
| G2-MonoDepth‡ | 0.106 | 0.023 | 0.006 | 0.996 | 2.000 | 0.118 | 0.026 | 0.007 | 0.995 | 3.500 | 0.131 | 0.029 | 0.008 | 0.995 | 3.250 |
| OMNI-DC | 0.142 | 0.031 | 0.008 | 0.994 | 4.000 | 0.111 | 0.020 | 0.005 | 0.996 | 1.000 | 0.124 | 0.023 | 0.006 | 0.995 | 1.000 |
| PriorDA | 0.151 | 0.052 | 0.014 | 0.993 | 5.000 | 0.142 | 0.043 | 0.011 | 0.993 | 5.000 | 0.146 | 0.044 | 0.012 | 0.993 | 5.000 |
| SPNet‡ | 0.102 | 0.020 | 0.005 | 0.996 | 1.000 | 0.112 | 0.022 | 0.006 | 0.996 | 1.750 | 0.126 | 0.025 | 0.006 | 0.995 | 1.750 |
| PromptDA | 0.191 | 0.077 | 0.022 | 0.988 | 6.000 | 0.192 | 0.076 | 0.021 | 0.988 | 6.000 | 0.196 | 0.079 | 0.022 | 0.987 | 6.000 |
| WorldMirror† | 0.285 | 0.174 | 0.052 | 0.976 | 10.000 | 0.298 | 0.169 | 0.047 | 0.978 | 10.000 | 0.305 | 0.160 | 0.043 | 0.975 | 10.000 |
| MapAnything | 0.934 | 0.317 | 0.086 | 0.917 | 13.000 | 0.921 | 0.303 | 0.080 | 0.925 | 13.000 | 0.925 | 0.312 | 0.082 | 0.925 | 13.000 |
| Pow3R† | 0.272 | 0.156 | 0.044 | 0.977 | 8.500 | 0.292 | 0.157 | 0.043 | 0.972 | 8.750 | 0.310 | 0.161 | 0.043 | 0.972 | 9.250 |
| LDCM (Ours) | 0.113 | 0.022 | 0.006 | 0.996 | 2.000 | 0.120 | 0.023 | 0.006 | 0.995 | 3.000 | 0.127 | 0.024 | 0.006 | 0.995 | 1.750 |

| Method | 1% | | | | | 500 | | | | | 100 | | | | |
|---|---|---|---|---|---|---|---|---|---|---|---|---|---|---|---|
| | RMSE↓ | MAE↓ | REL↓ | $\delta_1$↑ | Rk.↓ | RMSE↓ | MAE↓ | REL↓ | $\delta_1$↑ | Rk.↓ | RMSE↓ | MAE↓ | REL↓ | $\delta_1$↑ | Rk.↓ |
| DepthPro | 0.605 | 0.503 | 0.156 | 0.829 | 13.750 | 0.605 | 0.503 | 0.156 | 0.829 | 13.750 | 0.605 | 0.503 | 0.156 | 0.829 | 13.750 |
| UniDepth V1 | 1.166 | 1.082 | 0.370 | 0.236 | 16.000 | 1.166 | 1.082 | 0.370 | 0.236 | 16.000 | 1.166 | 1.082 | 0.370 | 0.236 | 16.000 |
| UniDepth V2 | 0.446 | 0.321 | 0.100 | 0.935 | 12.500 | 0.446 | 0.321 | 0.100 | 0.935 | 12.000 | 0.446 | 0.321 | 0.100 | 0.935 | 12.000 |
| DepthAnythingV2† | 0.352 | 0.173 | 0.041 | 0.978 | 9.250 | 0.381 | 0.179 | 0.042 | 0.978 | 9.250 | 0.381 | 0.183 | 0.042 | 0.976 | 8.500 |
| VGGT† | 0.337 | 0.177 | 0.047 | 0.964 | 10.750 | 0.338 | 0.181 | 0.048 | 0.961 | 10.250 | 0.341 | 0.183 | 0.049 | 0.959 | 9.500 |
| MoGe V1† | 0.231 | 0.110 | 0.031 | 0.985 | 7.000 | 0.232 | 0.111 | 0.032 | 0.985 | 6.500 | 0.236 | 0.112 | 0.032 | 0.984 | 4.000 |
| MoGe V2 | 0.633 | 0.540 | 0.156 | 0.707 | 14.500 | 0.633 | 0.540 | 0.156 | 0.707 | 14.500 | 0.633 | 0.540 | 0.156 | 0.707 | 14.500 |
| G2-MonoDepth‡ | 0.160 | 0.040 | 0.010 | 0.992 | 4.250 | 0.232 | 0.073 | 0.019 | 0.986 | 5.000 | 0.357 | 0.178 | 0.053 | 0.960 | 10.000 |
| OMNI-DC | 0.148 | 0.030 | 0.008 | 0.993 | 2.000 | 0.188 | 0.050 | 0.013 | 0.989 | 2.500 | 0.265 | 0.096 | 0.025 | 0.979 | 4.250 |
| PriorDA | 0.156 | 0.047 | 0.013 | 0.992 | 4.250 | 0.179 | 0.057 | 0.015 | 0.991 | 2.750 | 0.211 | 0.077 | 0.020 | 0.988 | 1.750 |
| SPNet‡ | 0.156 | 0.034 | 0.008 | 0.993 | 2.500 | 0.211 | 0.055 | 0.014 | 0.988 | 3.500 | 0.270 | 0.092 | 0.023 | 0.981 | 3.750 |
| PromptDA | 0.205 | 0.084 | 0.023 | 0.988 | 6.000 | 0.237 | 0.101 | 0.027 | 0.986 | 6.000 | 0.338 | 0.153 | 0.040 | 0.975 | 6.250 |
| WorldMirror† | 0.326 | 0.160 | 0.042 | 0.971 | 8.750 | 0.342 | 0.168 | 0.043 | 0.971 | 9.250 | 0.347 | 0.172 | 0.044 | 0.968 | 7.500 |
| MapAnything | 0.923 | 0.317 | 0.083 | 0.924 | 13.000 | 0.992 | 0.407 | 0.123 | 0.893 | 13.500 | 1.057 | 0.454 | 0.125 | 0.900 | 13.500 |
| Pow3R† | 0.332 | 0.164 | 0.042 | 0.972 | 9.000 | 0.343 | 0.166 | 0.043 | 0.972 | 9.000 | 0.346 | 0.167 | 0.042 | 0.969 | 7.500 |
| LDCM (Ours) | 0.142 | 0.029 | 0.007 | 0.994 | 1.000 | 0.169 | 0.041 | 0.011 | 0.991 | 1.000 | 0.202 | 0.061 | 0.015 | 0.988 | 1.000 |

| Method | SIFT | | | | | ORB | | | | | Virtual-Lidar-32-Lines | | | | |
|---|---|---|---|---|---|---|---|---|---|---|---|---|---|---|---|
| | RMSE↓ | MAE↓ | REL↓ | $\delta_1$↑ | Rk.↓ | RMSE↓ | MAE↓ | REL↓ | $\delta_1$↑ | Rk.↓ | RMSE↓ | MAE↓ | REL↓ | $\delta_1$↑ | Rk.↓ |
| DepthPro | 0.605 | 0.503 | 0.156 | 0.829 | 13.750 | 0.605 | 0.503 | 0.156 | 0.829 | 13.750 | 0.605 | 0.503 | 0.156 | 0.829 | 13.750 |
| UniDepth V1 | 1.166 | 1.082 | 0.370 | 0.236 | 16.000 | 1.166 | 1.082 | 0.370 | 0.236 | 16.000 | 1.166 | 1.082 | 0.370 | 0.236 | 16.000 |
| UniDepth V2 | 0.446 | 0.321 | 0.100 | 0.935 | 11.250 | 0.446 | 0.321 | 0.100 | 0.935 | 9.750 | 0.446 | 0.321 | 0.100 | 0.935 | 12.250 |
| DepthAnythingV2† | 0.321 | 0.185 | 0.048 | 0.972 | 5.750 | 0.334 | 0.198 | 0.052 | 0.963 | 4.750 | 0.373 | 0.206 | 0.049 | 0.969 | 10.500 |
| VGGT† | 0.388 | 0.256 | 0.073 | 0.933 | 10.750 | 0.399 | 0.270 | 0.077 | 0.930 | 8.750 | 0.337 | 0.178 | 0.048 | 0.963 | 10.250 |
| MoGe V1† | 0.257 | 0.151 | 0.048 | 0.967 | 4.750 | 0.270 | 0.165 | 0.053 | 0.957 | 4.500 | 0.231 | 0.110 | 0.031 | 0.985 | 6.500 |
| MoGe V2 | 0.633 | 0.540 | 0.156 | 0.707 | 14.500 | 0.633 | 0.540 | 0.156 | 0.707 | 14.500 | 0.633 | 0.540 | 0.156 | 0.707 | 14.500 |
| G2-MonoDepth‡ | 0.338 | 0.197 | 0.065 | 0.923 | 9.000 | 0.376 | 0.230 | 0.076 | 0.903 | 8.000 | 0.193 | 0.057 | 0.015 | 0.990 | 3.000 |
| OMNI-DC | 0.260 | 0.119 | 0.036 | 0.960 | 4.750 | 0.313 | 0.166 | 0.054 | 0.925 | 6.250 | 0.160 | 0.036 | 0.010 | 0.992 | 1.000 |
| PriorDA | 0.180 | 0.076 | 0.022 | 0.989 | 2.000 | 0.216 | 0.103 | 0.032 | 0.982 | 2.000 | 0.201 | 0.088 | 0.021 | 0.983 | 5.500 |
| SPNet‡ | 0.224 | 0.096 | 0.028 | 0.981 | 3.000 | 0.264 | 0.129 | 0.041 | 0.961 | 3.250 | 0.173 | 0.040 | 0.010 | 0.992 | 1.500 |
| PromptDA | 0.317 | 0.182 | 0.053 | 0.948 | 6.500 | 0.394 | 0.244 | 0.072 | 0.903 | 8.250 | 0.209 | 0.086 | 0.024 | 0.986 | 5.250 |
| WorldMirror† | 0.382 | 0.236 | 0.066 | 0.940 | 8.750 | 0.567 | 0.380 | 0.103 | 0.897 | 12.000 | 0.336 | 0.161 | 0.041 | 0.972 | 8.000 |
| MapAnything | 0.977 | 0.408 | 0.115 | 0.901 | 13.500 | 1.002 | 0.427 | 0.122 | 0.896 | 13.500 | 0.973 | 0.365 | 0.098 | 0.912 | 13.250 |
| Pow3R† | 0.394 | 0.254 | 0.069 | 0.934 | 10.250 | 0.411 | 0.274 | 0.078 | 0.935 | 9.000 | 0.336 | 0.170 | 0.044 | 0.969 | 8.750 |
| LDCM (Ours) | 0.170 | 0.053 | 0.015 | 0.991 | 1.000 | 0.186 | 0.067 | 0.020 | 0.989 | 1.000 | 0.196 | 0.076 | 0.018 | 0.983 | 4.500 |

| Method | Virtual-Lidar-16-Lines | | | | | Virtual-Lidar-8-Lines | | | | | Average | | | | |
|---|---|---|---|---|---|---|---|---|---|---|---|---|---|---|---|
| | RMSE↓ | MAE↓ | REL↓ | $\delta_1$↑ | Rk.↓ | RMSE↓ | MAE↓ | REL↓ | $\delta_1$↑ | Rk.↓ | RMSE↓ | MAE↓ | REL↓ | $\delta_1$↑ | Rk.↓ |
| DepthPro | 0.605 | 0.503 | 0.156 | 0.829 | 13.750 | 0.605 | 0.503 | 0.156 | 0.829 | 13.750 | 0.605 | 0.503 | 0.156 | 0.829 | 13.750 |
| UniDepth V1 | 1.166 | 1.082 | 0.370 | 0.236 | 16.000 | 1.166 | 1.082 | 0.370 | 0.236 | 16.000 | 1.166 | 1.082 | 0.370 | 0.236 | 16.000 |
| UniDepth V2 | 0.446 | 0.321 | 0.100 | 0.935 | 12.000 | 0.446 | 0.321 | 0.100 | 0.935 | 12.000 | 0.446 | 0.321 | 0.100 | 0.935 | 11.932 |
| DepthAnythingV2† | 0.333 | 0.166 | 0.040 | 0.978 | 8.250 | 0.336 | 0.166 | 0.040 | 0.979 | 7.750 | 0.349 | 0.179 | 0.043 | 0.975 | 8.295 |
| VGGT† | 0.338 | 0.179 | 0.048 | 0.963 | 10.750 | 0.339 | 0.180 | 0.048 | 0.962 | 10.500 | 0.348 | 0.194 | 0.053 | 0.957 | 10.318 |
| MoGe V1† | 0.232 | 0.111 | 0.032 | 0.985 | 7.000 | 0.233 | 0.111 | 0.032 | 0.984 | 5.250 | 0.238 | 0.120 | 0.035 | 0.981 | 6.045 |
| MoGe V2 | 0.633 | 0.540 | 0.156 | 0.707 | 14.500 | 0.633 | 0.540 | 0.156 | 0.707 | 14.500 | 0.633 | 0.540 | 0.156 | 0.707 | 14.500 |
| G2-MonoDepth‡ | 0.216 | 0.070 | 0.018 | 0.987 | 5.250 | 0.273 | 0.109 | 0.029 | 0.978 | 6.250 | 0.227 | 0.094 | 0.028 | 0.973 | 5.409 |
| OMNI-DC | 0.176 | 0.046 | 0.012 | 0.990 | 2.500 | 0.225 | 0.075 | 0.021 | 0.984 | 3.000 | 0.192 | 0.063 | 0.018 | 0.982 | 2.932 |
| PriorDA | 0.169 | 0.056 | 0.015 | 0.991 | 3.000 | 0.187 | 0.070 | 0.019 | 0.990 | 2.000 | 0.176 | 0.065 | 0.018 | 0.990 | 3.477 |
| SPNet‡ | 0.198 | 0.052 | 0.013 | 0.989 | 3.500 | 0.241 | 0.080 | 0.021 | 0.984 | 3.750 | 0.189 | 0.059 | 0.016 | 0.987 | 2.659 |
| PromptDA | 0.217 | 0.090 | 0.025 | 0.988 | 5.750 | 0.242 | 0.109 | 0.030 | 0.984 | 5.000 | 0.249 | 0.116 | 0.033 | 0.975 | 6.091 |
| WorldMirror† | 0.337 | 0.162 | 0.042 | 0.971 | 8.750 | 0.342 | 0.168 | 0.043 | 0.970 | 9.000 | 0.352 | 0.192 | 0.051 | 0.963 | 9.205 |
| MapAnything | 0.972 | 0.394 | 0.113 | 0.901 | 13.500 | 0.975 | 0.410 | 0.117 | 0.910 | 13.500 | 0.968 | 0.374 | 0.104 | 0.909 | 13.295 |
| Pow3R† | 0.342 | 0.169 | 0.043 | 0.971 | 10.000 | 0.343 | 0.170 | 0.044 | 0.971 | 10.000 | 0.338 | 0.183 | 0.049 | 0.965 | 9.091 |
| LDCM (Ours) | 0.165 | 0.039 | 0.010 | 0.992 | 1.000 | 0.181 | 0.050 | 0.013 | 0.991 | 1.000 | 0.161 | 0.044 | 0.012 | 0.991 | 1.659 |

Table 16: **Quantitative comparison of depth completion with baseline methods on the indoor scenes of the ETH3D dataset** Schops et al. (2017). Methods marked with † produce relative depth maps, where the metric depth is recovered by optimizing global scale and shift via least squares regression using the sparse depth prior. Methods marked with ‡ use scenario-specific configurations for indoor and outdoor scenes, respectively. The best and the second best results are highlighted.

| Method | 10% Noise | | | | | 5% | | | | | 3% | | | | |
|---|---|---|---|---|---|---|---|---|---|---|---|---|---|---|---|
| | RMSE↓ | MAE↓ | REL↓ | $\delta_1$↑ | Rk.↓ | RMSE↓ | MAE↓ | REL↓ | $\delta_1$↑ | Rk.↓ | RMSE↓ | MAE↓ | REL↓ | $\delta_1$↑ | Rk.↓ |
| DepthPro | 0.831 | 0.670 | 0.192 | 0.705 | 14.750 | 0.831 | 0.670 | 0.192 | 0.705 | 14.500 | 0.831 | 0.670 | 0.192 | 0.705 | 14.500 |
| UniDepth V1 | 2.549 | 2.330 | 0.695 | 0.071 | 16.000 | 2.549 | 2.330 | 0.695 | 0.071 | 16.000 | 2.549 | 2.330 | 0.695 | 0.071 | 16.000 |
| UniDepth V2 | 0.660 | 0.562 | 0.169 | 0.799 | 13.250 | 0.660 | 0.562 | 0.169 | 0.799 | 13.250 | 0.660 | 0.562 | 0.169 | 0.799 | 13.250 |
| DepthAnythingV2† | 0.808 | 0.203 | 0.037 | 0.989 | 10.000 | 1.017 | 0.220 | 0.038 | 0.988 | 10.500 | 1.002 | 0.220 | 0.038 | 0.988 | 10.750 |
| VGGT† | 0.391 | 0.199 | 0.049 | 0.966 | 10.250 | 0.386 | 0.201 | 0.050 | 0.959 | 11.000 | 0.386 | 0.202 | 0.050 | 0.959 | 10.500 |
| MoGe V1† | 0.214 | 0.122 | 0.031 | 0.995 | 7.000 | 0.204 | 0.114 | 0.028 | 0.995 | 7.000 | 0.204 | 0.114 | 0.028 | 0.994 | 7.000 |
| MoGe V2 | 0.542 | 0.419 | 0.117 | 0.784 | 12.750 | 0.542 | 0.419 | 0.117 | 0.784 | 12.750 | 0.542 | 0.419 | 0.117 | 0.784 | 12.750 |
| G2-MonoDepth‡ | 0.055 | 0.011 | 0.002 | 1.000 | 2.000 | 0.062 | 0.012 | 0.002 | 1.000 | 2.750 | 0.076 | 0.014 | 0.003 | 1.000 | 3.000 |
| OMNI-DC | 0.116 | 0.017 | 0.003 | 0.999 | 4.250 | 0.052 | 0.006 | 0.001 | 1.000 | 1.250 | 0.063 | 0.008 | 0.002 | 1.000 | 1.250 |
| PriorDA | 0.110 | 0.040 | 0.008 | 0.999 | 4.500 | 0.090 | 0.026 | 0.006 | 0.999 | 5.000 | 0.093 | 0.027 | 0.006 | 0.999 | 4.750 |
| SPNet‡ | 0.076 | 0.010 | 0.001 | 1.000 | 1.750 | 0.080 | 0.010 | 0.001 | 1.000 | 2.250 | 0.101 | 0.013 | 0.002 | 1.000 | 2.500 |
| PromptDA | 0.173 | 0.076 | 0.018 | 0.996 | 6.000 | 0.148 | 0.067 | 0.017 | 0.996 | 6.000 | 0.159 | 0.071 | 0.019 | 0.995 | 6.000 |
| WorldMirror† | 0.328 | 0.197 | 0.050 | 0.984 | 9.500 | 0.280 | 0.154 | 0.038 | 0.992 | 8.000 | 0.261 | 0.141 | 0.035 | 0.991 | 8.000 |
| MapAnything | 1.076 | 0.272 | 0.055 | 0.951 | 12.750 | 1.039 | 0.248 | 0.046 | 0.962 | 12.250 | 1.083 | 0.271 | 0.051 | 0.958 | 12.750 |
| Pow3R† | 0.323 | 0.192 | 0.044 | 0.980 | 8.750 | 0.309 | 0.180 | 0.043 | 0.975 | 9.500 | 0.286 | 0.159 | 0.038 | 0.980 | 9.250 |
| LDCM (Ours) | 0.046 | 0.009 | 0.002 | 1.000 | 1.250 | 0.051 | 0.009 | 0.002 | 1.000 | 1.750 | 0.058 | 0.009 | 0.002 | 1.000 | 1.250 |

| Method | 1% | | | | | 500 | | | | | 100 | | | | |
|---|---|---|---|---|---|---|---|---|---|---|---|---|---|---|---|
| | RMSE↓ | MAE↓ | REL↓ | $\delta_1$↑ | Rk.↓ | RMSE↓ | MAE↓ | REL↓ | $\delta_1$↑ | Rk.↓ | RMSE↓ | MAE↓ | REL↓ | $\delta_1$↑ | Rk.↓ |
| DepthPro | 0.831 | 0.670 | 0.192 | 0.705 | 14.500 | 0.831 | 0.670 | 0.192 | 0.705 | 14.500 | 0.831 | 0.670 | 0.192 | 0.705 | 14.250 |
| UniDepth V1 | 2.549 | 2.330 | 0.695 | 0.071 | 16.000 | 2.549 | 2.330 | 0.695 | 0.071 | 16.000 | 2.549 | 2.330 | 0.695 | 0.071 | 16.000 |
| UniDepth V2 | 0.660 | 0.562 | 0.169 | 0.799 | 13.250 | 0.660 | 0.562 | 0.169 | 0.799 | 13.250 | 0.660 | 0.562 | 0.169 | 0.799 | 13.000 |
| DepthAnythingV2† | 1.016 | 0.221 | 0.038 | 0.987 | 11.000 | 1.016 | 0.220 | 0.038 | 0.987 | 11.000 | 1.006 | 0.222 | 0.038 | 0.987 | 10.250 |
| VGGT† | 0.387 | 0.203 | 0.050 | 0.959 | 10.500 | 0.389 | 0.200 | 0.050 | 0.962 | 10.500 | 0.398 | 0.198 | 0.048 | 0.965 | 10.250 |
| MoGe V1† | 0.205 | 0.114 | 0.028 | 0.994 | 7.000 | 0.206 | 0.114 | 0.028 | 0.995 | 6.000 | 0.214 | 0.118 | 0.029 | 0.995 | 3.750 |
| MoGe V2 | 0.542 | 0.419 | 0.117 | 0.784 | 12.750 | 0.542 | 0.419 | 0.117 | 0.784 | 12.250 | 0.542 | 0.419 | 0.117 | 0.784 | 12.250 |
| G2-MonoDepth‡ | 0.118 | 0.024 | 0.005 | 0.999 | 3.250 | 0.215 | 0.061 | 0.012 | 0.997 | 5.000 | 0.443 | 0.190 | 0.045 | 0.972 | 9.750 |
| OMNI-DC | 0.089 | 0.012 | 0.002 | 0.999 | 1.250 | 0.153 | 0.031 | 0.006 | 0.998 | 2.500 | 0.309 | 0.089 | 0.017 | 0.992 | 4.750 |
| PriorDA | 0.103 | 0.029 | 0.006 | 0.999 | 3.500 | 0.133 | 0.037 | 0.007 | 0.999 | 2.500 | 0.222 | 0.064 | 0.012 | 0.997 | 2.250 |
| SPNet‡ | 0.145 | 0.020 | 0.003 | 0.999 | 3.000 | 0.223 | 0.039 | 0.006 | 0.998 | 3.750 | 0.323 | 0.085 | 0.015 | 0.994 | 4.250 |
| PromptDA | 0.188 | 0.078 | 0.019 | 0.995 | 6.000 | 0.268 | 0.106 | 0.025 | 0.990 | 7.000 | 0.396 | 0.154 | 0.033 | 0.987 | 7.500 |
| WorldMirror† | 0.241 | 0.125 | 0.031 | 0.992 | 8.000 | 0.234 | 0.121 | 0.029 | 0.992 | 7.500 | 0.246 | 0.127 | 0.031 | 0.992 | 5.250 |
| MapAnything | 1.162 | 0.333 | 0.068 | 0.945 | 12.750 | 1.390 | 0.549 | 0.144 | 0.873 | 13.250 | 1.676 | 0.702 | 0.149 | 0.876 | 13.750 |
| Pow3R† | 0.279 | 0.146 | 0.034 | 0.984 | 9.250 | 0.287 | 0.145 | 0.034 | 0.985 | 9.250 | 0.298 | 0.150 | 0.035 | 0.982 | 7.250 |
| LDCM (Ours) | 0.074 | 0.012 | 0.002 | 0.999 | 1.000 | 0.106 | 0.021 | 0.004 | 0.999 | 1.000 | 0.157 | 0.041 | 0.008 | 0.998 | 1.000 |

| Method | SIFT | | | | | ORB | | | | | Virtual-Lidar-32-Lines | | | | |
|---|---|---|---|---|---|---|---|---|---|---|---|---|---|---|---|
| | RMSE↓ | MAE↓ | REL↓ | $\delta_1$↑ | Rk.↓ | RMSE↓ | MAE↓ | REL↓ | $\delta_1$↑ | Rk.↓ | RMSE↓ | MAE↓ | REL↓ | $\delta_1$↑ | Rk.↓ |
| DepthPro | 0.831 | 0.670 | 0.192 | 0.705 | 14.750 | 0.831 | 0.670 | 0.192 | 0.705 | 14.750 | 0.831 | 0.670 | 0.192 | 0.705 | 14.500 |
| UniDepth V1 | 2.549 | 2.330 | 0.695 | 0.071 | 16.000 | 2.549 | 2.330 | 0.695 | 0.071 | 16.000 | 2.549 | 2.330 | 0.695 | 0.071 | 16.000 |
| UniDepth V2 | 0.660 | 0.562 | 0.169 | 0.799 | 13.500 | 0.660 | 0.562 | 0.169 | 0.799 | 13.500 | 0.660 | 0.562 | 0.169 | 0.799 | 13.250 |
| DepthAnythingV2† | 0.436 | 0.179 | 0.040 | 0.990 | 6.500 | 0.396 | 0.183 | 0.042 | 0.990 | 5.000 | 0.918 | 0.206 | 0.037 | 0.989 | 10.750 |
| VGGT† | 0.416 | 0.242 | 0.068 | 0.937 | 10.750 | 0.441 | 0.277 | 0.084 | 0.911 | 10.750 | 0.387 | 0.209 | 0.053 | 0.953 | 10.750 |
| MoGe V1† | 0.228 | 0.141 | 0.040 | 0.978 | 4.500 | 0.250 | 0.164 | 0.048 | 0.965 | 4.750 | 0.205 | 0.115 | 0.029 | 0.993 | 7.000 |
| MoGe V2 | 0.542 | 0.419 | 0.117 | 0.784 | 12.500 | 0.542 | 0.419 | 0.117 | 0.784 | 12.750 | 0.542 | 0.419 | 0.117 | 0.784 | 12.500 |
| G2-MonoDepth‡ | 0.435 | 0.234 | 0.076 | 0.910 | 10.500 | 0.440 | 0.237 | 0.077 | 0.914 | 9.250 | 0.158 | 0.048 | 0.009 | 0.998 | 4.750 |
| OMNI-DC | 0.248 | 0.114 | 0.035 | 0.966 | 5.000 | 0.284 | 0.144 | 0.045 | 0.954 | 5.000 | 0.107 | 0.017 | 0.003 | 0.999 | 1.500 |
| PriorDA | 0.189 | 0.084 | 0.028 | 0.984 | 2.250 | 0.227 | 0.108 | 0.033 | 0.980 | 2.250 | 0.111 | 0.030 | 0.006 | 0.999 | 3.000 |
| SPNet‡ | 0.287 | 0.108 | 0.031 | 0.981 | 4.000 | 0.308 | 0.127 | 0.036 | 0.974 | 3.750 | 0.164 | 0.026 | 0.004 | 0.999 | 3.000 |
| PromptDA | 0.352 | 0.205 | 0.055 | 0.951 | 8.750 | 0.367 | 0.221 | 0.061 | 0.938 | 7.750 | 0.182 | 0.076 | 0.018 | 0.997 | 6.000 |
| WorldMirror† | 0.279 | 0.171 | 0.046 | 0.978 | 6.000 | 0.494 | 0.319 | 0.081 | 0.936 | 10.250 | 0.233 | 0.122 | 0.030 | 0.992 | 8.000 |
| MapAnything | 1.427 | 0.545 | 0.129 | 0.889 | 13.250 | 1.353 | 0.485 | 0.114 | 0.907 | 13.000 | 1.330 | 0.460 | 0.108 | 0.910 | 13.000 |
| Pow3R† | 0.311 | 0.174 | 0.045 | 0.975 | 7.000 | 0.341 | 0.209 | 0.055 | 0.961 | 6.500 | 0.283 | 0.151 | 0.036 | 0.979 | 9.250 |
| LDCM (Ours) | 0.127 | 0.045 | 0.012 | 0.995 | 1.000 | 0.139 | 0.056 | 0.016 | 0.996 | 1.000 | 0.087 | 0.014 | 0.003 | 0.999 | 1.000 |

| Method | Virtual-Lidar-16-Lines | | | | | Virtual-Lidar-8-Lines | | | | | Average | | | | |
|---|---|---|---|---|---|---|---|---|---|---|---|---|---|---|---|
| | RMSE↓ | MAE↓ | REL↓ | $\delta_1$↑ | Rk.↓ | RMSE↓ | MAE↓ | REL↓ | $\delta_1$↑ | Rk.↓ | RMSE↓ | MAE↓ | REL↓ | $\delta_1$↑ | Rk.↓ |
| DepthPro | 0.831 | 0.670 | 0.192 | 0.705 | 14.500 | 0.831 | 0.670 | 0.192 | 0.705 | 14.500 | 0.831 | 0.670 | 0.192 | 0.705 | 14.545 |
| UniDepth V1 | 2.549 | 2.330 | 0.695 | 0.071 | 16.000 | 2.549 | 2.330 | 0.695 | 0.071 | 16.000 | 2.549 | 2.330 | 0.695 | 0.071 | 16.000 |
| UniDepth V2 | 0.660 | 0.562 | 0.169 | 0.799 | 13.250 | 0.660 | 0.562 | 0.169 | 0.799 | 13.250 | 0.660 | 0.562 | 0.169 | 0.799 | 13.273 |
| DepthAnythingV2† | 0.924 | 0.206 | 0.037 | 0.989 | 10.750 | 0.889 | 0.203 | 0.037 | 0.988 | 10.500 | 0.857 | 0.208 | 0.038 | 0.988 | 9.727 |
| VGGT† | 0.387 | 0.207 | 0.053 | 0.954 | 10.750 | 0.393 | 0.218 | 0.056 | 0.949 | 10.750 | 0.396 | 0.214 | 0.056 | 0.952 | 10.523 |
| MoGe V1† | 0.206 | 0.117 | 0.029 | 0.993 | 6.750 | 0.209 | 0.121 | 0.030 | 0.990 | 5.250 | 0.213 | 0.123 | 0.032 | 0.990 | 6.000 |
| MoGe V2 | 0.542 | 0.419 | 0.117 | 0.784 | 12.250 | 0.542 | 0.419 | 0.117 | 0.784 | 12.500 | 0.542 | 0.419 | 0.117 | 0.784 | 12.545 |
| G2-MonoDepth‡ | 0.202 | 0.060 | 0.012 | 0.997 | 4.750 | 0.315 | 0.126 | 0.027 | 0.990 | 6.500 | 0.229 | 0.092 | 0.025 | 0.980 | 5.591 |
| OMNI-DC | 0.139 | 0.028 | 0.006 | 0.998 | 2.500 | 0.223 | 0.061 | 0.013 | 0.995 | 3.000 | 0.162 | 0.048 | 0.012 | 0.991 | 2.932 |
| PriorDA | 0.129 | 0.039 | 0.008 | 0.999 | 2.500 | 0.166 | 0.059 | 0.013 | 0.997 | 2.000 | 0.143 | 0.049 | 0.012 | 0.996 | 3.136 |
| SPNet‡ | 0.202 | 0.041 | 0.007 | 0.998 | 3.500 | 0.285 | 0.094 | 0.019 | 0.994 | 4.500 | 0.199 | 0.052 | 0.011 | 0.994 | 3.295 |
| PromptDA | 0.233 | 0.090 | 0.021 | 0.994 | 6.250 | 0.289 | 0.120 | 0.028 | 0.990 | 5.750 | 0.250 | 0.115 | 0.029 | 0.984 | 6.636 |
| WorldMirror† | 0.234 | 0.122 | 0.030 | 0.992 | 8.000 | 0.242 | 0.131 | 0.032 | 0.988 | 7.250 | 0.279 | 0.157 | 0.039 | 0.984 | 7.795 |
| MapAnything | 1.323 | 0.520 | 0.133 | 0.890 | 13.250 | 1.325 | 0.487 | 0.115 | 0.908 | 13.000 | 1.289 | 0.443 | 0.101 | 0.915 | 13.000 |
| Pow3R† | 0.291 | 0.148 | 0.034 | 0.982 | 9.250 | 0.289 | 0.156 | 0.036 | 0.978 | 8.750 | 0.300 | 0.165 | 0.039 | 0.978 | 8.545 |
| LDCM (Ours) | 0.104 | 0.020 | 0.004 | 0.999 | 1.000 | 0.136 | 0.034 | 0.007 | 0.998 | 1.000 | 0.099 | 0.025 | 0.006 | 0.998 | 1.114 |

Table 17: **Quantitative comparison of depth completion with baseline methods on the outdoor scenes of the ETH3D dataset Schops et al. (2017).** Methods marked with † produce relative depth maps, where the metric depth is recovered by optimizing global scale and shift via least squares regression using the sparse depth prior. Methods marked with ‡ use scenario-specific configurations for indoor and outdoor scenes, respectively. The best and the second best results are highlighted.

| Method | Virtual-Lidar-64-Lines | | | | | Virtual-Lidar-32-Lines | | | | | Virtual-Lidar-16-Lines | | | | |
|---|---|---|---|---|---|---|---|---|---|---|---|---|---|---|---|
| | RMSE↓ | MAE↓ | REL↓ | $\delta_1$↑ | Rk.↓ | RMSE↓ | MAE↓ | REL↓ | $\delta_1$↑ | Rk.↓ | RMSE↓ | MAE↓ | REL↓ | $\delta_1$↑ | Rk.↓ |
| DepthPro | 5.567 | 4.454 | 0.411 | 0.248 | 15.500 | 5.567 | 4.454 | 0.411 | 0.248 | 15.500 | 5.567 | 4.454 | 0.411 | 0.248 | 15.500 |
| UniDepth V1 | 4.414 | 4.009 | 0.463 | 0.160 | 15.500 | 4.414 | 4.009 | 0.463 | 0.160 | 15.500 | 4.414 | 4.009 | 0.463 | 0.160 | 15.500 |
| UniDepth V2 | 2.600 | 1.775 | 0.230 | 0.652 | 13.500 | 2.600 | 1.775 | 0.230 | 0.652 | 13.500 | 2.600 | 1.775 | 0.230 | 0.652 | 13.250 |
| DepthAnythingV2† | 3.443 | 0.648 | 0.058 | 0.972 | 10.000 | 3.361 | 0.641 | 0.057 | 0.973 | 10.000 | 3.757 | 0.673 | 0.059 | 0.972 | 10.000 |
| VGGT† | 0.650 | 0.383 | 0.056 | 0.957 | 9.000 | 0.651 | 0.382 | 0.056 | 0.957 | 9.000 | 0.653 | 0.384 | 0.056 | 0.957 | 8.500 |
| MoGe V1† | 2.554 | 1.190 | 0.143 | 0.876 | 12.500 | 2.561 | 0.801 | 0.124 | 0.899 | 11.250 | 3.301 | 0.752 | 0.111 | 0.921 | 11.500 |
| MoGe V2 | 1.152 | 0.819 | 0.111 | 0.893 | 11.250 | 1.152 | 0.819 | 0.111 | 0.893 | 11.250 | 1.152 | 0.819 | 0.111 | 0.893 | 11.250 |
| G2-MonoDepth‡ | 0.308 | 0.068 | 0.011 | 0.996 | 4.500 | 0.442 | 0.111 | 0.017 | 0.992 | 4.750 | 0.610 | 0.175 | 0.027 | 0.985 | 4.750 |
| OMNI-DC | 0.234 | 0.035 | 0.004 | 0.997 | 2.000 | 0.281 | 0.050 | 0.006 | 0.997 | 2.000 | 0.354 | 0.085 | 0.010 | 0.995 | 2.250 |
| PriorDA | 0.267 | 0.083 | 0.010 | 0.996 | 4.000 | 0.291 | 0.090 | 0.011 | 0.996 | 3.500 | 0.334 | 0.111 | 0.014 | 0.995 | 2.500 |
| SPNet‡ | 0.289 | 0.042 | 0.005 | 0.998 | 2.750 | 0.474 | 0.070 | 0.008 | 0.996 | 3.500 | 0.671 | 0.123 | 0.015 | 0.993 | 5.000 |
| PromptDA | 0.935 | 0.355 | 0.044 | 0.964 | 7.250 | 0.972 | 0.365 | 0.045 | 0.961 | 7.750 | 1.030 | 0.371 | 0.041 | 0.970 | 7.500 |
| WorldMirror† | 0.584 | 0.327 | 0.048 | 0.963 | 6.750 | 0.631 | 0.352 | 0.050 | 0.958 | 7.250 | 0.647 | 0.362 | 0.052 | 0.955 | 7.750 |
| MapAnything | 2.781 | 0.994 | 0.109 | 0.897 | 11.750 | 2.804 | 1.092 | 0.126 | 0.885 | 13.000 | 2.814 | 1.191 | 0.148 | 0.865 | 12.750 |
| Pow3R† | 0.637 | 0.361 | 0.053 | 0.958 | 8.000 | 0.636 | 0.344 | 0.050 | 0.964 | 6.750 | 0.634 | 0.337 | 0.050 | 0.962 | 6.500 |
| LDCM (Ours) | 0.204 | 0.033 | 0.004 | 0.998 | 1.000 | 0.246 | 0.042 | 0.005 | 0.998 | 1.000 | 0.294 | 0.059 | 0.007 | 0.997 | 1.000 |

| Method | Virtual-Lidar-8-Lines | | | | | Virtual-Lidar-4-Lines | | | | | 10% Noise | | | | |
|---|---|---|---|---|---|---|---|---|---|---|---|---|---|---|---|
| | RMSE↓ | MAE↓ | REL↓ | $\delta_1$↑ | Rk.↓ | RMSE↓ | MAE↓ | REL↓ | $\delta_1$↑ | Rk.↓ | RMSE↓ | MAE↓ | REL↓ | $\delta_1$↑ | Rk.↓ |
| DepthPro | 5.567 | 4.454 | 0.411 | 0.248 | 15.500 | 5.567 | 4.454 | 0.411 | 0.248 | 15.500 | 5.567 | 4.454 | 0.411 | 0.248 | 15.500 |
| UniDepth V1 | 4.414 | 4.009 | 0.463 | 0.160 | 15.500 | 4.414 | 4.009 | 0.463 | 0.160 | 15.500 | 4.414 | 4.009 | 0.463 | 0.160 | 15.500 |
| UniDepth V2 | 2.600 | 1.775 | 0.230 | 0.652 | 13.250 | 2.600 | 1.775 | 0.230 | 0.652 | 13.250 | 2.600 | 1.775 | 0.230 | 0.652 | 13.750 |
| DepthAnythingV2† | 3.392 | 0.642 | 0.059 | 0.971 | 9.250 | 3.319 | 0.608 | 0.063 | 0.963 | 6.000 | 3.052 | 0.620 | 0.057 | 0.972 | 10.000 |
| VGGT† | 0.670 | 0.393 | 0.058 | 0.953 | 7.000 | 0.742 | 0.463 | 0.074 | 0.926 | 4.500 | 0.661 | 0.387 | 0.054 | 0.965 | 7.000 |
| MoGe V1† | 3.993 | 0.707 | 0.097 | 0.941 | 11.500 | 4.979 | 0.539 | 0.097 | 0.948 | 8.000 | 2.184 | 1.702 | 0.183 | 0.838 | 12.500 |
| MoGe V2 | 1.152 | 0.819 | 0.111 | 0.893 | 11.250 | 1.152 | 0.819 | 0.111 | 0.893 | 8.750 | 1.152 | 0.819 | 0.111 | 0.893 | 11.500 |
| G2-MonoDepth‡ | 1.005 | 0.409 | 0.057 | 0.954 | 7.000 | 1.862 | 1.117 | 0.171 | 0.774 | 11.750 | 0.128 | 0.031 | 0.004 | 0.999 | 3.250 |
| OMNI-DC | 0.594 | 0.221 | 0.025 | 0.983 | 3.000 | 1.282 | 0.751 | 0.095 | 0.888 | 8.500 | 0.177 | 0.028 | 0.003 | 0.999 | 3.000 |
| PriorDA | 0.489 | 0.181 | 0.022 | 0.989 | 2.000 | 1.071 | 0.559 | 0.090 | 0.896 | 6.250 | 0.280 | 0.113 | 0.012 | 0.997 | 5.000 |
| SPNet‡ | 1.104 | 0.327 | 0.045 | 0.970 | 5.250 | 2.088 | 0.898 | 0.134 | 0.856 | 11.000 | 0.110 | 0.019 | 0.002 | 1.000 | 1.000 |
| PromptDA | 1.311 | 0.492 | 0.057 | 0.952 | 8.500 | 1.372 | 0.749 | 0.092 | 0.902 | 7.500 | 0.912 | 0.423 | 0.049 | 0.962 | 8.250 |
| WorldMirror† | 0.715 | 0.406 | 0.057 | 0.941 | 7.250 | 0.732 | 0.436 | 0.071 | 0.919 | 4.000 | 0.694 | 0.404 | 0.056 | 0.972 | 7.250 |
| MapAnything | 2.917 | 1.231 | 0.145 | 0.867 | 12.750 | 3.036 | 1.401 | 0.153 | 0.866 | 12.000 | 2.189 | 0.653 | 0.079 | 0.921 | 11.250 |
| Pow3R† | 0.645 | 0.333 | 0.050 | 0.962 | 5.000 | 0.712 | 0.390 | 0.063 | 0.933 | 2.500 | 0.658 | 0.415 | 0.056 | 0.960 | 8.000 |
| LDCM (Ours) | 0.420 | 0.107 | 0.012 | 0.994 | 1.000 | 0.551 | 0.229 | 0.029 | 0.986 | 1.000 | 0.116 | 0.023 | 0.002 | 0.999 | 1.750 |

| Method | 5% | | | | | 3% | | | | | 1% | | | | |
|---|---|---|---|---|---|---|---|---|---|---|---|---|---|---|---|
| | RMSE↓ | MAE↓ | REL↓ | $\delta_1$↑ | Rk.↓ | RMSE↓ | MAE↓ | REL↓ | $\delta_1$↑ | Rk.↓ | RMSE↓ | MAE↓ | REL↓ | $\delta_1$↑ | Rk.↓ |
| DepthPro | 5.567 | 4.454 | 0.411 | 0.248 | 15.500 | 5.567 | 4.454 | 0.411 | 0.248 | 15.500 | 5.567 | 4.454 | 0.411 | 0.248 | 15.500 |
| UniDepth V1 | 4.414 | 4.009 | 0.463 | 0.160 | 15.500 | 4.414 | 4.009 | 0.463 | 0.160 | 15.500 | 4.414 | 4.009 | 0.463 | 0.160 | 15.500 |
| UniDepth V2 | 2.600 | 1.775 | 0.230 | 0.652 | 13.750 | 2.600 | 1.775 | 0.230 | 0.652 | 13.750 | 2.600 | 1.775 | 0.230 | 0.652 | 13.750 |
| DepthAnythingV2† | 3.707 | 0.672 | 0.059 | 0.971 | 10.250 | 3.746 | 0.673 | 0.060 | 0.971 | 10.250 | 3.769 | 0.668 | 0.060 | 0.971 | 10.000 |
| VGGT† | 0.649 | 0.382 | 0.056 | 0.959 | 7.750 | 0.649 | 0.382 | 0.056 | 0.959 | 7.250 | 0.650 | 0.382 | 0.055 | 0.959 | 8.250 |
| MoGe V1† | 2.057 | 1.574 | 0.177 | 0.838 | 12.500 | 2.151 | 1.342 | 0.157 | 0.858 | 12.500 | 2.139 | 0.940 | 0.134 | 0.884 | 12.500 |
| MoGe V2 | 1.152 | 0.819 | 0.111 | 0.893 | 11.500 | 1.152 | 0.819 | 0.111 | 0.893 | 11.500 | 1.152 | 0.819 | 0.111 | 0.893 | 11.500 |
| G2-MonoDepth‡ | 0.156 | 0.036 | 0.005 | 0.999 | 3.250 | 0.202 | 0.044 | 0.006 | 0.998 | 3.750 | 0.367 | 0.079 | 0.012 | 0.995 | 4.500 |
| OMNI-DC | 0.155 | 0.020 | 0.002 | 0.999 | 1.500 | 0.190 | 0.025 | 0.003 | 0.998 | 2.000 | 0.255 | 0.041 | 0.005 | 0.997 | 2.000 |
| PriorDA | 0.242 | 0.079 | 0.009 | 0.997 | 5.000 | 0.253 | 0.080 | 0.009 | 0.997 | 5.000 | 0.281 | 0.086 | 0.010 | 0.996 | 4.000 |
| SPNet‡ | 0.143 | 0.022 | 0.003 | 0.999 | 1.500 | 0.182 | 0.028 | 0.003 | 0.999 | 1.750 | 0.383 | 0.052 | 0.006 | 0.997 | 3.250 |
| PromptDA | 0.817 | 0.307 | 0.038 | 0.966 | 7.000 | 0.837 | 0.311 | 0.039 | 0.968 | 7.000 | 0.956 | 0.344 | 0.043 | 0.966 | 7.000 |
| WorldMirror† | 0.773 | 0.415 | 0.055 | 0.962 | 8.000 | 0.761 | 0.418 | 0.056 | 0.953 | 8.750 | 0.699 | 0.386 | 0.053 | 0.953 | 8.750 |
| MapAnything | 2.116 | 0.593 | 0.067 | 0.934 | 11.000 | 2.243 | 0.661 | 0.074 | 0.927 | 11.000 | 2.451 | 0.817 | 0.094 | 0.911 | 11.250 |
| Pow3R† | 0.644 | 0.391 | 0.056 | 0.955 | 8.000 | 0.650 | 0.383 | 0.054 | 0.958 | 7.750 | 0.634 | 0.346 | 0.051 | 0.963 | 7.000 |
| LDCM (Ours) | 0.146 | 0.025 | 0.003 | 0.999 | 2.000 | 0.168 | 0.027 | 0.003 | 0.999 | 1.250 | 0.225 | 0.037 | 0.004 | 0.998 | 1.000 |

| Method | SIFT | | | | | ORB | | | | | Average | | | | |
|---|---|---|---|---|---|---|---|---|---|---|---|---|---|---|---|
| | RMSE↓ | MAE↓ | REL↓ | $\delta_1$↑ | Rk.↓ | RMSE↓ | MAE↓ | REL↓ | $\delta_1$↑ | Rk.↓ | RMSE↓ | MAE↓ | REL↓ | $\delta_1$↑ | Rk.↓ |
| DepthPro | 5.567 | 4.454 | 0.411 | 0.248 | 15.500 | 5.567 | 4.454 | 0.411 | 0.248 | 15.500 | 5.567 | 4.454 | 0.411 | 0.248 | 15.500 |
| UniDepth V1 | 4.414 | 4.009 | 0.463 | 0.160 | 15.250 | 4.414 | 4.009 | 0.463 | 0.160 | 15.500 | 4.414 | 4.009 | 0.463 | 0.160 | 15.455 |
| UniDepth V2 | 2.600 | 1.775 | 0.230 | 0.652 | 13.250 | 2.600 | 1.775 | 0.230 | 0.652 | 13.500 | 2.600 | 1.775 | 0.230 | 0.652 | 13.500 |
| DepthAnythingV2† | 2.613 | 0.597 | 0.060 | 0.964 | 8.250 | 2.415 | 0.603 | 0.062 | 0.954 | 7.500 | 3.325 | 0.640 | 0.059 | 0.969 | 9.227 |
| VGGT† | 0.727 | 0.486 | 0.080 | 0.913 | 8.500 | 0.823 | 0.582 | 0.101 | 0.897 | 7.750 | 0.684 | 0.419 | 0.064 | 0.946 | 7.682 |
| MoGe V1† | 4.949 | 0.572 | 0.106 | 0.946 | 10.500 | 3.109 | 0.622 | 0.116 | 0.894 | 11.250 | 3.089 | 0.976 | 0.131 | 0.895 | 11.500 |
| MoGe V2 | 1.152 | 0.819 | 0.111 | 0.893 | 11.500 | 1.152 | 0.819 | 0.111 | 0.893 | 10.000 | 1.152 | 0.819 | 0.111 | 0.893 | 11.023 |
| G2-MonoDepth‡ | 0.924 | 0.398 | 0.077 | 0.918 | 8.000 | 0.885 | 0.404 | 0.077 | 0.920 | 6.250 | 0.626 | 0.261 | 0.042 | 0.957 | 5.614 |
| OMNI-DC | 0.487 | 0.177 | 0.026 | 0.978 | 2.750 | 0.594 | 0.232 | 0.033 | 0.968 | 3.250 | 0.418 | 0.151 | 0.019 | 0.982 | 2.932 |
| PriorDA | 0.421 | 0.172 | 0.026 | 0.980 | 2.000 | 0.510 | 0.218 | 0.033 | 0.975 | 2.000 | 0.404 | 0.161 | 0.022 | 0.983 | 3.750 |
| SPNet‡ | 0.826 | 0.221 | 0.034 | 0.974 | 4.750 | 0.761 | 0.231 | 0.037 | 0.971 | 3.750 | 0.639 | 0.185 | 0.027 | 0.978 | 3.955 |
| PromptDA | 1.076 | 0.492 | 0.062 | 0.938 | 7.750 | 1.184 | 0.583 | 0.076 | 0.905 | 7.750 | 1.037 | 0.436 | 0.053 | 0.950 | 7.568 |
| WorldMirror† | 0.730 | 0.460 | 0.072 | 0.920 | 7.500 | 1.491 | 0.935 | 0.123 | 0.836 | 11.750 | 0.769 | 0.446 | 0.063 | 0.939 | 7.727 |
| MapAnything | 2.796 | 1.145 | 0.143 | 0.869 | 13.000 | 2.617 | 0.989 | 0.124 | 0.884 | 12.750 | 2.615 | 0.979 | 0.115 | 0.893 | 12.045 |
| Pow3R† | 0.686 | 0.414 | 0.068 | 0.933 | 6.250 | 0.727 | 0.474 | 0.080 | 0.906 | 6.250 | 0.660 | 0.381 | 0.057 | 0.950 | 6.545 |
| LDCM (Ours) | 0.316 | 0.089 | 0.012 | 0.995 | 1.000 | 0.333 | 0.101 | 0.013 | 0.992 | 1.000 | 0.274 | 0.070 | 0.009 | 0.996 | 1.182 |

Table 18: **Quantitative comparison of point map estimation with baseline methods on the KITTI dataset Geiger et al. (2012); Uhrig et al. (2017).** Methods marked with ‡ use scenario-specific configurations for indoor and outdoor scenes, respectively. The best and the second-best results are highlighted.

| Method | Lidar-64-Lines | | | | | Lidar-32-Lines | | | | | Lidar-16-Lines | | | | |
|---|---|---|---|---|---|---|---|---|---|---|---|---|---|---|---|
| | MAE$^p$↓ | RMSE$^p$↓ | REL$^p$↓ | $\delta_1^p$↑ | Rk.↓ | MAE$^p$↓ | RMSE$^p$↓ | REL$^p$↓ | $\delta_1^p$↑ | Rk.↓ | MAE$^p$↓ | RMSE$^p$↓ | REL$^p$↓ | $\delta_1^p$↑ | Rk.↓ |
| UniDepth V1 | 2.207 | 3.540 | 0.120 | 0.954 | 7.750 | 2.207 | 3.540 | 0.120 | 0.954 | 7.750 | 2.207 | 3.540 | 0.120 | 0.954 | 7.750 |
| UniDepth V2 | 1.813 | 3.540 | 0.096 | 0.961 | 7.000 | 1.813 | 3.540 | 0.096 | 0.961 | 7.000 | 1.813 | 3.540 | 0.096 | 0.961 | 7.000 |
| MoGe V2 | 3.536 | 4.899 | 0.208 | 0.484 | 9.000 | 3.536 | 4.899 | 0.208 | 0.484 | 9.000 | 3.536 | 4.899 | 0.208 | 0.484 | 9.000 |
| G2-MonoDepth‡ | 1.224 | 2.183 | 0.079 | 0.984 | 4.000 | 1.267 | 2.346 | 0.080 | 0.983 | 3.750 | 1.386 | 2.665 | 0.083 | 0.979 | 4.000 |
| OMNI-DC | 1.134 | 1.777 | 0.071 | 0.992 | 2.000 | 1.198 | 1.983 | 0.074 | 0.988 | 2.000 | 1.278 | 2.231 | 0.078 | 0.984 | 2.000 |
| PriorDA | 1.333 | 2.285 | 0.080 | 0.982 | 5.000 | 1.394 | 2.409 | 0.083 | 0.979 | 5.000 | 1.495 | 2.607 | 0.088 | 0.974 | 4.750 |
| SPNet‡ | 1.200 | 2.113 | 0.077 | 0.986 | 3.000 | 1.253 | 2.332 | 0.080 | 0.984 | 3.000 | 1.341 | 2.597 | 0.083 | 0.980 | 3.000 |
| PromptDA | 1.537 | 2.826 | 0.088 | 0.972 | 6.000 | 1.559 | 2.910 | 0.088 | 0.970 | 6.000 | 1.659 | 3.148 | 0.094 | 0.963 | 6.000 |
| LDCM (ours) | 0.851 | 1.656 | 0.049 | 0.993 | 1.000 | 0.881 | 1.812 | 0.051 | 0.991 | 1.000 | 0.934 | 2.017 | 0.052 | 0.989 | 1.000 |

| Method | Lidar-8-Lines | | | | | Lidar-4-Lines | | | | | 10% | | | | |
|---|---|---|---|---|---|---|---|---|---|---|---|---|---|---|---|
| | MAE$^p$↓ | RMSE$^p$↓ | REL$^p$↓ | $\delta_1^p$↑ | Rk.↓ | MAE$^p$↓ | RMSE$^p$↓ | REL$^p$↓ | $\delta_1^p$↑ | Rk.↓ | MAE$^p$↓ | RMSE$^p$↓ | REL$^p$↓ | $\delta_1^p$↑ | Rk.↓ |
| UniDepth V1 | 2.207 | 3.540 | 0.120 | 0.954 | 7.750 | 2.207 | 3.540 | 0.120 | 0.954 | 5.500 | 2.207 | 3.540 | 0.120 | 0.954 | 7.750 |
| UniDepth V2 | 1.813 | 3.540 | 0.096 | 0.961 | 6.500 | 1.813 | 3.540 | 0.096 | 0.961 | 6.750 | 1.813 | 3.540 | 0.096 | 0.961 | 6.750 |
| MoGe V2 | 3.536 | 4.899 | 0.208 | 0.484 | 9.000 | 3.536 | 4.899 | 0.208 | 0.484 | 9.000 | 3.536 | 4.899 | 0.208 | 0.484 | 9.000 |
| G2-MonoDepth‡ | 1.599 | 3.125 | 0.092 | 0.969 | 4.250 | 2.195 | 4.272 | 0.116 | 0.922 | 7.000 | 1.283 | 2.366 | 0.079 | 0.983 | 3.500 |
| OMNI-DC | 1.408 | 2.592 | 0.083 | 0.978 | 2.000 | 1.821 | 3.760 | 0.096 | 0.953 | 3.750 | 1.274 | 2.298 | 0.079 | 0.983 | 2.750 |
| PriorDA | 1.623 | 2.867 | 0.094 | 0.966 | 4.750 | 1.936 | 3.706 | 0.103 | 0.945 | 5.000 | 1.398 | 2.472 | 0.083 | 0.978 | 5.000 |
| SPNet‡ | 1.465 | 2.857 | 0.086 | 0.977 | 3.000 | 1.878 | 3.946 | 0.097 | 0.955 | 4.250 | 1.256 | 2.309 | 0.078 | 0.985 | 2.250 |
| PromptDA | 1.744 | 3.413 | 0.096 | 0.956 | 6.250 | 2.187 | 4.467 | 0.118 | 0.932 | 7.000 | 1.660 | 3.133 | 0.094 | 0.960 | 6.250 |
| LDCM (ours) | 1.022 | 2.301 | 0.054 | 0.987 | 1.000 | 1.309 | 3.212 | 0.062 | 0.974 | 1.000 | 0.917 | 1.953 | 0.052 | 0.990 | 1.000 |

| Method | 5% | | | | | 3% | | | | | 1% | | | | |
|---|---|---|---|---|---|---|---|---|---|---|---|---|---|---|---|
| | MAE$^p$↓ | RMSE$^p$↓ | REL$^p$↓ | $\delta_1^p$↑ | Rk.↓ | MAE$^p$↓ | RMSE$^p$↓ | REL$^p$↓ | $\delta_1^p$↑ | Rk.↓ | MAE$^p$↓ | RMSE$^p$↓ | REL$^p$↓ | $\delta_1^p$↑ | Rk.↓ |
| UniDepth V1 | 2.207 | 3.540 | 0.120 | 0.954 | 7.750 | 2.207 | 3.540 | 0.120 | 0.954 | 7.250 | 2.207 | 3.540 | 0.120 | 0.954 | 7.250 |
| UniDepth V2 | 1.813 | 3.540 | 0.096 | 0.961 | 6.500 | 1.813 | 3.540 | 0.096 | 0.961 | 6.500 | 1.813 | 3.540 | 0.096 | 0.961 | 6.000 |
| MoGe V2 | 3.536 | 4.899 | 0.208 | 0.484 | 9.000 | 3.536 | 4.899 | 0.208 | 0.484 | 9.000 | 3.536 | 4.899 | 0.208 | 0.484 | 9.000 |
| G2-MonoDepth‡ | 1.333 | 2.519 | 0.080 | 0.982 | 3.500 | 1.401 | 2.712 | 0.082 | 0.978 | 3.750 | 1.679 | 3.352 | 0.092 | 0.963 | 4.750 |
| OMNI-DC | 1.329 | 2.464 | 0.081 | 0.981 | 3.500 | 1.388 | 2.638 | 0.083 | 0.978 | 3.500 | 1.603 | 3.196 | 0.092 | 0.965 | 4.000 |
| PriorDA | 1.436 | 2.575 | 0.084 | 0.976 | 5.000 | 1.476 | 2.677 | 0.085 | 0.974 | 4.750 | 1.594 | 2.998 | 0.090 | 0.967 | 2.750 |
| SPNet‡ | 1.304 | 2.454 | 0.079 | 0.984 | 2.000 | 1.353 | 2.596 | 0.081 | 0.982 | 2.000 | 1.512 | 3.015 | 0.086 | 0.975 | 2.250 |
| PromptDA | 1.779 | 3.312 | 0.102 | 0.956 | 6.500 | 1.806 | 3.424 | 0.102 | 0.953 | 6.750 | 2.082 | 4.041 | 0.117 | 0.934 | 7.500 |
| LDCM (ours) | 0.958 | 2.088 | 0.053 | 0.988 | 1.000 | 0.881 | 2.129 | 0.047 | 0.987 | 1.000 | 1.007 | 2.483 | 0.051 | 0.983 | 1.000 |

| Method | SIFT | | | | | ORB | | | | | Average | | | | |
|---|---|---|---|---|---|---|---|---|---|---|---|---|---|---|---|
| | MAE$^p$↓ | RMSE$^p$↓ | REL$^p$↓ | $\delta_1^p$↑ | Rk.↓ | MAE$^p$↓ | RMSE$^p$↓ | REL$^p$↓ | $\delta_1^p$↑ | Rk.↓ | MAE$^p$↓ | RMSE$^p$↓ | REL$^p$↓ | $\delta_1^p$↑ | Rk.↓ |
| UniDepth V1 | 2.207 | 3.540 | 0.120 | 0.954 | 3.750 | 2.207 | 3.540 | 0.120 | 0.954 | 4.000 | 2.207 | 3.540 | 0.120 | 0.954 | 6.773 |
| UniDepth V2 | 1.813 | 3.540 | 0.096 | 0.961 | 2.000 | 1.813 | 3.540 | 0.096 | 0.961 | 2.250 | 1.813 | 3.540 | 0.096 | 0.961 | 5.409 |
| MoGe V2 | 3.536 | 4.899 | 0.208 | 0.484 | 9.000 | 3.536 | 4.899 | 0.208 | 0.484 | 9.000 | 3.536 | 4.899 | 0.208 | 0.484 | 9.000 |
| G2-MonoDepth‡ | 2.736 | 4.677 | 0.158 | 0.795 | 8.000 | 2.261 | 4.085 | 0.133 | 0.864 | 6.750 | 1.669 | 3.118 | 0.098 | 0.946 | 4.841 |
| OMNI-DC | 2.316 | 4.173 | 0.139 | 0.875 | 6.000 | 2.210 | 4.001 | 0.136 | 0.882 | 6.250 | 1.542 | 2.828 | 0.092 | 0.960 | 3.409 |
| PriorDA | 1.830 | 3.358 | 0.103 | 0.938 | 3.000 | 1.792 | 3.245 | 0.104 | 0.935 | 2.750 | 1.573 | 2.836 | 0.091 | 0.965 | 4.341 |
| SPNet‡ | 2.096 | 3.845 | 0.120 | 0.888 | 4.500 | 1.921 | 3.631 | 0.113 | 0.908 | 4.500 | 1.507 | 2.881 | 0.089 | 0.964 | 3.068 |
| PromptDA | 2.609 | 4.581 | 0.154 | 0.868 | 7.000 | 2.642 | 4.475 | 0.159 | 0.852 | 8.000 | 1.933 | 3.612 | 0.110 | 0.938 | 6.659 |
| LDCM (ours) | 1.290 | 2.918 | 0.064 | 0.959 | 1.250 | 1.247 | 2.820 | 0.065 | 0.956 | 1.250 | 1.027 | 2.308 | 0.055 | 0.982 | 1.045 |

Table 19: **Quantitative comparison of point map estimation with baseline methods on the indoor scenes of the DIODE dataset** Vasiljevic et al. (2019). Methods marked with ‡ use scenario-specific configurations for indoor and outdoor scenes, respectively. The best and the second-best results are highlighted.

| Method | 10% Noise | | | | | 5% | | | | | 3% | | | | |
|---|---|---|---|---|---|---|---|---|---|---|---|---|---|---|---|
| | $MAE^p\downarrow$ | $RMSE^p\downarrow$ | $REL^p\downarrow$ | $\delta_1^p\uparrow$ | Rk.$\downarrow$ | $MAE^p\downarrow$ | $RMSE^p\downarrow$ | $REL^p\downarrow$ | $\delta_1^p\uparrow$ | Rk.$\downarrow$ | $MAE^p\downarrow$ | $RMSE^p\downarrow$ | $REL^p\downarrow$ | $\delta_1^p\uparrow$ | Rk.$\downarrow$ |
| UniDepth V1 | 0.911 | 1.017 | 0.159 | 0.779 | 7.500 | 0.911 | 1.017 | 0.159 | 0.779 | 7.500 | 0.911 | 1.017 | 0.159 | 0.779 | 7.500 |
| UniDepth V2 | 0.730 | 0.872 | 0.164 | 0.694 | 7.500 | 0.730 | 0.872 | 0.164 | 0.694 | 7.500 | 0.730 | 0.872 | 0.164 | 0.694 | 7.500 |
| MoGe V2 | 1.048 | 1.185 | 0.242 | 0.410 | 9.000 | 1.048 | 1.185 | 0.242 | 0.410 | 9.000 | 1.048 | 1.185 | 0.242 | 0.410 | 9.000 |
| G2-MonoDepth‡ | 0.118 | 0.136 | 0.027 | 1.000 | 1.750 | 0.118 | 0.137 | 0.027 | 1.000 | 2.250 | 0.118 | 0.140 | 0.028 | 1.000 | 2.750 |
| OMNI-DC | 0.122 | 0.163 | 0.028 | 0.999 | 4.000 | 0.118 | 0.137 | 0.027 | 1.000 | 2.250 | 0.118 | 0.139 | 0.027 | 1.000 | 2.000 |
| PriorDA | 0.132 | 0.179 | 0.029 | 0.999 | 4.750 | 0.123 | 0.155 | 0.028 | 0.999 | 5.000 | 0.123 | 0.156 | 0.028 | 0.999 | 4.750 |
| SPNet‡ | 0.118 | 0.138 | 0.027 | 1.000 | 2.000 | 0.117 | 0.135 | 0.027 | 1.000 | 1.750 | 0.118 | 0.136 | 0.027 | 1.000 | 1.750 |
| PromptDA | 0.138 | 0.189 | 0.033 | 0.997 | 6.000 | 0.133 | 0.180 | 0.032 | 0.996 | 6.000 | 0.133 | 0.183 | 0.031 | 0.997 | 6.000 |
| LDCM (ours) | **0.107** | **0.127** | **0.021** | **1.000** | **1.000** | **0.107** | **0.128** | **0.021** | **1.000** | **1.000** | **0.107** | **0.130** | **0.021** | **1.000** | **1.000** |

| Method | 1% | | | | | 500 | | | | | 100 | | | | |
|---|---|---|---|---|---|---|---|---|---|---|---|---|---|---|---|
| | $MAE^p\downarrow$ | $RMSE^p\downarrow$ | $REL^p\downarrow$ | $\delta_1^p\uparrow$ | Rk.$\downarrow$ | $MAE^p\downarrow$ | $RMSE^p\downarrow$ | $REL^p\downarrow$ | $\delta_1^p\uparrow$ | Rk.$\downarrow$ | $MAE^p\downarrow$ | $RMSE^p\downarrow$ | $REL^p\downarrow$ | $\delta_1^p\uparrow$ | Rk.$\downarrow$ |
| UniDepth V1 | 0.911 | 1.017 | 0.159 | 0.779 | 7.500 | 0.911 | 1.017 | 0.159 | 0.779 | 7.500 | 0.911 | 1.017 | 0.159 | 0.779 | 7.000 |
| UniDepth V2 | 0.730 | 0.872 | 0.164 | 0.694 | 7.500 | 0.730 | 0.872 | 0.164 | 0.694 | 7.500 | 0.730 | 0.872 | 0.164 | 0.694 | 7.000 |
| MoGe V2 | 1.048 | 1.185 | 0.242 | 0.410 | 9.000 | 1.048 | 1.185 | 0.242 | 0.410 | 9.000 | 1.048 | 1.185 | 0.242 | 0.410 | 9.000 |
| G2-MonoDepth‡ | 0.120 | 0.149 | 0.028 | 0.999 | 3.250 | 0.144 | 0.220 | 0.034 | 0.996 | 5.000 | 0.651 | 0.843 | 0.191 | 0.598 | 7.000 |
| OMNI-DC | 0.118 | 0.145 | 0.028 | 0.999 | 2.500 | 0.127 | 0.184 | 0.029 | 0.998 | 2.500 | 0.161 | 0.275 | 0.036 | 0.991 | 4.000 |
| PriorDA | 0.123 | 0.159 | 0.028 | 0.999 | 3.750 | 0.128 | 0.179 | 0.029 | 0.999 | 2.250 | 0.140 | 0.217 | 0.031 | 0.996 | 1.500 |
| SPNet‡ | 0.118 | 0.140 | 0.027 | 0.999 | 2.000 | 0.127 | 0.187 | 0.029 | 0.998 | 2.750 | 0.156 | 0.268 | 0.035 | 0.992 | 3.000 |
| PromptDA | 0.136 | 0.191 | 0.032 | 0.995 | 6.000 | 0.167 | 0.263 | 0.042 | 0.989 | 6.000 | 0.228 | 0.428 | 0.061 | 0.978 | 5.000 |
| LDCM (ours) | **0.107** | **0.134** | **0.021** | **1.000** | **1.000** | **0.111** | **0.167** | **0.022** | **0.999** | **1.000** | **0.128** | **0.235** | **0.024** | **0.996** | **1.250** |

| Method | SIFT | | | | | ORB | | | | | Virtual-Lidar-32-Lines | | | | |
|---|---|---|---|---|---|---|---|---|---|---|---|---|---|---|---|
| | $MAE^p\downarrow$ | $RMSE^p\downarrow$ | $REL^p\downarrow$ | $\delta_1^p\uparrow$ | Rk.$\downarrow$ | $MAE^p\downarrow$ | $RMSE^p\downarrow$ | $REL^p\downarrow$ | $\delta_1^p\uparrow$ | Rk.$\downarrow$ | $MAE^p\downarrow$ | $RMSE^p\downarrow$ | $REL^p\downarrow$ | $\delta_1^p\uparrow$ | Rk.$\downarrow$ |
| UniDepth V1 | 0.911 | 1.017 | 0.159 | 0.779 | 7.000 | 0.911 | 1.017 | 0.159 | 0.779 | 7.500 | 0.911 | 1.017 | 0.159 | 0.779 | 7.500 |
| UniDepth V2 | 0.730 | 0.872 | 0.164 | 0.694 | 6.500 | 0.730 | 0.872 | 0.164 | 0.694 | 6.500 | 0.730 | 0.872 | 0.164 | 0.694 | 7.500 |
| MoGe V2 | 1.048 | 1.185 | 0.242 | 0.410 | 8.750 | 1.048 | 1.185 | 0.242 | 0.410 | 9.000 | 1.048 | 1.185 | 0.242 | 0.410 | 9.000 |
| G2-MonoDepth‡ | 0.777 | 1.000 | 0.251 | 0.545 | 7.750 | 0.866 | 1.111 | 0.270 | 0.504 | 8.000 | 0.127 | 0.178 | 0.029 | 0.998 | 4.750 |
| OMNI-DC | 0.313 | 0.429 | 0.099 | 0.867 | 3.250 | 0.448 | 0.616 | 0.135 | 0.790 | 4.250 | 0.121 | 0.158 | 0.028 | 0.999 | 1.750 |
| PriorDA | 0.174 | 0.232 | 0.050 | 0.975 | 1.500 | 0.205 | 0.280 | 0.066 | 0.967 | 1.500 | 0.124 | 0.164 | 0.029 | 0.999 | 3.250 |
| SPNet‡ | 0.350 | 0.478 | 0.117 | 0.818 | 5.000 | 0.398 | 0.554 | 0.132 | 0.794 | 3.000 | 0.121 | 0.163 | 0.028 | 0.999 | 2.000 |
| PromptDA | 0.336 | 0.448 | 0.108 | 0.874 | 3.750 | 0.511 | 0.708 | 0.160 | 0.791 | 5.000 | 0.144 | 0.211 | 0.036 | 0.993 | 6.000 |
| LDCM (ours) | **0.172** | **0.243** | **0.045** | **0.964** | **1.500** | **0.201** | **0.300** | **0.053** | **0.952** | **1.500** | **0.108** | **0.144** | **0.021** | **0.999** | **1.000** |

| Method | Virtual-Lidar-16-Lines | | | | | Virtual-Lidar-8-Lines | | | | | Average | | | | |
|---|---|---|---|---|---|---|---|---|---|---|---|---|---|---|---|
| | $MAE^p\downarrow$ | $RMSE^p\downarrow$ | $REL^p\downarrow$ | $\delta_1^p\uparrow$ | Rk.$\downarrow$ | $MAE^p\downarrow$ | $RMSE^p\downarrow$ | $REL^p\downarrow$ | $\delta_1^p\uparrow$ | Rk.$\downarrow$ | $MAE^p\downarrow$ | $RMSE^p\downarrow$ | $REL^p\downarrow$ | $\delta_1^p\uparrow$ | Rk.$\downarrow$ |
| UniDepth V1 | 0.911 | 1.017 | 0.159 | 0.779 | 7.500 | 0.911 | 1.017 | 0.159 | 0.779 | 7.500 | 0.911 | 1.017 | 0.159 | 0.779 | 7.318 |
| UniDepth V2 | 0.730 | 0.872 | 0.164 | 0.694 | 7.500 | 0.730 | 0.872 | 0.164 | 0.694 | 7.500 | 0.730 | 0.872 | 0.164 | 0.694 | 7.273 |
| MoGe V2 | 1.048 | 1.185 | 0.242 | 0.410 | 9.000 | 1.048 | 1.185 | 0.242 | 0.410 | 9.000 | 1.048 | 1.185 | 0.242 | 0.410 | 8.955 |
| G2-MonoDepth‡ | 0.136 | 0.211 | 0.031 | 0.997 | 5.000 | 0.175 | 0.285 | 0.042 | 0.987 | 5.750 | 0.305 | 0.401 | 0.087 | 0.875 | 4.841 |
| OMNI-DC | 0.126 | 0.176 | 0.029 | 0.998 | 2.250 | 0.145 | 0.229 | 0.034 | 0.994 | 4.000 | 0.174 | 0.241 | 0.045 | 0.967 | 2.977 |
| PriorDA | 0.127 | 0.177 | 0.029 | 0.999 | 2.500 | 0.136 | 0.197 | 0.031 | 0.998 | 1.250 | 0.140 | 0.190 | 0.034 | 0.994 | 2.909 |
| SPNet‡ | 0.126 | 0.182 | 0.029 | 0.998 | 2.750 | 0.139 | 0.220 | 0.032 | 0.996 | 3.000 | 0.172 | 0.236 | 0.046 | 0.963 | 2.636 |
| PromptDA | 0.150 | 0.232 | 0.036 | 0.993 | 6.000 | 0.170 | 0.275 | 0.041 | 0.987 | 5.000 | 0.204 | 0.301 | 0.056 | 0.963 | 5.523 |
| LDCM (ours) | **0.110** | **0.155** | **0.021** | **0.999** | **1.000** | **0.137** | **0.202** | **0.028** | **0.998** | **1.500** | **0.127** | **0.179** | **0.027** | **0.992** | **1.159** |

Table 20: **Quantitative comparison of point map estimation with baseline methods on the outdoor scenes of the DIODE dataset** Vasiljevic et al. (2019). Methods marked with ‡ use scenario-specific configurations for indoor and outdoor scenes, respectively. The best and the second-best results are highlighted.

| Method | Virtual-Lidar-64-Lines | | | | | Virtual-Lidar-32-Lines | | | | | Virtual-Lidar-16-Lines | | | | |
|---|---|---|---|---|---|---|---|---|---|---|---|---|---|---|---|
| | $MAE^p \downarrow$ | $RMSE^p \downarrow$ | $REL^p \downarrow$ | $\delta_1^p \uparrow$ | Rk.$\downarrow$ | $MAE^p \downarrow$ | $RMSE^p \downarrow$ | $REL^p \downarrow$ | $\delta_1^p \uparrow$ | Rk.$\downarrow$ | $MAE^p \downarrow$ | $RMSE^p \downarrow$ | $REL^p \downarrow$ | $\delta_1^p \uparrow$ | Rk.$\downarrow$ |
| UniDepth V1 | 4.280 | 6.372 | 0.196 | 0.644 | 7.500 | 4.280 | 6.372 | 0.196 | 0.644 | 7.500 | 4.280 | 6.372 | 0.196 | 0.644 | 7.500 |
| UniDepth V2 | 9.686 | 12.049 | 0.521 | 0.505 | 9.000 | 9.686 | 12.049 | 0.521 | 0.505 | 9.000 | 9.686 | 12.049 | 0.521 | 0.505 | 9.000 |
| MoGe V2 | 4.041 | 5.505 | 0.205 | 0.626 | 7.500 | 4.041 | 5.505 | 0.205 | 0.626 | 7.500 | 4.041 | 5.505 | 0.205 | 0.626 | 7.500 |
| G2-MonoDepth | 1.867 | 3.009 | 0.105 | 0.962 | 4.250 | 1.983 | 3.309 | 0.116 | 0.953 | 4.500 | 2.188 | 3.726 | 0.130 | 0.933 | 5.000 |
| OMNI-DC | 1.838 | 2.970 | 0.101 | 0.965 | 3.000 | 1.927 | 3.230 | 0.108 | 0.958 | 3.500 | 2.102 | 3.673 | 0.120 | 0.946 | 3.750 |
| PriorDA | 1.980 | 3.071 | 0.104 | 0.953 | 4.750 | 2.018 | 3.184 | 0.106 | 0.950 | 4.000 | 2.087 | 3.349 | 0.111 | 0.944 | 2.750 |
| SPNet | 1.806 | 2.857 | 0.099 | 0.966 | 2.000 | 1.885 | 3.113 | 0.105 | 0.960 | 2.000 | 2.052 | 3.547 | 0.117 | 0.950 | 2.500 |
| PromptDA | 2.392 | 4.186 | 0.123 | 0.923 | 6.000 | 2.484 | 4.427 | 0.128 | 0.916 | 6.000 | 2.607 | 4.638 | 0.133 | 0.907 | 6.000 |
| LDCM (ours) | 1.508 | 2.581 | 0.084 | 0.977 | 1.000 | 1.563 | 2.755 | 0.088 | 0.973 | 1.000 | 1.667 | 3.022 | 0.093 | 0.965 | 1.000 |

| Method | Virtual-Lidar-8-Lines | | | | | Virtual-Lidar-4-Lines | | | | | 10% Noise | | | | |
|---|---|---|---|---|---|---|---|---|---|---|---|---|---|---|---|
| | $MAE^p \downarrow$ | $RMSE^p \downarrow$ | $REL^p \downarrow$ | $\delta_1^p \uparrow$ | Rk.$\downarrow$ | $MAE^p \downarrow$ | $RMSE^p \downarrow$ | $REL^p \downarrow$ | $\delta_1^p \uparrow$ | Rk.$\downarrow$ | $MAE^p \downarrow$ | $RMSE^p \downarrow$ | $REL^p \downarrow$ | $\delta_1^p \uparrow$ | Rk.$\downarrow$ |
| UniDepth V1 | 4.280 | 6.372 | 0.196 | 0.644 | 7.500 | 4.280 | 6.372 | 0.196 | 0.644 | 6.250 | 4.280 | 6.372 | 0.196 | 0.644 | 7.500 |
| UniDepth V2 | 9.686 | 12.049 | 0.521 | 0.505 | 9.000 | 9.686 | 12.049 | 0.521 | 0.505 | 9.000 | 9.686 | 12.049 | 0.521 | 0.505 | 9.000 |
| MoGe V2 | 4.041 | 5.505 | 0.205 | 0.626 | 7.500 | 4.041 | 5.505 | 0.205 | 0.626 | 5.750 | 4.041 | 5.505 | 0.205 | 0.626 | 7.500 |
| G2-MonoDepth | 2.659 | 4.565 | 0.152 | 0.887 | 5.250 | 4.215 | 6.818 | 0.237 | 0.658 | 7.000 | 1.645 | 2.195 | 0.087 | 0.980 | 2.750 |
| OMNI-DC | 2.503 | 4.513 | 0.133 | 0.916 | 3.750 | 3.689 | 6.325 | 0.200 | 0.769 | 4.500 | 1.661 | 2.301 | 0.087 | 0.980 | 3.500 |
| PriorDA | 2.272 | 3.798 | 0.118 | 0.929 | 2.000 | 2.921 | 5.112 | 0.155 | 0.857 | 2.000 | 2.035 | 3.117 | 0.102 | 0.951 | 5.000 |
| SPNet | 2.359 | 4.143 | 0.134 | 0.923 | 3.250 | 3.125 | 5.372 | 0.194 | 0.827 | 3.000 | 1.638 | 2.207 | 0.085 | 0.981 | 2.250 |
| PromptDA | 2.896 | 5.216 | 0.144 | 0.886 | 5.750 | 4.086 | 6.913 | 0.208 | 0.762 | 6.500 | 2.385 | 3.954 | 0.119 | 0.920 | 6.000 |
| LDCM (ours) | 1.850 | 3.456 | 0.099 | 0.953 | 1.000 | 2.313 | 4.361 | 0.119 | 0.905 | 1.000 | 1.345 | 2.032 | 0.076 | 0.991 | 1.000 |

| Method | 5% | | | | | 3% | | | | | 1% | | | | |
|---|---|---|---|---|---|---|---|---|---|---|---|---|---|---|---|
| | $MAE^p \downarrow$ | $RMSE^p \downarrow$ | $REL^p \downarrow$ | $\delta_1^p \uparrow$ | Rk.$\downarrow$ | $MAE^p \downarrow$ | $RMSE^p \downarrow$ | $REL^p \downarrow$ | $\delta_1^p \uparrow$ | Rk.$\downarrow$ | $MAE^p \downarrow$ | $RMSE^p \downarrow$ | $REL^p \downarrow$ | $\delta_1^p \uparrow$ | Rk.$\downarrow$ |
| UniDepth V1 | 4.280 | 6.372 | 0.196 | 0.644 | 7.500 | 4.280 | 6.372 | 0.196 | 0.644 | 7.500 | 4.280 | 6.372 | 0.196 | 0.644 | 7.500 |
| UniDepth V2 | 9.686 | 12.049 | 0.521 | 0.505 | 9.000 | 9.686 | 12.049 | 0.521 | 0.505 | 9.000 | 9.686 | 12.049 | 0.521 | 0.505 | 9.000 |
| MoGe V2 | 4.041 | 5.505 | 0.205 | 0.626 | 7.500 | 4.041 | 5.505 | 0.205 | 0.626 | 7.500 | 4.041 | 5.505 | 0.205 | 0.626 | 7.500 |
| G2-MonoDepth | 1.671 | 2.313 | 0.089 | 0.977 | 3.750 | 1.702 | 2.437 | 0.092 | 0.975 | 3.750 | 1.776 | 2.796 | 0.099 | 0.967 | 4.000 |
| OMNI-DC | 1.670 | 2.335 | 0.088 | 0.978 | 3.250 | 1.698 | 2.450 | 0.091 | 0.976 | 3.250 | 1.777 | 2.754 | 0.096 | 0.969 | 3.000 |
| PriorDA | 1.950 | 2.933 | 0.101 | 0.955 | 5.000 | 1.951 | 2.947 | 0.101 | 0.955 | 5.000 | 1.973 | 3.029 | 0.103 | 0.953 | 5.000 |
| SPNet | 1.655 | 2.265 | 0.087 | 0.979 | 2.000 | 1.679 | 2.362 | 0.089 | 0.977 | 2.000 | 1.750 | 2.650 | 0.095 | 0.971 | 2.000 |
| PromptDA | 2.336 | 3.900 | 0.119 | 0.923 | 6.000 | 2.356 | 3.962 | 0.120 | 0.923 | 6.000 | 2.444 | 4.183 | 0.127 | 0.919 | 6.000 |
| LDCM (ours) | 1.366 | 2.098 | 0.077 | 0.988 | 1.000 | 1.394 | 2.202 | 0.079 | 0.986 | 1.000 | 1.462 | 2.436 | 0.082 | 0.980 | 1.000 |

| Method | SIFT | | | | | ORB | | | | | Average | | | | |
|---|---|---|---|---|---|---|---|---|---|---|---|---|---|---|---|
| | $MAE^p \downarrow$ | $RMSE^p \downarrow$ | $REL^p \downarrow$ | $\delta_1^p \uparrow$ | Rk.$\downarrow$ | $MAE^p \downarrow$ | $RMSE^p \downarrow$ | $REL^p \downarrow$ | $\delta_1^p \uparrow$ | Rk.$\downarrow$ | $MAE^p \downarrow$ | $RMSE^p \downarrow$ | $REL^p \downarrow$ | $\delta_1^p \uparrow$ | Rk.$\downarrow$ |
| UniDepth V1 | 4.280 | 6.372 | 0.196 | 0.644 | 7.500 | 4.280 | 6.372 | 0.196 | 0.644 | 7.500 | 4.280 | 6.372 | 0.196 | 0.644 | 7.386 |
| UniDepth V2 | 9.686 | 12.049 | 0.521 | 0.505 | 9.000 | 9.686 | 12.049 | 0.521 | 0.505 | 9.000 | 9.686 | 12.049 | 0.521 | 0.505 | 9.000 |
| MoGe V2 | 4.041 | 5.505 | 0.205 | 0.626 | 7.500 | 4.041 | 5.505 | 0.205 | 0.626 | 7.500 | 4.041 | 5.505 | 0.205 | 0.626 | 7.341 |
| G2-MonoDepth | 2.120 | 3.399 | 0.121 | 0.924 | 5.000 | 2.291 | 3.734 | 0.126 | 0.903 | 5.000 | 2.194 | 3.482 | 0.123 | 0.920 | 4.568 |
| OMNI-DC | 1.971 | 3.284 | 0.104 | 0.952 | 3.000 | 2.146 | 3.692 | 0.110 | 0.933 | 3.000 | 2.089 | 3.412 | 0.113 | 0.940 | 3.409 |
| PriorDA | 2.078 | 3.326 | 0.106 | 0.944 | 4.000 | 2.232 | 3.698 | 0.113 | 0.927 | 4.000 | 2.136 | 3.415 | 0.111 | 0.938 | 3.955 |
| SPNet | 1.883 | 2.993 | 0.102 | 0.956 | 2.000 | 1.962 | 3.215 | 0.104 | 0.946 | 2.000 | 1.981 | 3.157 | 0.110 | 0.949 | 2.273 |
| PromptDA | 2.699 | 4.737 | 0.132 | 0.894 | 6.000 | 3.043 | 5.181 | 0.150 | 0.856 | 6.000 | 2.703 | 4.663 | 0.137 | 0.894 | 6.023 |
| LDCM (ours) | 1.551 | 2.734 | 0.086 | 0.974 | 1.000 | 1.642 | 2.996 | 0.090 | 0.962 | 1.000 | 1.606 | 2.788 | 0.088 | 0.969 | 1.000 |

Table 21: **Quantitative comparison of point map estimation with baseline methods on the iBims-1 dataset** Koch et al. (2018). Methods marked with ‡ use scenario-specific configurations for indoor and outdoor scenes, respectively. The  best  and the  second-best  results are highlighted.

| Method | 10% Noise | | | | | 5% | | | | | 3% | | | | |
|---|---|---|---|---|---|---|---|---|---|---|---|---|---|---|---|
| | $MAE^p\downarrow$ | $RMSE^p\downarrow$ | $REL^p\downarrow$ | $\delta_1^p\uparrow$ | Rk.$\downarrow$ | $MAE^p\downarrow$ | $RMSE^p\downarrow$ | $REL^p\downarrow$ | $\delta_1^p\uparrow$ | Rk.$\downarrow$ | $MAE^p\downarrow$ | $RMSE^p\downarrow$ | $REL^p\downarrow$ | $\delta_1^p\uparrow$ | Rk.$\downarrow$ |
| UniDepth V1 | 1.154 | 1.239 | 0.370 | 0.239 | 9.000 | 1.154 | 1.239 | 0.370 | 0.239 | 9.000 | 1.154 | 1.239 | 0.370 | 0.239 | 9.000 |
| UniDepth V2 | 0.365 | 0.489 | 0.107 | 0.932 | 7.000 | 0.365 | 0.489 | 0.107 | 0.932 | 7.000 | 0.365 | 0.489 | 0.107 | 0.932 | 7.000 |
| MoGe V2 | 0.574 | 0.667 | 0.156 | 0.740 | 8.000 | 0.574 | 0.667 | 0.156 | 0.740 | 8.000 | 0.574 | 0.667 | 0.156 | 0.740 | 8.000 |
| G2-MonoDepth‡ | 0.131 | 0.181 | 0.036 | 0.995 | 3.000 | 0.133 | 0.191 | 0.036 | 0.994 | 3.500 | 0.136 | 0.201 | 0.037 | 0.994 | 3.500 |
| OMNI-DC | 0.137 | 0.209 | 0.037 | 0.993 | 4.000 | 0.131 | 0.186 | 0.036 | 0.995 | 2.250 | 0.133 | 0.197 | 0.037 | 0.994 | 3.000 |
| PriorDA | 0.145 | 0.213 | 0.040 | 0.993 | 4.750 | 0.142 | 0.210 | 0.039 | 0.992 | 5.000 | 0.143 | 0.213 | 0.039 | 0.992 | 5.000 |
| SPNet‡ | 0.128 | 0.175 | 0.035 | 0.996 | 1.750 | 0.130 | 0.181 | 0.036 | 0.995 | 1.750 | 0.132 | 0.189 | 0.036 | 0.995 | 1.750 |
| PromptDA | 0.163 | 0.257 | 0.045 | 0.988 | 6.000 | 0.165 | 0.261 | 0.045 | 0.987 | 6.000 | 0.166 | 0.263 | 0.045 | 0.988 | 6.000 |
| LDCM (ours) | 0.075 | 0.151 | 0.021 | 0.996 | 1.000 | 0.076 | 0.158 | 0.022 | 0.995 | 1.000 | 0.078 | 0.164 | 0.022 | 0.995 | 1.000 |

| Method | 1% | | | | | 500 | | | | | 100 | | | | |
|---|---|---|---|---|---|---|---|---|---|---|---|---|---|---|---|
| | $MAE^p\downarrow$ | $RMSE^p\downarrow$ | $REL^p\downarrow$ | $\delta_1^p\uparrow$ | Rk.$\downarrow$ | $MAE^p\downarrow$ | $RMSE^p\downarrow$ | $REL^p\downarrow$ | $\delta_1^p\uparrow$ | Rk.$\downarrow$ | $MAE^p\downarrow$ | $RMSE^p\downarrow$ | $REL^p\downarrow$ | $\delta_1^p\uparrow$ | Rk.$\downarrow$ |
| UniDepth V1 | 1.154 | 1.239 | 0.370 | 0.239 | 9.000 | 1.154 | 1.239 | 0.370 | 0.239 | 9.000 | 1.154 | 1.239 | 0.370 | 0.239 | 9.000 |
| UniDepth V2 | 0.365 | 0.489 | 0.107 | 0.932 | 7.000 | 0.365 | 0.489 | 0.107 | 0.932 | 7.000 | 0.365 | 0.489 | 0.107 | 0.932 | 7.000 |
| MoGe V2 | 0.574 | 0.667 | 0.156 | 0.740 | 8.000 | 0.574 | 0.667 | 0.156 | 0.740 | 8.000 | 0.574 | 0.667 | 0.156 | 0.740 | 8.000 |
| G2-MonoDepth‡ | 0.143 | 0.226 | 0.039 | 0.991 | 4.250 | 0.166 | 0.274 | 0.044 | 0.985 | 5.000 | 0.249 | 0.392 | 0.069 | 0.958 | 6.000 |
| OMNI-DC | 0.138 | 0.214 | 0.038 | 0.992 | 2.500 | 0.152 | 0.247 | 0.042 | 0.988 | 3.000 | 0.187 | 0.310 | 0.051 | 0.977 | 4.000 |
| PriorDA | 0.146 | 0.221 | 0.040 | 0.991 | 4.500 | 0.153 | 0.236 | 0.042 | 0.990 | 2.750 | 0.168 | 0.281 | 0.045 | 0.987 | 2.000 |
| SPNet‡ | 0.138 | 0.213 | 0.037 | 0.992 | 2.000 | 0.151 | 0.253 | 0.041 | 0.988 | 2.750 | 0.176 | 0.311 | 0.046 | 0.982 | 3.250 |
| PromptDA | 0.172 | 0.275 | 0.046 | 0.987 | 6.000 | 0.186 | 0.300 | 0.049 | 0.985 | 5.750 | 0.216 | 0.352 | 0.057 | 0.980 | 4.750 |
| LDCM (ours) | 0.082 | 0.177 | 0.023 | 0.994 | 1.000 | 0.094 | 0.204 | 0.026 | 0.991 | 1.000 | 0.112 | 0.241 | 0.030 | 0.988 | 1.000 |

| Method | SIFT | | | | | ORB | | | | | Virtual-Lidar-32-Lines | | | | |
|---|---|---|---|---|---|---|---|---|---|---|---|---|---|---|---|
| | $MAE^p\downarrow$ | $RMSE^p\downarrow$ | $REL^p\downarrow$ | $\delta_1^p\uparrow$ | Rk.$\downarrow$ | $MAE^p\downarrow$ | $RMSE^p\downarrow$ | $REL^p\downarrow$ | $\delta_1^p\uparrow$ | Rk.$\downarrow$ | $MAE^p\downarrow$ | $RMSE^p\downarrow$ | $REL^p\downarrow$ | $\delta_1^p\uparrow$ | Rk.$\downarrow$ |
| UniDepth V1 | 1.154 | 1.239 | 0.370 | 0.239 | 9.000 | 1.154 | 1.239 | 0.370 | 0.239 | 9.000 | 1.154 | 1.239 | 0.370 | 0.239 | 9.000 |
| UniDepth V2 | 0.365 | 0.489 | 0.107 | 0.932 | 6.750 | 0.365 | 0.489 | 0.107 | 0.932 | 6.250 | 0.365 | 0.489 | 0.107 | 0.932 | 7.000 |
| MoGe V2 | 0.574 | 0.667 | 0.156 | 0.740 | 8.000 | 0.574 | 0.667 | 0.156 | 0.740 | 8.000 | 0.574 | 0.667 | 0.156 | 0.740 | 8.000 |
| G2-MonoDepth‡ | 0.273 | 0.396 | 0.082 | 0.920 | 6.250 | 0.303 | 0.432 | 0.092 | 0.900 | 6.250 | 0.154 | 0.255 | 0.041 | 0.989 | 4.000 |
| OMNI-DC | 0.210 | 0.324 | 0.059 | 0.958 | 4.000 | 0.251 | 0.375 | 0.074 | 0.925 | 4.250 | 0.143 | 0.227 | 0.039 | 0.991 | 2.750 |
| PriorDA | 0.164 | 0.242 | 0.046 | 0.988 | 2.000 | 0.184 | 0.273 | 0.053 | 0.981 | 2.000 | 0.188 | 0.270 | 0.048 | 0.982 | 5.750 |
| SPNet‡ | 0.170 | 0.259 | 0.048 | 0.985 | 3.000 | 0.192 | 0.290 | 0.056 | 0.971 | 3.000 | 0.142 | 0.226 | 0.038 | 0.991 | 2.000 |
| PromptDA | 0.258 | 0.378 | 0.072 | 0.947 | 5.000 | 0.315 | 0.453 | 0.088 | 0.903 | 5.750 | 0.174 | 0.277 | 0.047 | 0.986 | 5.250 |
| LDCM (ours) | 0.103 | 0.208 | 0.029 | 0.990 | 1.000 | 0.119 | 0.230 | 0.034 | 0.986 | 1.000 | 0.085 | 0.185 | 0.024 | 0.993 | 1.000 |

| Method | Virtual-Lidar-16-Lines | | | | | Virtual-Lidar-8-Lines | | | | | Average | | | | |
|---|---|---|---|---|---|---|---|---|---|---|---|---|---|---|---|
| | $MAE^p\downarrow$ | $RMSE^p\downarrow$ | $REL^p\downarrow$ | $\delta_1^p\uparrow$ | Rk.$\downarrow$ | $MAE^p\downarrow$ | $RMSE^p\downarrow$ | $REL^p\downarrow$ | $\delta_1^p\uparrow$ | Rk.$\downarrow$ | $MAE^p\downarrow$ | $RMSE^p\downarrow$ | $REL^p\downarrow$ | $\delta_1^p\uparrow$ | Rk.$\downarrow$ |
| UniDepth V1 | 1.154 | 1.239 | 0.370 | 0.239 | 9.000 | 1.154 | 1.239 | 0.370 | 0.239 | 9.000 | 1.154 | 1.239 | 0.370 | 0.239 | 9.000 |
| UniDepth V2 | 0.365 | 0.489 | 0.107 | 0.932 | 7.000 | 0.365 | 0.489 | 0.107 | 0.932 | 7.000 | 0.365 | 0.489 | 0.107 | 0.932 | 6.909 |
| MoGe V2 | 0.574 | 0.667 | 0.156 | 0.740 | 8.000 | 0.574 | 0.667 | 0.156 | 0.740 | 8.000 | 0.574 | 0.667 | 0.156 | 0.740 | 8.000 |
| G2-MonoDepth‡ | 0.165 | 0.277 | 0.044 | 0.986 | 5.250 | 0.197 | 0.328 | 0.052 | 0.977 | 5.750 | 0.186 | 0.287 | 0.052 | 0.972 | 4.750 |
| OMNI-DC | 0.150 | 0.240 | 0.041 | 0.989 | 2.750 | 0.173 | 0.284 | 0.048 | 0.982 | 4.250 | 0.164 | 0.256 | 0.046 | 0.980 | 3.341 |
| PriorDA | 0.152 | 0.233 | 0.041 | 0.990 | 2.750 | 0.162 | 0.248 | 0.044 | 0.988 | 2.000 | 0.159 | 0.240 | 0.043 | 0.989 | 3.500 |
| SPNet‡ | 0.150 | 0.248 | 0.040 | 0.989 | 2.750 | 0.168 | 0.280 | 0.045 | 0.985 | 3.000 | 0.152 | 0.239 | 0.042 | 0.988 | 2.455 |
| PromptDA | 0.177 | 0.282 | 0.047 | 0.987 | 5.750 | 0.193 | 0.305 | 0.052 | 0.983 | 4.750 | 0.199 | 0.309 | 0.054 | 0.975 | 5.545 |
| LDCM (ours) | 0.091 | 0.198 | 0.025 | 0.992 | 1.000 | 0.101 | 0.215 | 0.028 | 0.990 | 1.000 | 0.092 | 0.194 | 0.026 | 0.992 | 1.000 |

Table 22: **Quantitative comparison of point map estimation with baseline methods on the indoor scenes of the ETH3D dataset** Schops et al. (2017). Methods marked with ‡ use scenario-specific configurations for indoor and outdoor scenes, respectively. The best and the second-best results are highlighted.

| Method | 10% Noise | | | | | 5% | | | | | 3% | | | | |
|---|---|---|---|---|---|---|---|---|---|---|---|---|---|---|---|
| | $MAE^p\downarrow$ | $RMSE^p\downarrow$ | $REL^p\downarrow$ | $\delta_1^p\uparrow$ | Rk.$\downarrow$ | $MAE^p\downarrow$ | $RMSE^p\downarrow$ | $REL^p\downarrow$ | $\delta_1^p\uparrow$ | Rk.$\downarrow$ | $MAE^p\downarrow$ | $RMSE^p\downarrow$ | $REL^p\downarrow$ | $\delta_1^p\uparrow$ | Rk.$\downarrow$ |
| UniDepth V1 | 2.429 | 2.649 | 0.641 | 0.066 | 9.000 | 2.429 | 2.649 | 0.641 | 0.066 | 9.000 | 2.429 | 2.649 | 0.641 | 0.066 | 9.000 |
| UniDepth V2 | 0.624 | 0.726 | 0.166 | 0.825 | 8.000 | 0.624 | 0.726 | 0.166 | 0.825 | 8.000 | 0.624 | 0.726 | 0.166 | 0.825 | 8.000 |
| MoGe V2 | 0.500 | 0.620 | 0.123 | 0.839 | 7.000 | 0.500 | 0.620 | 0.123 | 0.839 | 7.000 | 0.500 | 0.620 | 0.123 | 0.839 | 7.000 |
| G2-MonoDepth‡ | 0.361 | 0.412 | 0.088 | 0.956 | 2.750 | 0.362 | 0.416 | 0.088 | 0.956 | 4.000 | 0.363 | 0.421 | 0.089 | 0.956 | 4.500 |
| OMNI-DC | 0.367 | 0.436 | 0.089 | 0.954 | 5.250 | 0.361 | 0.413 | 0.088 | 0.956 | 3.000 | 0.361 | 0.417 | 0.088 | 0.956 | 3.000 |
| PriorDA | 0.371 | 0.438 | 0.089 | 0.956 | 5.000 | 0.365 | 0.427 | 0.089 | 0.956 | 5.250 | 0.365 | 0.429 | 0.089 | 0.956 | 5.000 |
| SPNet‡ | 0.361 | 0.414 | 0.088 | 0.956 | 3.000 | 0.361 | 0.413 | 0.088 | 0.956 | 3.000 | 0.361 | 0.419 | 0.088 | 0.956 | 3.250 |
| PromptDA | 0.331 | 0.417 | 0.078 | 0.968 | 2.500 | 0.319 | 0.397 | 0.078 | 0.967 | 2.000 | 0.321 | 0.404 | 0.078 | 0.967 | 2.000 |
| LDCM (ours) | **0.256** | **0.300** | **0.070** | **0.999** | **1.000** | **0.255** | **0.300** | **0.070** | **0.999** | **1.000** | **0.255** | **0.302** | **0.070** | **0.999** | **1.000** |

| Method | 1% | | | | | 500 | | | | | 100 | | | | |
|---|---|---|---|---|---|---|---|---|---|---|---|---|---|---|---|
| | $MAE^p\downarrow$ | $RMSE^p\downarrow$ | $REL^p\downarrow$ | $\delta_1^p\uparrow$ | Rk.$\downarrow$ | $MAE^p\downarrow$ | $RMSE^p\downarrow$ | $REL^p\downarrow$ | $\delta_1^p\uparrow$ | Rk.$\downarrow$ | $MAE^p\downarrow$ | $RMSE^p\downarrow$ | $REL^p\downarrow$ | $\delta_1^p\uparrow$ | Rk.$\downarrow$ |
| UniDepth V1 | 2.429 | 2.649 | 0.641 | 0.066 | 9.000 | 2.429 | 2.649 | 0.641 | 0.066 | 9.000 | 2.429 | 2.649 | 0.641 | 0.066 | 9.000 |
| UniDepth V2 | 0.624 | 0.726 | 0.166 | 0.825 | 8.000 | 0.624 | 0.726 | 0.166 | 0.825 | 8.000 | 0.624 | 0.726 | 0.166 | 0.825 | 8.000 |
| MoGe V2 | 0.500 | 0.620 | 0.123 | 0.839 | 7.000 | 0.500 | 0.620 | 0.123 | 0.839 | 7.000 | 0.500 | 0.620 | 0.123 | 0.839 | 6.750 |
| G2-MonoDepth‡ | 0.367 | 0.440 | 0.089 | 0.955 | 5.000 | 0.388 | 0.511 | 0.092 | 0.952 | 6.000 | 0.468 | 0.662 | 0.109 | 0.915 | 6.250 |
| OMNI-DC | 0.363 | 0.428 | 0.089 | 0.956 | 3.000 | 0.372 | 0.460 | 0.090 | 0.954 | 3.500 | 0.407 | 0.566 | 0.095 | 0.947 | 4.750 |
| PriorDA | 0.367 | 0.435 | 0.089 | 0.956 | 3.750 | 0.371 | 0.449 | 0.090 | 0.955 | 2.750 | 0.385 | 0.486 | 0.092 | 0.952 | 2.500 |
| SPNet‡ | 0.365 | 0.439 | 0.089 | 0.956 | 3.750 | 0.375 | 0.484 | 0.090 | 0.955 | 4.000 | 0.399 | 0.559 | 0.093 | 0.950 | 3.750 |
| PromptDA | 0.328 | 0.423 | 0.079 | 0.966 | 2.000 | 0.348 | 0.485 | 0.082 | 0.963 | 2.750 | 0.396 | 0.608 | 0.088 | 0.955 | 3.000 |
| LDCM (ours) | **0.257** | **0.311** | **0.070** | **0.998** | **1.000** | **0.263** | **0.334** | **0.071** | **0.998** | **1.000** | **0.279** | **0.382** | **0.073** | **0.996** | **1.000** |

| Method | SIFT | | | | | ORB | | | | | Virtual-Lidar-32-Lines | | | | |
|---|---|---|---|---|---|---|---|---|---|---|---|---|---|---|---|
| | $MAE^p\downarrow$ | $RMSE^p\downarrow$ | $REL^p\downarrow$ | $\delta_1^p\uparrow$ | Rk.$\downarrow$ | $MAE^p\downarrow$ | $RMSE^p\downarrow$ | $REL^p\downarrow$ | $\delta_1^p\uparrow$ | Rk.$\downarrow$ | $MAE^p\downarrow$ | $RMSE^p\downarrow$ | $REL^p\downarrow$ | $\delta_1^p\uparrow$ | Rk.$\downarrow$ |
| UniDepth V1 | 2.429 | 2.649 | 0.641 | 0.066 | 9.000 | 2.429 | 2.649 | 0.641 | 0.066 | 9.000 | 2.429 | 2.649 | 0.641 | 0.066 | 9.000 |
| UniDepth V2 | 0.624 | 0.726 | 0.166 | 0.825 | 8.000 | 0.624 | 0.726 | 0.166 | 0.825 | 8.000 | 0.624 | 0.726 | 0.166 | 0.825 | 8.000 |
| MoGe V2 | 0.500 | 0.620 | 0.123 | 0.839 | 6.250 | 0.500 | 0.620 | 0.123 | 0.839 | 6.250 | 0.500 | 0.620 | 0.123 | 0.839 | 7.000 |
| G2-MonoDepth‡ | 0.517 | 0.675 | 0.136 | 0.844 | 6.750 | 0.520 | 0.695 | 0.135 | 0.853 | 6.750 | 0.376 | 0.460 | 0.091 | 0.953 | 6.000 |
| OMNI-DC | 0.429 | 0.532 | 0.107 | 0.904 | 4.500 | 0.447 | 0.567 | 0.112 | 0.890 | 4.250 | 0.365 | 0.437 | 0.089 | 0.955 | 3.250 |
| PriorDA | 0.394 | 0.477 | 0.099 | 0.939 | 2.000 | 0.408 | 0.511 | 0.101 | 0.936 | 2.000 | 0.367 | 0.438 | 0.089 | 0.956 | 3.500 |
| SPNet‡ | 0.408 | 0.515 | 0.100 | 0.924 | 3.000 | 0.419 | 0.539 | 0.102 | 0.922 | 3.000 | 0.367 | 0.450 | 0.089 | 0.955 | 4.000 |
| PromptDA | 0.436 | 0.557 | 0.106 | 0.909 | 4.500 | 0.455 | 0.582 | 0.112 | 0.893 | 4.500 | 0.326 | 0.420 | 0.078 | 0.968 | 2.000 |
| LDCM (ours) | **0.279** | **0.350** | **0.075** | **0.991** | **1.000** | **0.285** | **0.360** | **0.077** | **0.986** | **1.000** | **0.258** | **0.319** | **0.070** | **0.998** | **1.000** |

| Method | Virtual-Lidar-16-Lines | | | | | Virtual-Lidar-8-Lines | | | | | Average | | | | |
|---|---|---|---|---|---|---|---|---|---|---|---|---|---|---|---|
| | $MAE^p\downarrow$ | $RMSE^p\downarrow$ | $REL^p\downarrow$ | $\delta_1^p\uparrow$ | Rk.$\downarrow$ | $MAE^p\downarrow$ | $RMSE^p\downarrow$ | $REL^p\downarrow$ | $\delta_1^p\uparrow$ | Rk.$\downarrow$ | $MAE^p\downarrow$ | $RMSE^p\downarrow$ | $REL^p\downarrow$ | $\delta_1^p\uparrow$ | Rk.$\downarrow$ |
| UniDepth V1 | 2.429 | 2.649 | 0.641 | 0.066 | 9.000 | 2.429 | 2.649 | 0.641 | 0.066 | 9.000 | 2.429 | 2.649 | 0.641 | 0.066 | 9.000 |
| UniDepth V2 | 0.624 | 0.726 | 0.166 | 0.825 | 8.000 | 0.624 | 0.726 | 0.166 | 0.825 | 8.000 | 0.624 | 0.726 | 0.166 | 0.825 | 8.000 |
| MoGe V2 | 0.500 | 0.620 | 0.123 | 0.839 | 7.000 | 0.500 | 0.620 | 0.123 | 0.839 | 7.000 | 0.500 | 0.620 | 0.123 | 0.839 | 6.841 |
| G2-MonoDepth‡ | 0.383 | 0.487 | 0.092 | 0.952 | 5.455 | 0.421 | 0.563 | 0.098 | 0.943 | 6.000 | 0.411 | 0.522 | 0.101 | 0.930 | 5.455 |
| OMNI-DC | 0.370 | 0.451 | 0.090 | 0.955 | 3.000 | 0.389 | 0.505 | 0.093 | 0.950 | 4.000 | 0.385 | 0.474 | 0.094 | 0.943 | 3.773 |
| PriorDA | 0.371 | 0.447 | 0.090 | 0.955 | 3.000 | 0.379 | 0.465 | 0.091 | 0.952 | 2.750 | 0.377 | 0.455 | 0.092 | 0.952 | 3.409 |
| SPNet‡ | 0.373 | 0.469 | 0.090 | 0.954 | 4.500 | 0.399 | 0.523 | 0.094 | 0.948 | 5.000 | 0.381 | 0.475 | 0.092 | 0.948 | 3.659 |
| PromptDA | 0.338 | 0.459 | 0.080 | 0.965 | 2.500 | 0.360 | 0.500 | 0.085 | 0.959 | 2.250 | 0.360 | 0.477 | 0.086 | 0.953 | 2.727 |
| LDCM (ours) | **0.263** | **0.332** | **0.071** | **0.998** | **1.000** | **0.272** | **0.354** | **0.072** | **0.996** | **1.000** | **0.266** | **0.331** | **0.072** | **0.996** | **1.000** |

Table 23: **Quantitative comparison of point map estimation with baseline methods on the outdoor scenes of the ETH3D dataset** Schops et al. (2017). Methods marked with ‡ use scenario-specific configurations for indoor and outdoor scenes, respectively. The best and the second-best results are highlighted.

| Method | Virtual-Lidar-64-Lines | | | | | Virtual-Lidar-32-Lines | | | | | Virtual-Lidar-16-Lines | | | | |
|---|---|---|---|---|---|---|---|---|---|---|---|---|---|---|---|
| | $MAE^p\downarrow$ | $RMSE^p\downarrow$ | $REL^p\downarrow$ | $\delta_1^p\uparrow$ | Rk.↓ | $MAE^p\downarrow$ | $RMSE^p\downarrow$ | $REL^p\downarrow$ | $\delta_1^p\uparrow$ | Rk.↓ | $MAE^p\downarrow$ | $RMSE^p\downarrow$ | $REL^p\downarrow$ | $\delta_1^p\uparrow$ | Rk.↓ |
| UniDepth V1 | 4.653 | 5.100 | 0.461 | 0.145 | 9.000 | 4.653 | 5.100 | 0.461 | 0.145 | 9.000 | 4.653 | 5.100 | 0.461 | 0.145 | 9.000 |
| UniDepth V2 | 1.879 | 2.844 | 0.216 | 0.712 | 8.000 | 1.879 | 2.844 | 0.216 | 0.712 | 8.000 | 1.879 | 2.844 | 0.216 | 0.712 | 8.000 |
| MoGe V2 | 0.931 | 1.206 | 0.115 | 0.890 | 6.750 | 0.931 | 1.206 | 0.115 | 0.890 | 6.750 | 0.931 | 1.206 | 0.115 | 0.890 | 6.500 |
| G2-MonoDepth‡ | 0.638 | 0.843 | 0.088 | 0.929 | 4.250 | 0.665 | 0.943 | 0.092 | 0.924 | 4.750 | 0.713 | 1.091 | 0.098 | 0.916 | 5.250 |
| OMNI-DC | 0.626 | 0.793 | 0.086 | 0.931 | 2.000 | 0.634 | 0.823 | 0.087 | 0.930 | 2.250 | 0.655 | 0.871 | 0.089 | 0.927 | 2.250 |
| PriorDA | 0.641 | 0.804 | 0.087 | 0.929 | 3.750 | 0.646 | 0.818 | 0.088 | 0.929 | 2.750 | 0.657 | 0.846 | 0.089 | 0.927 | 2.250 |
| SPNet‡ | 0.631 | 0.893 | 0.087 | 0.931 | 3.250 | 0.653 | 1.069 | 0.089 | 0.929 | 4.000 | 0.704 | 1.354 | 0.096 | 0.925 | 5.000 |
| PromptDA | 0.756 | 1.329 | 0.094 | 0.923 | 6.250 | 0.762 | 1.365 | 0.094 | 0.921 | 6.250 | 0.772 | 1.431 | 0.092 | 0.927 | 4.750 |
| LDCM (ours) | **0.393** | **0.510** | **0.041** | **0.998** | **1.000** | **0.402** | **0.539** | **0.042** | **0.998** | **1.000** | **0.414** | **0.572** | **0.043** | **0.997** | **1.000** |

| Method | Virtual-Lidar-8-Lines | | | | | Virtual-Lidar-4-Lines | | | | | 10% Noise | | | | |
|---|---|---|---|---|---|---|---|---|---|---|---|---|---|---|---|
| | $MAE^p\downarrow$ | $RMSE^p\downarrow$ | $REL^p\downarrow$ | $\delta_1^p\uparrow$ | Rk.↓ | $MAE^p\downarrow$ | $RMSE^p\downarrow$ | $REL^p\downarrow$ | $\delta_1^p\uparrow$ | Rk.↓ | $MAE^p\downarrow$ | $RMSE^p\downarrow$ | $REL^p\downarrow$ | $\delta_1^p\uparrow$ | Rk.↓ |
| UniDepth V1 | 4.653 | 5.100 | 0.461 | 0.145 | 9.000 | 4.653 | 5.100 | 0.461 | 0.145 | 9.000 | 4.653 | 5.100 | 0.461 | 0.145 | 9.000 |
| UniDepth V2 | 1.879 | 2.844 | 0.216 | 0.712 | 8.000 | 1.879 | 2.844 | 0.216 | 0.712 | 8.000 | 1.879 | 2.844 | 0.216 | 0.712 | 8.000 |
| MoGe V2 | 0.931 | 1.206 | 0.115 | 0.890 | 5.500 | 0.931 | 1.206 | 0.115 | 0.890 | 2.000 | 0.931 | 1.206 | 0.115 | 0.890 | 7.000 |
| G2-MonoDepth‡ | 0.907 | 1.448 | 0.118 | 0.876 | 6.250 | 1.541 | 2.283 | 0.201 | 0.724 | 6.750 | 0.618 | 0.732 | 0.085 | 0.932 | 3.000 |
| OMNI-DC | 0.757 | 1.067 | 0.098 | 0.913 | 3.000 | 1.209 | 1.726 | 0.143 | 0.820 | 5.000 | 0.623 | 0.758 | 0.085 | 0.931 | 3.750 |
| PriorDA | 0.705 | 0.957 | 0.093 | 0.921 | 2.000 | 1.030 | 1.479 | 0.140 | 0.850 | 3.500 | 0.657 | 0.817 | 0.088 | 0.930 | 5.000 |
| SPNet‡ | 0.858 | 1.816 | 0.116 | 0.902 | 5.500 | 1.222 | 2.483 | 0.160 | 0.823 | 6.000 | 0.613 | 0.725 | 0.084 | 0.933 | 2.000 |
| PromptDA | 0.872 | 1.711 | 0.103 | 0.913 | 4.500 | 1.137 | 1.783 | 0.131 | 0.863 | 3.750 | 0.715 | 1.180 | 0.088 | 0.929 | 5.750 |
| LDCM (ours) | **0.451** | **0.676** | **0.046** | **0.994** | **1.000** | **0.552** | **0.821** | **0.058** | **0.982** | **1.000** | **0.394** | **0.476** | **0.040** | **0.999** | **1.000** |

| Method | 5% | | | | | 3% | | | | | 1% | | | | |
|---|---|---|---|---|---|---|---|---|---|---|---|---|---|---|---|
| | $MAE^p\downarrow$ | $RMSE^p\downarrow$ | $REL^p\downarrow$ | $\delta_1^p\uparrow$ | Rk.↓ | $MAE^p\downarrow$ | $RMSE^p\downarrow$ | $REL^p\downarrow$ | $\delta_1^p\uparrow$ | Rk.↓ | $MAE^p\downarrow$ | $RMSE^p\downarrow$ | $REL^p\downarrow$ | $\delta_1^p\uparrow$ | Rk.↓ |
| UniDepth V1 | 4.653 | 5.100 | 0.461 | 0.145 | 9.000 | 4.653 | 5.100 | 0.461 | 0.145 | 9.000 | 4.653 | 5.100 | 0.461 | 0.145 | 9.000 |
| UniDepth V2 | 1.879 | 2.844 | 0.216 | 0.712 | 8.000 | 1.879 | 2.844 | 0.216 | 0.712 | 8.000 | 1.879 | 2.844 | 0.216 | 0.712 | 8.000 |
| MoGe V2 | 0.931 | 1.206 | 0.115 | 0.890 | 7.000 | 0.931 | 1.206 | 0.115 | 0.890 | 6.750 | 0.931 | 1.206 | 0.115 | 0.890 | 6.750 |
| G2-MonoDepth‡ | 0.621 | 0.748 | 0.085 | 0.932 | 2.750 | 0.626 | 0.776 | 0.086 | 0.931 | 3.750 | 0.648 | 0.894 | 0.089 | 0.927 | 4.500 |
| OMNI-DC | 0.617 | 0.746 | 0.085 | 0.932 | 2.000 | 0.619 | 0.761 | 0.085 | 0.932 | 2.000 | 0.629 | 0.805 | 0.086 | 0.930 | 2.000 |
| PriorDA | 0.638 | 0.789 | 0.087 | 0.930 | 5.000 | 0.639 | 0.795 | 0.087 | 0.930 | 4.750 | 0.644 | 0.814 | 0.088 | 0.929 | 3.250 |
| SPNet‡ | 0.617 | 0.760 | 0.085 | 0.932 | 2.500 | 0.621 | 0.801 | 0.085 | 0.932 | 3.000 | 0.644 | 0.994 | 0.089 | 0.930 | 3.500 |
| PromptDA | 0.707 | 1.202 | 0.089 | 0.929 | 6.000 | 0.733 | 1.253 | 0.091 | 0.926 | 6.250 | 0.771 | 1.358 | 0.095 | 0.921 | 6.250 |
| LDCM (ours) | **0.394** | **0.486** | **0.040** | **0.999** | **1.000** | **0.396** | **0.498** | **0.040** | **0.999** | **1.000** | **0.405** | **0.539** | **0.041** | **0.998** | **1.000** |

| Method | SIFT | | | | | ORB | | | | | Average | | | | |
|---|---|---|---|---|---|---|---|---|---|---|---|---|---|---|---|
| | $MAE^p\downarrow$ | $RMSE^p\downarrow$ | $REL^p\downarrow$ | $\delta_1^p\uparrow$ | Rk.↓ | $MAE^p\downarrow$ | $RMSE^p\downarrow$ | $REL^p\downarrow$ | $\delta_1^p\uparrow$ | Rk.↓ | $MAE^p\downarrow$ | $RMSE^p\downarrow$ | $REL^p\downarrow$ | $\delta_1^p\uparrow$ | Rk.↓ |
| UniDepth V1 | 4.653 | 5.100 | 0.461 | 0.145 | 9.000 | 4.653 | 5.100 | 0.461 | 0.145 | 9.000 | 4.653 | 5.100 | 0.461 | 0.145 | 9.000 |
| UniDepth V2 | 1.879 | 2.844 | 0.216 | 0.712 | 8.000 | 1.879 | 2.844 | 0.216 | 0.712 | 8.000 | 1.879 | 2.844 | 0.216 | 0.712 | 8.000 |
| MoGe V2 | 0.931 | 1.206 | 0.115 | 0.890 | 5.750 | 0.931 | 1.206 | 0.115 | 0.890 | 5.000 | 0.931 | 1.206 | 0.115 | 0.890 | 5.977 |
| G2-MonoDepth‡ | 0.876 | 1.346 | 0.129 | 0.848 | 6.250 | 0.885 | 1.318 | 0.130 | 0.857 | 6.000 | 0.794 | 1.129 | 0.109 | 0.891 | 4.864 |
| OMNI-DC | 0.717 | 0.973 | 0.097 | 0.914 | 3.000 | 0.765 | 1.080 | 0.101 | 0.902 | 3.500 | 0.714 | 0.946 | 0.095 | 0.915 | 2.795 |
| PriorDA | 0.688 | 0.890 | 0.094 | 0.920 | 2.000 | 0.730 | 0.979 | 0.099 | 0.912 | 2.000 | 0.698 | 0.908 | 0.095 | 0.919 | 3.295 |
| SPNet‡ | 0.754 | 1.453 | 0.106 | 0.909 | 4.500 | 0.743 | 1.321 | 0.102 | 0.910 | 6.000 | 0.733 | 1.243 | 0.100 | 0.914 | 3.932 |
| PromptDA | 0.871 | 1.454 | 0.106 | 0.899 | 5.250 | 0.963 | 1.576 | 0.119 | 0.865 | 6.500 | 0.824 | 1.422 | 0.100 | 0.911 | 5.591 |
| LDCM (ours) | **0.444** | **0.615** | **0.046** | **0.993** | **1.000** | **0.454** | **0.644** | **0.047** | **0.990** | **1.000** | **0.427** | **0.580** | **0.044** | **0.995** | **1.000** |

Table 24: **Quantitative comparison of affine-invariant point map estimation with baseline methods on the KITTI dataset** Geiger et al. (2012); Uhrig et al. (2017). The best and the second-best results are highlighted.

| Method | Lidar-64-Lines | | | Lidar-32-Lines | | | Lidar-16-Lines | | | Lidar-8-Lines | | |
|---|---|---|---|---|---|---|---|---|---|---|---|---|
| | $REL^p \downarrow$ | $\delta_1^p \uparrow$ | Rk.$\downarrow$ | $REL^p \downarrow$ | $\delta_1^p \uparrow$ | Rk.$\downarrow$ | $REL^p \downarrow$ | $\delta_1^p \uparrow$ | Rk.$\downarrow$ | $REL^p \downarrow$ | $\delta_1^p \uparrow$ | Rk.$\downarrow$ |
| VGGT | 0.147 | 0.823 | 5.000 | 0.147 | 0.823 | 4.500 | 0.147 | 0.823 | 4.500 | 0.147 | 0.823 | 4.500 |
| MoGe V2 | 0.056 | 0.968 | 2.000 | 0.056 | 0.968 | 2.000 | 0.056 | 0.968 | 2.000 | 0.056 | 0.968 | 2.000 |
| WorldMirror | 0.095 | 0.920 | 3.000 | 0.103 | 0.900 | 3.000 | 0.118 | 0.865 | 3.000 | 0.129 | 0.838 | 3.500 |
| MapAnything | 0.362 | 0.347 | 6.000 | 0.364 | 0.345 | 6.000 | 0.364 | 0.346 | 6.000 | 0.366 | 0.345 | 6.000 |
| Pow3R | 0.140 | 0.886 | 4.000 | 0.147 | 0.870 | 4.000 | 0.151 | 0.858 | 4.500 | 0.153 | 0.848 | 4.000 |
| LDCM | **0.031** | **0.993** | **1.000** | **0.033** | **0.991** | **1.000** | **0.034** | **0.989** | **1.000** | **0.036** | **0.987** | **1.000** |
| Method | Lidar-4-Lines | | | 10% | | | 5% | | | 3% | | |
| | $REL^p \downarrow$ | $\delta_1^p \uparrow$ | Rk.$\downarrow$ | $REL^p \downarrow$ | $\delta_1^p \uparrow$ | Rk.$\downarrow$ | $REL^p \downarrow$ | $\delta_1^p \uparrow$ | Rk.$\downarrow$ | $REL^p \downarrow$ | $\delta_1^p \uparrow$ | Rk.$\downarrow$ |
| VGGT | 0.147 | 0.823 | 4.000 | 0.147 | 0.823 | 4.500 | 0.147 | 0.823 | 4.500 | 0.147 | 0.823 | 4.500 |
| MoGe V2 | 0.056 | 0.968 | 2.000 | 0.056 | 0.968 | 2.000 | 0.056 | 0.968 | 2.000 | 0.056 | 0.968 | 2.000 |
| WorldMirror | 0.136 | 0.821 | 4.000 | 0.102 | 0.900 | 3.000 | 0.100 | 0.901 | 3.000 | 0.100 | 0.902 | 3.000 |
| MapAnything | 0.367 | 0.344 | 6.000 | 0.365 | 0.345 | 6.000 | 0.365 | 0.345 | 6.000 | 0.367 | 0.341 | 6.000 |
| Pow3R | 0.154 | 0.844 | 4.000 | 0.154 | 0.847 | 4.500 | 0.155 | 0.843 | 4.500 | 0.155 | 0.841 | 4.500 |
| LDCM | **0.043** | **0.976** | **1.000** | **0.034** | **0.990** | **1.000** | **0.035** | **0.989** | **1.000** | **0.036** | **0.987** | **1.000** |
| Method | 1% | | | SIFT | | | ORB | | | Average | | |
| | $REL^p \downarrow$ | $\delta_1^p \uparrow$ | Rk.$\downarrow$ | $REL^p \downarrow$ | $\delta_1^p \uparrow$ | Rk.$\downarrow$ | $REL^p \downarrow$ | $\delta_1^p \uparrow$ | Rk.$\downarrow$ | $REL^p \downarrow$ | $\delta_1^p \uparrow$ | Rk.$\downarrow$ |
| VGGT | 0.147 | 0.823 | 4.500 | 0.147 | 0.823 | 4.500 | 0.147 | 0.823 | 4.500 | 0.147 | 0.823 | 4.500 |
| MoGe V2 | 0.056 | 0.968 | 2.000 | 0.056 | 0.968 | 1.500 | 0.056 | 0.968 | 1.500 | 0.056 | 0.968 | 1.909 |
| WorldMirror | 0.100 | 0.902 | 3.000 | 0.100 | 0.902 | 3.000 | 0.102 | 0.897 | 3.000 | 0.108 | 0.886 | 3.136 |
| MapAnything | 0.370 | 0.338 | 6.000 | 0.367 | 0.344 | 6.000 | 0.367 | 0.343 | 6.000 | 0.366 | 0.344 | 6.000 |
| Pow3R | 0.155 | 0.839 | 4.500 | 0.155 | 0.839 | 4.500 | 0.155 | 0.840 | 4.500 | 0.152 | 0.850 | 4.318 |
| LDCM | **0.040** | **0.983** | **1.000** | **0.054** | **0.961** | **1.500** | **0.051** | **0.963** | **1.500** | **0.039** | **0.983** | **1.091** |

Table 25: **Quantitative comparison of affine-invariant point map estimation with baseline methods on the indoor scenes of the DIODE dataset** Vasiljevic et al. (2019). The best and the second-best results are highlighted.

| Method | 10% Noise | | | 5% | | | 3% | | | 1% | | |
|---|---|---|---|---|---|---|---|---|---|---|---|---|
| | $REL^p \downarrow$ | $\delta_1^p \uparrow$ | Rk.$\downarrow$ | $REL^p \downarrow$ | $\delta_1^p \uparrow$ | Rk.$\downarrow$ | $REL^p \downarrow$ | $\delta_1^p \uparrow$ | Rk.$\downarrow$ | $REL^p \downarrow$ | $\delta_1^p \uparrow$ | Rk.$\downarrow$ |
| VGGT | 0.107 | 0.926 | 5.000 | 0.107 | 0.926 | 5.000 | 0.107 | 0.926 | 5.000 | 0.107 | 0.926 | 4.500 |
| MoGe V2 | 0.052 | 0.972 | 2.000 | 0.052 | 0.972 | 2.000 | 0.052 | 0.972 | 2.000 | 0.052 | 0.972 | 2.000 |
| WorldMirror | 0.079 | 0.951 | 3.500 | 0.072 | 0.952 | 3.500 | 0.070 | 0.953 | 3.000 | 0.071 | 0.953 | 3.000 |
| MapAnything | 0.169 | 0.762 | 6.000 | 0.168 | 0.762 | 6.000 | 0.168 | 0.762 | 6.000 | 0.169 | 0.763 | 6.000 |
| Pow3R | 0.099 | 0.960 | 3.500 | 0.104 | 0.954 | 3.500 | 0.106 | 0.951 | 4.000 | 0.108 | 0.946 | 4.500 |
| LDCM | **0.009** | **1.000** | **1.000** | **0.009** | **1.000** | **1.000** | **0.009** | **1.000** | **1.000** | **0.009** | **0.999** | **1.000** |
| Method | 500 | | | 100 | | | SIFT | | | ORB | | |
| | $REL^p \downarrow$ | $\delta_1^p \uparrow$ | Rk.$\downarrow$ | $REL^p \downarrow$ | $\delta_1^p \uparrow$ | Rk.$\downarrow$ | $REL^p \downarrow$ | $\delta_1^p \uparrow$ | Rk.$\downarrow$ | $REL^p \downarrow$ | $\delta_1^p \uparrow$ | Rk.$\downarrow$ |
| VGGT | 0.107 | 0.926 | 4.500 | 0.107 | 0.926 | 4.500 | 0.107 | 0.926 | 4.500 | 0.107 | 0.926 | 4.500 |
| MoGe V2 | 0.052 | 0.972 | 2.000 | 0.052 | 0.972 | 2.000 | 0.052 | 0.972 | 2.000 | 0.052 | 0.972 | 1.500 |
| WorldMirror | 0.072 | 0.955 | 3.000 | 0.073 | 0.955 | 3.000 | 0.073 | 0.956 | 3.000 | 0.078 | 0.945 | 3.000 |
| MapAnything | 0.175 | 0.753 | 6.000 | 0.173 | 0.758 | 6.000 | 0.176 | 0.753 | 6.000 | 0.175 | 0.758 | 6.000 |
| Pow3R | 0.110 | 0.944 | 4.500 | 0.110 | 0.943 | 4.500 | 0.109 | 0.944 | 4.500 | 0.109 | 0.944 | 4.500 |
| LDCM | **0.010** | **0.999** | **1.000** | **0.012** | **0.996** | **1.000** | **0.029** | **0.979** | **1.000** | **0.036** | **0.972** | **1.000** |
| Method | Virtual-Lidar-32-Lines | | | Virtual-Lidar-16-Lines | | | Virtual-Lidar-8-Lines | | | Average | | |
| | $REL^p \downarrow$ | $\delta_1^p \uparrow$ | Rk.$\downarrow$ | $REL^p \downarrow$ | $\delta_1^p \uparrow$ | Rk.$\downarrow$ | $REL^p \downarrow$ | $\delta_1^p \uparrow$ | Rk.$\downarrow$ | $REL^p \downarrow$ | $\delta_1^p \uparrow$ | Rk.$\downarrow$ |
| VGGT | 0.107 | 0.926 | 4.500 | 0.107 | 0.926 | 4.500 | 0.107 | 0.926 | 4.500 | 0.107 | 0.926 | 4.636 |
| MoGe V2 | 0.052 | 0.972 | 2.000 | 0.052 | 0.972 | 2.000 | 0.052 | 0.972 | 2.000 | 0.052 | 0.972 | 1.955 |
| WorldMirror | 0.072 | 0.955 | 3.000 | 0.072 | 0.955 | 3.000 | 0.073 | 0.955 | 3.000 | 0.073 | 0.953 | 3.091 |
| MapAnything | 0.175 | 0.753 | 6.000 | 0.173 | 0.758 | 6.000 | 0.173 | 0.759 | 6.000 | 0.172 | 0.758 | 6.000 |
| Pow3R | 0.109 | 0.945 | 4.500 | 0.109 | 0.943 | 4.500 | 0.110 | 0.944 | 4.500 | 0.108 | 0.947 | 4.273 |
| LDCM | **0.010** | **0.999** | **1.000** | **0.010** | **0.999** | **1.000** | **0.011** | **0.997** | **1.000** | **0.014** | **0.995** | **1.000** |

Table 26: **Quantitative comparison of affine-invariant point map estimation with baseline methods on the outdoor scenes of the DIODE dataset** Vasiljevic et al. (2019). The best and the second-best results are highlighted.

| Method | Virtual-Lidar-64-Lines | | | Virtual-Lidar-32-Lines | | | Virtual-Lidar-16-Lines | | | Virtual-Lidar-8-Lines | | |
|---|---|---|---|---|---|---|---|---|---|---|---|---|
| | $REL^p \downarrow$ | $\delta_1^p \uparrow$ | Rk.$\downarrow$ | $REL^p \downarrow$ | $\delta_1^p \uparrow$ | Rk.$\downarrow$ | $REL^p \downarrow$ | $\delta_1^p \uparrow$ | Rk.$\downarrow$ | $REL^p \downarrow$ | $\delta_1^p \uparrow$ | Rk.$\downarrow$ |
| VGGT | 0.215 | 0.700 | 5.000 | 0.215 | 0.700 | 5.000 | 0.215 | 0.700 | 5.000 | 0.215 | 0.700 | 5.000 |
| MoGe V2 | 0.124 | 0.841 | 2.000 | 0.124 | 0.841 | 2.000 | 0.124 | 0.841 | 2.000 | 0.124 | 0.841 | 2.000 |
| WorldMirror | 0.156 | 0.786 | 3.000 | 0.156 | 0.788 | 3.000 | 0.154 | 0.792 | 3.000 | 0.155 | 0.792 | 3.000 |
| MapAnything | 0.299 | 0.506 | 6.000 | 0.310 | 0.481 | 6.000 | 0.309 | 0.489 | 6.000 | 0.317 | 0.487 | 6.000 |
| Pow3R | 0.200 | 0.745 | 4.000 | 0.200 | 0.747 | 4.000 | 0.201 | 0.743 | 4.000 | 0.201 | 0.745 | 4.000 |
| LDCM | **0.072** | **0.965** | **1.000** | **0.079** | **0.950** | **1.000** | **0.090** | **0.920** | **1.000** | **0.097** | **0.903** | **1.000** |
| Method | Virtual-Lidar-4-Lines | | | 10% Noise | | | 5% | | | 3% | | |
| | $REL^p \downarrow$ | $\delta_1^p \uparrow$ | Rk.$\downarrow$ | $REL^p \downarrow$ | $\delta_1^p \uparrow$ | Rk.$\downarrow$ | $REL^p \downarrow$ | $\delta_1^p \uparrow$ | Rk.$\downarrow$ | $REL^p \downarrow$ | $\delta_1^p \uparrow$ | Rk.$\downarrow$ |
| VGGT | 0.215 | 0.700 | 5.000 | 0.215 | 0.700 | 5.000 | 0.215 | 0.700 | 5.000 | 0.215 | 0.700 | 5.000 |
| MoGe V2 | 0.124 | 0.841 | 2.000 | 0.124 | 0.841 | 2.000 | 0.124 | 0.841 | 2.000 | 0.124 | 0.841 | 2.000 |
| WorldMirror | 0.154 | 0.793 | 3.000 | 0.140 | 0.822 | 3.000 | 0.151 | 0.799 | 3.000 | 0.157 | 0.782 | 3.000 |
| MapAnything | 0.311 | 0.500 | 6.000 | 0.296 | 0.500 | 6.000 | 0.291 | 0.510 | 6.000 | 0.291 | 0.517 | 6.000 |
| Pow3R | 0.201 | 0.745 | 4.000 | 0.181 | 0.770 | 4.000 | 0.190 | 0.756 | 4.000 | 0.194 | 0.752 | 4.000 |
| LDCM | **0.117** | **0.856** | **1.000** | **0.059** | **0.988** | **1.000** | **0.061** | **0.985** | **1.000** | **0.063** | **0.981** | **1.000** |
| Method | 1% | | | SIFT | | | ORB | | | Average | | |
| | $REL^p \downarrow$ | $\delta_1^p \uparrow$ | Rk.$\downarrow$ | $REL^p \downarrow$ | $\delta_1^p \uparrow$ | Rk.$\downarrow$ | $REL^p \downarrow$ | $\delta_1^p \uparrow$ | Rk.$\downarrow$ | $REL^p \downarrow$ | $\delta_1^p \uparrow$ | Rk.$\downarrow$ |
| VGGT | 0.215 | 0.700 | 5.000 | 0.215 | 0.700 | 5.000 | 0.215 | 0.700 | 5.000 | 0.215 | 0.700 | 5.000 |
| MoGe V2 | 0.124 | 0.841 | 2.000 | 0.124 | 0.841 | 2.000 | 0.124 | 0.841 | 2.000 | 0.124 | 0.841 | 2.000 |
| WorldMirror | 0.158 | 0.781 | 3.000 | 0.157 | 0.784 | 3.000 | 0.170 | 0.750 | 3.500 | 0.155 | 0.788 | 3.045 |
| MapAnything | 0.293 | 0.515 | 6.000 | 0.301 | 0.505 | 6.000 | 0.302 | 0.505 | 6.000 | 0.302 | 0.501 | 6.000 |
| Pow3R | 0.199 | 0.747 | 4.000 | 0.200 | 0.748 | 4.000 | 0.197 | 0.751 | 3.500 | 0.197 | 0.750 | 3.955 |
| LDCM | **0.068** | **0.971** | **1.000** | **0.071** | **0.966** | **1.000** | **0.073** | **0.957** | **1.000** | **0.077** | **0.949** | **1.000** |

Table 27: **Quantitative comparison of affine-invariant point map estimation with baseline methods on the iBims dataset** Koch et al. (2018). The best and the second-best results are highlighted.

| Method | 10% Noise | | | 5% | | | 3% | | | 1% | | |
|---|---|---|---|---|---|---|---|---|---|---|---|---|
| | $REL^p \downarrow$ | $\delta_1^p \uparrow$ | Rk.$\downarrow$ | $REL^p \downarrow$ | $\delta_1^p \uparrow$ | Rk.$\downarrow$ | $REL^p \downarrow$ | $\delta_1^p \uparrow$ | Rk.$\downarrow$ | $REL^p \downarrow$ | $\delta_1^p \uparrow$ | Rk.$\downarrow$ |
| VGGT | 0.048 | 0.967 | 4.000 | 0.048 | 0.967 | 4.000 | 0.048 | 0.967 | 3.500 | 0.048 | 0.967 | 3.500 |
| MoGe V2 | 0.046 | 0.972 | 2.500 | 0.046 | 0.972 | 2.500 | 0.046 | 0.972 | 2.500 | 0.046 | 0.972 | 2.500 |
| WorldMirror | 0.044 | 0.972 | 2.000 | 0.043 | 0.968 | 2.500 | 0.043 | 0.965 | 3.000 | 0.042 | 0.963 | 3.500 |
| MapAnything | 0.233 | 0.612 | 6.000 | 0.231 | 0.616 | 6.000 | 0.231 | 0.614 | 6.000 | 0.230 | 0.618 | 6.000 |
| Pow3R | 0.077 | 0.952 | 5.000 | 0.068 | 0.962 | 5.000 | 0.064 | 0.965 | 4.500 | 0.061 | 0.967 | 4.000 |
| LDCM | **0.013** | **0.996** | **1.000** | **0.013** | **0.995** | **1.000** | **0.014** | **0.995** | **1.000** | **0.015** | **0.994** | **1.000** |
| Method | 500 | | | 100 | | | SIFT | | | ORB | | |
| | $REL^p \downarrow$ | $\delta_1^p \uparrow$ | Rk.$\downarrow$ | $REL^p \downarrow$ | $\delta_1^p \uparrow$ | Rk.$\downarrow$ | $REL^p \downarrow$ | $\delta_1^p \uparrow$ | Rk.$\downarrow$ | $REL^p \downarrow$ | $\delta_1^p \uparrow$ | Rk.$\downarrow$ |
| VGGT | 0.048 | 0.967 | 4.500 | 0.048 | 0.967 | 4.500 | 0.048 | 0.967 | 4.000 | 0.048 | 0.967 | 3.000 |
| MoGe V2 | 0.046 | 0.972 | 2.500 | 0.046 | 0.972 | 2.500 | 0.046 | 0.972 | 2.500 | 0.046 | 0.972 | 2.000 |
| WorldMirror | 0.042 | 0.968 | 2.500 | 0.042 | 0.968 | 2.500 | 0.042 | 0.967 | 3.000 | 0.060 | 0.946 | 4.500 |
| MapAnything | 0.235 | 0.597 | 6.000 | 0.234 | 0.602 | 6.000 | 0.232 | 0.614 | 6.000 | 0.234 | 0.613 | 6.000 |
| Pow3R | 0.061 | 0.968 | 4.000 | 0.061 | 0.968 | 4.000 | 0.062 | 0.968 | 4.000 | 0.062 | 0.967 | 4.000 |
| LDCM | **0.018** | **0.991** | **1.000** | **0.022** | **0.987** | **1.000** | **0.020** | **0.990** | **1.000** | **0.024** | **0.989** | **1.000** |
| Method | Virtual-Lidar-32-Lines | | | Virtual-Lidar-16-Lines | | | Virtual-Lidar-8-Lines | | | Average | | |
| | $REL^p \downarrow$ | $\delta_1^p \uparrow$ | Rk.$\downarrow$ | $REL^p \downarrow$ | $\delta_1^p \uparrow$ | Rk.$\downarrow$ | $REL^p \downarrow$ | $\delta_1^p \uparrow$ | Rk.$\downarrow$ | $REL^p \downarrow$ | $\delta_1^p \uparrow$ | Rk.$\downarrow$ |
| VGGT | 0.048 | 0.967 | 3.500 | 0.048 | 0.967 | 4.000 | 0.048 | 0.967 | 4.500 | 0.048 | 0.967 | 3.909 |
| MoGe V2 | 0.046 | 0.972 | 2.500 | 0.046 | 0.972 | 2.500 | 0.046 | 0.972 | 2.500 | 0.046 | 0.972 | 2.455 |
| WorldMirror | 0.042 | 0.967 | 2.500 | 0.042 | 0.967 | 3.000 | 0.042 | 0.968 | 2.500 | 0.044 | 0.965 | 2.864 |
| MapAnything | 0.233 | 0.610 | 6.000 | 0.233 | 0.609 | 6.000 | 0.232 | 0.613 | 6.000 | 0.233 | 0.611 | 6.000 |
| Pow3R | 0.062 | 0.966 | 5.000 | 0.061 | 0.968 | 4.000 | 0.061 | 0.968 | 4.000 | 0.064 | 0.965 | 4.318 |
| LDCM | **0.015** | **0.993** | **1.000** | **0.017** | **0.991** | **1.000** | **0.020** | **0.990** | **1.000** | **0.017** | **0.992** | **1.000** |

Table 28: **Quantitative comparison of affine-invariant point map estimation with baseline methods on the indoor scenes of the ETH3D dataset** Schops et al. (2017). The best and the second-best results are highlighted.

| Method | 10% Noise | | | 5% | | | 3% | | | 1% | | |
|---|---|---|---|---|---|---|---|---|---|---|---|---|
| | $REL^p \downarrow$ | $\delta_1^p \uparrow$ | Rk.$\downarrow$ | $REL^p \downarrow$ | $\delta_1^p \uparrow$ | Rk.$\downarrow$ | $REL^p \downarrow$ | $\delta_1^p \uparrow$ | Rk.$\downarrow$ | $REL^p \downarrow$ | $\delta_1^p \uparrow$ | Rk.$\downarrow$ |
| VGGT | 0.045 | 0.988 | 2.000 | 0.045 | 0.988 | 3.500 | 0.045 | 0.988 | 3.500 | 0.045 | 0.988 | 3.500 |
| MoGe V2 | 0.041 | 0.986 | 2.500 | 0.041 | 0.986 | 3.000 | 0.041 | 0.986 | 3.500 | 0.041 | 0.986 | 3.000 |
| WorldMirror | 0.048 | 0.986 | 4.000 | 0.042 | 0.989 | 2.500 | 0.040 | 0.990 | 1.500 | 0.041 | 0.992 | 1.500 |
| MapAnything | 0.255 | 0.559 | 6.000 | 0.252 | 0.562 | 6.000 | 0.252 | 0.562 | 6.000 | 0.254 | 0.560 | 6.000 |
| Pow3R | 0.070 | 0.987 | 4.000 | 0.068 | 0.990 | 3.500 | 0.069 | 0.990 | 3.500 | 0.070 | 0.990 | 4.000 |
| LDCM | **0.047** | **0.994** | **2.000** | **0.047** | **0.994** | **2.500** | **0.047** | **0.994** | **2.500** | **0.047** | **0.994** | **2.500** |
| Method | 500 | | | 100 | | | SIFT | | | ORB | | |
| | $REL^p \downarrow$ | $\delta_1^p \uparrow$ | Rk.$\downarrow$ | $REL^p \downarrow$ | $\delta_1^p \uparrow$ | Rk.$\downarrow$ | $REL^p \downarrow$ | $\delta_1^p \uparrow$ | Rk.$\downarrow$ | $REL^p \downarrow$ | $\delta_1^p \uparrow$ | Rk.$\downarrow$ |
| VGGT | 0.045 | 0.988 | 3.000 | 0.045 | 0.988 | 3.000 | 0.045 | 0.988 | 3.000 | 0.045 | 0.988 | 2.500 |
| MoGe V2 | 0.041 | 0.986 | 3.000 | 0.041 | 0.986 | 3.000 | 0.041 | 0.986 | 3.000 | 0.041 | 0.986 | 2.500 |
| WorldMirror | 0.043 | 0.991 | 2.000 | 0.043 | 0.990 | 2.000 | 0.043 | 0.990 | 2.000 | 0.048 | 0.980 | 4.000 |
| MapAnything | 0.261 | 0.545 | 6.000 | 0.259 | 0.549 | 6.000 | 0.258 | 0.549 | 6.000 | 0.257 | 0.554 | 6.000 |
| Pow3R | 0.073 | 0.988 | 4.000 | 0.073 | 0.988 | 4.000 | 0.073 | 0.988 | 4.000 | 0.073 | 0.989 | 3.500 |
| LDCM | **0.047** | **0.993** | **2.500** | **0.048** | **0.992** | **2.500** | **0.049** | **0.993** | **2.500** | **0.050** | **0.992** | **2.500** |
| Method | Virtual-Lidar-32-Lines | | | Virtual-Lidar-16-Lines | | | Virtual-Lidar-8-Lines | | | Average | | |
| | $REL^p \downarrow$ | $\delta_1^p \uparrow$ | Rk.$\downarrow$ | $REL^p \downarrow$ | $\delta_1^p \uparrow$ | Rk.$\downarrow$ | $REL^p \downarrow$ | $\delta_1^p \uparrow$ | Rk.$\downarrow$ | $REL^p \downarrow$ | $\delta_1^p \uparrow$ | Rk.$\downarrow$ |
| VGGT | 0.045 | 0.988 | 3.500 | 0.045 | 0.988 | 3.000 | 0.045 | 0.988 | 3.500 | 0.045 | 0.988 | 3.091 |
| MoGe V2 | 0.041 | 0.986 | 3.000 | 0.041 | 0.986 | 3.000 | 0.041 | 0.986 | 3.000 | 0.041 | 0.986 | 2.955 |
| WorldMirror | 0.041 | 0.991 | 1.500 | 0.042 | 0.991 | 2.000 | 0.043 | 0.990 | 2.000 | 0.043 | 0.989 | 2.273 |
| MapAnything | 0.259 | 0.551 | 6.000 | 0.257 | 0.554 | 6.000 | 0.256 | 0.552 | 6.000 | 0.256 | 0.554 | 6.000 |
| Pow3R | 0.071 | 0.989 | 4.000 | 0.072 | 0.988 | 4.000 | 0.073 | 0.989 | 4.000 | 0.071 | 0.989 | 3.864 |
| LDCM | **0.047** | **0.994** | **2.500** | **0.047** | **0.993** | **2.500** | **0.048** | **0.992** | **2.500** | **0.048** | **0.993** | **2.455** |

Table 29: **Quantitative comparison of affine-invariant point map estimation with baseline methods on the outdoor scenes of the ETH3D dataset** Schops et al. (2017). The best and the second-best results are highlighted.

| Method | Virtual-Lidar-64-Lines | | | Virtual-Lidar-32-Lines | | | Virtual-Lidar-16-Lines | | | Virtual-Lidar-8-Lines | | |
|---|---|---|---|---|---|---|---|---|---|---|---|---|
| | $REL^p \downarrow$ | $\delta_1^p \uparrow$ | Rk.$\downarrow$ | $REL^p \downarrow$ | $\delta_1^p \uparrow$ | Rk.$\downarrow$ | $REL^p \downarrow$ | $\delta_1^p \uparrow$ | Rk.$\downarrow$ | $REL^p \downarrow$ | $\delta_1^p \uparrow$ | Rk.$\downarrow$ |
| VGGT | 0.061 | 0.967 | 4.500 | 0.061 | 0.967 | 4.500 | 0.061 | 0.967 | 4.500 | 0.061 | 0.967 | 4.500 |
| MoGe V2 | 0.046 | 0.974 | 2.500 | 0.046 | 0.974 | 2.500 | 0.046 | 0.974 | 2.500 | 0.046 | 0.974 | 2.500 |
| WorldMirror | 0.048 | 0.970 | 3.500 | 0.048 | 0.969 | 3.500 | 0.048 | 0.971 | 3.500 | 0.050 | 0.969 | 3.500 |
| MapAnything | 0.273 | 0.542 | 6.000 | 0.277 | 0.535 | 6.000 | 0.276 | 0.540 | 6.000 | 0.275 | 0.543 | 6.000 |
| Pow3R | 0.076 | 0.977 | 3.500 | 0.075 | 0.978 | 3.500 | 0.075 | 0.978 | 3.500 | 0.075 | 0.980 | 3.500 |
| LDCM | **0.027** | **0.997** | **1.000** | **0.028** | **0.997** | **1.000** | **0.029** | **0.996** | **1.000** | **0.032** | **0.993** | **1.000** |
| Method | Virtual-Lidar-4-Lines | | | 10% Noise | | | 5% | | | 3% | | |
| | $REL^p \downarrow$ | $\delta_1^p \uparrow$ | Rk.$\downarrow$ | $REL^p \downarrow$ | $\delta_1^p \uparrow$ | Rk.$\downarrow$ | $REL^p \downarrow$ | $\delta_1^p \uparrow$ | Rk.$\downarrow$ | $REL^p \downarrow$ | $\delta_1^p \uparrow$ | Rk.$\downarrow$ |
| VGGT | 0.061 | 0.967 | 4.500 | 0.061 | 0.967 | 3.500 | 0.061 | 0.967 | 3.500 | 0.061 | 0.967 | 3.500 |
| MoGe V2 | 0.046 | 0.974 | 2.500 | 0.046 | 0.974 | 2.000 | 0.046 | 0.974 | 2.000 | 0.046 | 0.974 | 2.000 |
| WorldMirror | 0.048 | 0.973 | 3.500 | 0.058 | 0.961 | 3.500 | 0.061 | 0.953 | 4.000 | 0.063 | 0.951 | 4.500 |
| MapAnything | 0.276 | 0.546 | 6.000 | 0.271 | 0.545 | 6.000 | 0.268 | 0.549 | 6.000 | 0.268 | 0.553 | 6.000 |
| Pow3R | 0.075 | 0.980 | 3.500 | 0.082 | 0.960 | 5.000 | 0.079 | 0.969 | 4.000 | 0.078 | 0.972 | 4.000 |
| LDCM | **0.040** | **0.985** | **1.000** | **0.026** | **0.998** | **1.000** | **0.027** | **0.998** | **1.000** | **0.027** | **0.998** | **1.000** |
| Method | 1% | | | SIFT | | | ORB | | | Average | | |
| | $REL^p \downarrow$ | $\delta_1^p \uparrow$ | Rk.$\downarrow$ | $REL^p \downarrow$ | $\delta_1^p \uparrow$ | Rk.$\downarrow$ | $REL^p \downarrow$ | $\delta_1^p \uparrow$ | Rk.$\downarrow$ | $REL^p \downarrow$ | $\delta_1^p \uparrow$ | Rk.$\downarrow$ |
| VGGT | 0.061 | 0.967 | 4.000 | 0.061 | 0.967 | 4.500 | 0.061 | 0.967 | 3.500 | 0.061 | 0.967 | 4.091 |
| MoGe V2 | 0.046 | 0.974 | 2.500 | 0.046 | 0.974 | 2.500 | 0.046 | 0.974 | 2.000 | 0.046 | 0.974 | 2.318 |
| WorldMirror | 0.054 | 0.961 | 4.000 | 0.050 | 0.969 | 3.500 | 0.077 | 0.931 | 4.500 | 0.055 | 0.962 | 3.773 |
| MapAnything | 0.271 | 0.548 | 6.000 | 0.276 | 0.543 | 6.000 | 0.275 | 0.545 | 6.000 | 0.273 | 0.544 | 6.000 |
| Pow3R | 0.076 | 0.975 | 3.500 | 0.076 | 0.977 | 3.500 | 0.077 | 0.973 | 3.500 | 0.077 | 0.974 | 3.727 |
| LDCM | **0.027** | **0.997** | **1.000** | **0.031** | **0.994** | **1.000** | **0.030** | **0.994** | **1.000** | **0.029** | **0.995** | **1.000** |

