# OpenReview forum: "Large Depth Completion Model from Sparse Observations"
_ICLR.cc/2026/Conference — ICLR 2026 Poster_

### Official Review · Reviewer_TPZH · 2025-10-29

**Soundness:** 3
**Presentation:** 3
**Contribution:** 3
**Rating:** 6
**Confidence:** 5

**Summary:**

This paper proposes the Large Depth Completion Model (LDCM) to address the problem of estimating dense and metrically accurate depth maps from sparse depth observations. It mainly involves a Poisson-based coarse depth initialization and a point map regression head for 3D structure prediction. The authors performs comprehensive evaluations across multiple benchmark datasets under varying sparsity of inputs, demonstrating state-of-the-art results.

**Strengths:**

1. Robust Results. The extensive experiments on diverse benchmarks (including KITTI, ETH3D, DIODE, etc.) demonstrate that LDCM consistently outperforms existing methods, especially in zero-shot settings and under sparse input conditions. This indicates a high level of generalization and robustness of the proposed model.

2. Extensive Training and Comprehensive Evaluation. The authors have invested considerable effort in training LDCM on a large number of datasets. The extensive evaluation spans a wide range of sparsity levels, providing a comprehensive comparison.

3. Effective Preprocessing Method. The Poisson-based depth initialization strategy that incorporates geometric priors from monocular depth models to construct initial coarse depth is interesting.

**Weaknesses:**

1. Inference Time. While LDCM demonstrates excellent performance across multiple benchmarks, the model’s inference time has not been explicitly discussed. It would be useful to include an analysis of the model's runtime efficiency.

2. More Comparison Methods. Pow3R introduces sparse priors to predict pointmaps, a direct comparison with the pow3r model is necessary.

3. Visualization Results. Although the paper provides extensive quantitative evaluations, additional visualizations of the depth completion and point map estimation results would enhance the reader's understanding of the model’s performance.

[1] CVPR 2025, Pow3R: Empowering Unconstrained 3D Reconstruction with Camera and Scene Priors

**Questions:**

The sparse points shown in the head figure are difficult to visualize clearly. To improve clarity and provide a better qualitative analysis, it would be beneficial to plot the sparse points directly on the RGB images.

---

> ### Author Response · Authors · 2025-11-21
> **Official reply to reviewer TPZH**
>
> **W1: Inference time.**
>
> Thanks for your suggestions. In Table 11, we present an inference time comparison with state-of-the-art approaches for comprehensive evaluation. As shown in the table, LDCM achieves comparable inference speed to existing methods while delivering superior accuracy.
>
> **W2: Comparison with Pow3R.**
>
> Thank you for your suggestions. We compare both metric depth prediction and affine-invariant point map estimation against additional methods that leverage auxiliary priors as input, including WorldMirror, MapAnything, and Pow3R, as shown in Tables 1, 3, and 6. For detailed per-dataset metrics, please refer to Tables 12 through 29. LDCM achieves the best overall performance across these evaluations.
>
> **W3: More visualization results.**
>
> Thank you for your suggestions. We provide additional qualitative comparisons in Figures 7, 8, and 9. Notably, LDCM produces sharper geometric structures and more accurate depth distributions—particularly in regions with complex geometry or extreme sparsity. Its predictions exhibit significantly clearer object boundaries and finer structural details, demonstrating the effectiveness of our approach.
>
> **Q1: The sparse points shown in the head figure are difficult to visualize clearly.**
>
> Thanks for your suggestions. We've updated the head figure to provide clearer presentation.

---

> > ### Comment · Reviewer_TPZH · 2025-11-22
> >
> > I appreciate the feedback provided by authors. I have no further questions and I vote for acceptance.

---

> > > ### Author Response · Authors · 2025-11-24
> > > **Official reply to Reviewer TPZH**
> > >
> > > Thank you for your thoughtful follow-up and for acknowledging the revisions we made. We sincerely appreciate your constructive feedback!

---

### Official Review · Reviewer_KZnL · 2025-10-29

**Soundness:** 3
**Presentation:** 3
**Contribution:** 3
**Rating:** 8
**Confidence:** 4

**Summary:**

The paper introduces LDCM, a transformer-based model to estimate dense metric depth from a single image and a sparse depth. It proposed a Poisson-based preprocessing strategy that leverages a pretrained depth foundation model (Depth Anything V2) to produce an initial coarse depth map and replaced the conventional depth head with a pointmap head. Trained on a large collection of high-quality datasets, the proposed model demonstrates superior performance across various benchmarks under different sparse levels, outperforming previous state-of-the-art methods.

**Strengths:**

1. The proposed model is simple but effective. It outperforms a recent SOTA baseline, such as PriorDA and PromptDA.
2. Compared to previous coarse alignment method, the proposed Poisson-based strategy is effective, providing a strong initialization.
3. The authors perform a detailed set of zero-shot experiments on multiple datasets, demonstrating the effectiveness of LDCM in both depth completion and point map estimation. The model consistently outperforms prior methods in a variety of settings.
4. This paper is well written and the implementation details are comprehensive and easy to follow.

**Weaknesses:**

1. The main concern comes from the limited qualitative comparison. The manuscript mainly provides quantitative results, where the proposed LDCM ranks first on most of the benchmarks. The authors may consider adding more visualized results to better illustrate the performance differences and provide more intuitive insights into the effectiveness of the proposed LDCM.
2. The inference time of each component should be mentioned to provide a comprehensive evaluation for the proposed model.
3. Recently, there are lots of 3D reconstruction models which introduce additional depth priors (e.g., WorldMirror, MapAnything, Pow3R). To may understanding, the sparse prior provides both metric and relative geometry guidance. Although these methods generate relative geometry, the metric scale can be recovered by the proposed Poisson strategy or least square alignment. Comparing performance with these methods would provide a more comprehensive evaluation.

**Questions:**

Please see "Weaknesses" section.

---

> ### Author Response · Authors · 2025-11-21
> **Official reply to reviewer KZnL**
>
> **W1: Adding more visualized results.**
>
> Thank you for your suggestions. We provide additional qualitative comparisons in Figures 7, 8, and 9. Notably, LDCM produces sharper geometric structures and more accurate depth distributions—particularly in regions with complex geometry or extreme sparsity. Its predictions exhibit significantly clearer boundaries and finer details, demonstrating the effectiveness of our method.
>
> **W2: Inference time.**
>
> Thank you for your suggestions. We provide a detailed breakdown of the inference time for each stage of our model, measured at a resolution of 480×640 on an NVIDIA L20 GPU. Our pipeline consists of four stages: Depth Anything Small (0.006 s), global alignment (0.006 s), Poisson-based alignment (0.020 s), and the subsequent refinement module (0.040 s), resulting in a total runtime of 0.072 seconds. In Table 11, we present an inference time comparison with state-of-the-art approaches for comprehensive evaluation.
>
> **W3: Comparison with WorldMirror, MapAnything, Pow3R**
>
> Thank you for your suggestions. We compare both metric depth prediction and affine-invariant point map estimation with these methods, as shown in Tables 1, 3, and 6. For detailed per-dataset metrics, please refer to Tables 12 through 29. LDCM outperforms all these methods across multiple benchmarks, demonstrating its effectiveness.

---

> > ### Comment · Reviewer_KZnL · 2025-11-24
> >
> > Thank the authors for their feedback. My concerns are addressed and I would like to keep my rating for acceptance.

---

> > > ### Author Response · Authors · 2025-11-24
> > > **Official reply to Reviewer KZnL**
> > >
> > > Thank you for your positive feedback and for noting that our revisions have resolved your concerns. We truly appreciate your support.

---

### Official Review · Reviewer_LhSg · 2025-10-30

**Soundness:** 3
**Presentation:** 3
**Contribution:** 3
**Rating:** 6
**Confidence:** 4

**Summary:**

This paper presents the Large Depth Completion Model (LDCM), a novel framework for generating dense, metrically accurate depth maps from sparse and irregular depth inputs. The key innovation lies in a two-stage approach: first, it proposes a preprocessing pipeline that uses a pretrained depth foundation model to estimate a gradient field, which is then integrated via a Poisson solver to generate a coarse but geometrically coherent initial depth map. Second, for depth prediction, LDCM introduces the use of a pointmap representation—a 3D point cloud-like structure—to directly model scene geometry, enabling more precise depth completion.

The method achieves state-of-the-art performance across six major benchmark datasets, excelling in both depth completion and pointmap estimation tasks, with consistent top rankings on all evaluated metrics.

**Strengths:**

1. The paper is well-structured and transparent, offering comprehensive implementation details, including a thorough list of training datasets, which enhances reproducibility. Also, the code in the supplementary material shows the details of the model.
2. The proposed gradient-field-based initialization using a foundation model and Poisson reconstruction effectively preserves geometric structure and metric scale, providing a strong starting point for refinement.
3. This work is the first to introduce the pointmap as a core representation in depth completion, enabling direct 3D scene modeling and improving structural fidelity.
4. Evaluation is comprehensive, covering multiple benchmarks, with detailed quantitative results, ablation studies, and comparisons that convincingly demonstrate the superiority of the proposed method.

**Weaknesses:**

1. While the quantitative performance is impressive, the paper would benefit significantly from more visual comparisons (e.g., side-by-side depth map visualizations) to intuitively illustrate LDCM’s advantages over baselines, especially in challenging regions like object boundaries or low-sampling areas.
2. Recent advances in diffusion-based models have demonstrated strong performance in depth estimation and completion (e.g., [1][2]), despite their typically longer inference times. While LDCM may prioritize efficiency, including a comparison with these state-of-the-art diffusion methods would provide a more comprehensive benchmarking landscape.
3. The training data includes synthetic and high-quality real datasets. However, it is crucial to discuss the impact of introducing more real-world data.
4. The paper compares with monocular depth estimation models, which are inherently scale-ambiguous. Providing these details of scale alignment is crucial for fair and reproducible comparison.

Overall, I’m inclined to accept this paper, but I encourage the authors to provide more comparison and ablation experiments.


[1] Marigold-DC: Zero-Shot Monocular Depth Completion with Guided Diffusion

[2] DepthLab: From Partial to Complete


 Minor weakness

1. There is a minor inconsistency in Table 8 (e.g., "Unidepthv1" vs. "Unidepth V1") that also appears in subsequent tables. This should be corrected for clarity and consistency.

**Questions:**

N/A

---

> ### Author Response · Authors · 2025-11-21
> **Official reply to reviewer LhSg**
>
> **W1: While the quantitative performance is impressive, the paper would benefit significantly from more visual comparisons.**
>
> Thank you for your suggestions. We provide additional qualitative comparisons in Figures 7, 8, and 9. Notably, LDCM produces sharper geometric structures and more accurate depth distributions—particularly in regions with complex geometry or extreme sparsity. Its predictions exhibit significantly clearer boundaries and finer details, demonstrating the effectiveness of our method.
>
> **W2: Comparison with diffusion-based methods.**
>
> Thank you for your suggestions. In Table 8, we present additional comparisons with diffusion-based models: Marigold-DC and DepthLab. Due to their prohibitively long inference times, we evaluate these methods primarily on three benchmarks with varying levels of sparse input: NYUv2 (500 and 100 points), VOID (150, 500, and 1,500 points), and KITTI (64, 32, and 16 scan lines). Notably, LDCM consistently outperforms both methods by a large margin.
>
> **W3: Ablation on the training data.**
>
> We perform an ablation study on the training data used to train the LDCM. In addition to the original datasets, we introduce an extra dataset: DrivingStereo. The quantitative results are presented in Table 9. As shown, the inclusion of this additional data does not significantly affect metric performance. However, as illustrated in Fig. 5, incorporating more real-world data leads to visually less sharp predictions, likely due to imperfect supervision signals of real datasets. In the future, we'll investigate how to effectively leverage the large-amount real datasets.
>
> **W4: Details for monocular scale alignment.**
>
> Thanks for your suggestions. We've updated it. Please refer to Line 989
>
> **W5: Fonts error.**
>
> Thanks, we've fixed it.

---

> > ### Comment · Reviewer_LhSg · 2025-11-24
> > **Official comment by reviewer LhSg**
> >
> > The authors have added further details and included additional comparison methods. My concerns have been satisfactorily addressed, and I maintain my initial positive rating.

---

> > > ### Author Response · Authors · 2025-11-25
> > > **Official reply to Reviewer LhSg**
> > >
> > > Thank you for your insightful feedback and for acknowledging the revisions we made. We truly appreciate your constructive suggestion!

---

### Official Review · Reviewer_69y4 · 2025-10-31

**Soundness:** 3
**Presentation:** 2
**Contribution:** 3
**Rating:** 4
**Confidence:** 3

**Summary:**

The paper proposes a two-stage pipeline that produces a metric-consistent dense depth map from a sparse prior + a monocular relative-depth foundation model via a Poisson-style reconstruction, and then feeds that coarse depth and RGB image into the network that regresses a metric 3D point map. The method is evaluated zero-shot across many benchmarks and reports better performance than previous works in both depth completion and point-map estimation tasks.

**Strengths:**

	The proposed Poisson-based initialization framework is architecturally simple yet highly effective, avoiding complex model design.
	The paper provides extensive empirical evaluation through cross-dataset, zero-shot testing on multiple benchmarks (KITTI, iBims-1, DIODE, ETH3D, etc.), demonstrating strong generalization.

**Weaknesses:**

	Table 3 evaluates different alignment strategies (Global, LWLR, Poisson), but it would be insightful to include a comparison using Equation (1) only, i.e., a baseline without the global affine correction ($\alpha, \beta$) to show the quantitative gain from incorporating the global affine term.

	Since DepthAnything, VGGT, and MoGe also predict relative depth as the authors mentioned, in Tables 1 and 5, it would further strengthen the paper to show their results after applying the same Poisson-based alignment used in this paper. Demonstrating that the current result of this paper still outperforms these baselines after identical alignment would make the contribution more compelling. Also, in table 2 (point map evaluation), it would further strengthen the evaluation if the authors could include comparisons with recent point map models such as VGGT or DUSt3R, even under RGB-only settings.

	As shown in Appendix Tables 8 and 9, the performance degrades more noticeably than some baselines when depth noise is added. The paper lacks an analysis explaining this sensitivity, e.g., whether it arises from their framework design.

	The inference time comparison is not discussed in the methodology, experimental results, or tables. Including the inference time would help verify the effectiveness and practicality of the proposed method.

I will reconsider the score when all the concerns are handled well.

**Questions:**

	Could the authors clarify the computational cost of the Poisson reconstruction stage (Eq. 1–4) compared to simpler alignment methods like LWLR?

	The related work ‘monocular geometry estimation’ section could be improved by adding recent and concurrent 3D foundation models such as
- [1] Jiang, Lihan, et al. "AnySplat: Feed-forward 3D Gaussian Splatting from Unconstrained Views." SIGGRAPH Asia 2025
- [2] Keetha, Nikhil, et al. "MapAnything: Universal feed-forward metric 3D reconstruction." arXiv preprint arXiv:2509.13414 (2025).

Including and discussing these would better situate the proposed approach in the broader landscape of geometry foundation models.
Further reference recommendation for depth-completion task:
- [1] Jeong, Chanhwi, et al. "Test-Time Prompt Tuning for Zero-Shot Depth Completion." ICCV 2025.

---

> ### Author Response · Authors · 2025-11-21
> **Official reply to reviewer 69y4 (1/2)**
>
> **W1: Table 3 evaluates different alignment strategies (Global, LWLR, Poisson), but it would be insightful to include a comparison using Equation (1) only.**
>
> Thanks for your suggestion. We removed the global alignment strategy and directly perform Poisson alignment using the relative depth from DepthAnythingV2, with the results displayed in Table 4 (displayed in page 9 of the manuscript). We reproduce it below for convenience. As shown in the table, global alignment is essential—its omission leads to a significant performance drop.
>
> |Configuration|KITTI|iBims-1|DIODE|ETH3D|Average|
> |:-:|:-:|:-:|:-:|:-:|:-:|
> |Sparse|-|-|-|-|-|
> |Global alignment|0.095|0.075|0.102|0.078|0.087|
> |LWLR| 0.078 |0.108|0.108|0.061|0.088|
> |Poisson w/o global alignment|0.069|0.208|0.174|0.138|0.147|
> |Poisson|0.033|0.073|0.088|0.044|0.059|
>
> **W2.1: Since DepthAnything, VGGT, and MoGe also predict relative depth as the authors mentioned, in Tables 1 and 5, it would further strengthen the paper to show their results after applying the same Poisson-based alignment used in this paper.**
>
> We evaluate on the same depth completion benchmarks and apply the Poisson-based alignment strategy to various relative depth estimators, with results reported in Table 10 (displayed in page 21 of the manuscript). We reproduce it below for convenience. As shown in the table, the Poisson alignment consistently improves performance across all methods, while our LDCM outperforms all competitors by a clear margin.
> |Method|RMSE|MAE|REL|d1|Rk.|
> |:-:|:-:|:-:|:-:|:-:|:-:|
> |DepthAnythingV2|2.555|1.092|0.071|0.943|5.364|
> |DepthAnythingV2 w/ Poisson|1.364|0.498|0.039|0.965|3.261|
> |VGGT|2.086|1.243|0.121|0.876|6.391|
> |VGGT w/ Poisson|1.267|0.546|0.047|0.953|3.975|
> |MoGe V1|3.157|2.201|0.142|0.872|5.427|
> |MoGe V1 w/ Poisson|1.046|0.413|0.035|0.967|2.284|
> |LDCM (Ours)|0.862|0.237|0.017|0.987|1.050|
>
> **W2.2: it would further strengthen the evaluation if the authors could include comparisons with recent point map models such as VGGT or DUSt3R.**
>
> Thank you for your suggestion. Following MoGe, we evaluate the accuracy of affine-invariant point map estimation. In Table 3 (displayed in page 9 of the manuscript), we compare the performance of several methods, including monocular estimators (VGGT, MoGe V2) as well as approaches that leverage additional priors as input (WorldMirror, MapAnything, and Pow3R). We reproduce the averaged results below. Our method achieves superior performance across all settings, demonstrating that it preserves—rather than compromises—the accuracy of relative geometry estimation.
>
> |Method|REL|d1|Rk.|
> |:-:|:-:|:-:|:-:|
> |VGGT|0.114|0.879|4.327|
> |MoGe V2|0.064|0.947|2.191|
> |WorldMirror|0.086|0.914|3.032|
> |MapAnything|0.268|0.553|6.000|
> |Pow3R|0.119|0.899|4.132|
> |LDCM (Ours)|0.037|0.983|1.164|
>
> **W3: As shown in Appendix Tables 8 and 9, the performance degrades more noticeably than some baselines when depth noise is added. The paper lacks an analysis explaining this sensitivity, e.g., whether it arises from their framework design.**
>
> Thank you for your suggestion. Tables 8 and 9 have been updated to Tables 13 and 14 (displayed in page 26 and 27 of the manuscript), respectively. On these benchmarks, our LDCM is only slightly inferior to SPNet and G2-MonoDepth, with a minor performance gap, when depth noise is added. We hypothesize that this is because these methods employ separate model configurations specifically tailored for indoor and outdoor datasets. Instead, our method uses a single unified model that achieves consistently competitive performance across diverse scenes without scene-specific tuning. Moreover, on other datasets and under different settings, our method continues to deliver comparable or even superior performance.
>
> To investigate the robustness of our framework to noise, we provide an example in Figure 10 (displayed in page 24 of the manuscript). This example demonstrates that the Poisson alignment strategy would be affected by noisy points. However, thanks to the augmentation strategy employed during training, our model effectively mitigates such noise and produces high-quality results.

---

> ### Author Response · Authors · 2025-11-26
> **Official reply to reviewer 69y4 (2/2)**
>
> **W4/Q1: The inference time comparison is not discussed in the methodology, experimental results, or tables. Including the inference time would help verify the effectiveness and practicality of the proposed method.**
>
> Thank you for your suggestion. We provide a detailed breakdown of the inference time for each stage of our model, measured at a resolution of 480×640 on an NVIDIA L20 GPU. Our pipeline consists of four stages: Depth Anything Small (0.006 s), global alignment (0.006 s), Poisson-based alignment (0.020 s), and the subsequent refinement module (0.040 s), resulting in a total runtime of 0.072 seconds. For comparison, LWLR runs in 0.010 s under the same conditions.
>
> The Poisson-based alignment stage is slightly slower than LWLR’s lightweight formulation. To address this, we plan to explore acceleration strategies in future work—for example, multiscale Poisson alignment, where the solution at a lower resolution can serve as an initialization for higher-resolution optimization, thereby reducing convergence time and improving overall speed.
>
> Additionally, Table 11 (displayed in page of the manuscript) presents a comprehensive comparison of inference times between our method and other approaches. As shown, our model achieves a favorable trade-off between accuracy and efficiency, demonstrating its practicality for real-world applications.
>
> |Method|OMNI-DC|PriorDA|DepthPro|VGGT|MoGe V2|DepthAnythingV2|LDCM (Ours)|
> |:-:|:-:|:-:|:-:|:-:|:-:|:-:|:-:|
> |Inference time (s)|0.128|0.064|0.554|0.196|0.220|0.019|0.072|
>
> **Q2: More citation to better situate the proposed approach in the broader landscape of geometry foundation models.**
>
> Thank you for your suggestion. In Lines 152–154, we introduce the process of TestPromptDC. For geometry estimation foundation models, we reorganize the structure and provide a detailed introduction of AnySplat and MapAnything in Lines 183–193.

---

> ### Author Response · Authors · 2025-11-27
> **Official reply to reviewer 69y4**
>
> Dear Reviewer,
>
> Thank you sincerely for your thoughtful and constructive feedback.
>
> As the discussion period is drawing to a close, we wanted to kindly check whether our responses have adequately addressed your concerns. If further clarification or additional materials would be helpful, we are happy to provide them promptly.
>
> We truly appreciate your time and insights and welcome any additional questions you may have.
>
> Best regards,
>
> The LDCM authors

---

> ### Comment · Reviewer_69y4 · 2025-11-27
>
> Thank you for the feedback. The authors have addressed my concerns with additional clarifications and ablation studies. Thus, I will raise my score to 6.

---

> > ### Author Response · Authors · 2025-11-27
> > **Official reply to reviewer 69y4**
> >
> > We sincerely appreciate your thoughtful comments and your acknowledgment of the revisions we have made in response to your feedback. Your constructive suggestions have greatly improved our manuscript!

---

### Author Response · Authors · 2025-11-21
**General responce**

We sincerely thank the reviewers for their insightful comments, valuable feedback, and their recognition of the strengths of our work:
+ **Simple and Effective Model**: The reviewers acknowledged the effectiveness and simplicity of our proposed Poisson-based initialization method (Reviews 69y4, LhSg, KZnL, TPZH), noting that it avoids unnecessarily complex model design (69y4).
+ **Clear presentation and Comprehensive Implementation Details**: Reviewers praised our paper as “well-structured” (LhSg), highlighted its “comprehensive implementation details” (LhSg, KZnL), and “easy to follow” (KZnL).
+ **Extensive Evaluation and Strong Expermental Results**: The reviewers commended our extensive evaluation across multiple benchmarks in both depth completion and point map estimation tasks, as well as the strong generalization performance of our approach (69y4, LhSg, KZnL, TPZH).
+ **Introducing of Point Map Representation**: Reviewer LhSg specifically noted the novelty and utility of introducing the point map representation for depth completion.

We first summarize updates of the revised manuscript, followed by responses to individual comments. All revivisions are highlighted in **red** in the updated version.
+ **Global Affine Item  (69y4-W1; Line 431):** Conducted ablation on the global align correction.
+ **Apply Poisson to Relative Depth Estimator (69y4-W2.1; Line 1100):** Applied Poisson alignment strategy to relative depth estimators.
+ **Comparison and Updated Overall Tables (69y4-W2.2;KZnL-W3;TPZH-W2):** Compared peformance with WorldMirror, MapAnything, and Pow3R, etc. Evaluated affine-invariant point map estimation.
+ **Analysis for noise (69y4-W3; Line 1180):** Noise analysis for the input.
+ **Inference time (69y4-W4;69y4-Q1;KZnL-W2;TPZH-W1; Line 1110):** Inference time for each component.
+ **Related work (69y4-Q2; Line 152):** Added references for depth completion.
+ **Related work (69y4-Q2; Line 183):** Added references for geometry estimation.
+ **Visualization (LhSg-W1; KZnL-W1; TPZH-W2;Line 1128):** Added more visualization results.
+ **Comparison with diffusion-based methods (LhSg-W2: Line 1045:** Added comparison results with diffusion-based methods.
+ **Ablation for training data (LhSg-W3: Line 1074):** Added ablation study for training data.
+ **Details for scale alignment of monocular depth estimation methods (LhSg-W4: Line-989):** Added details for scale alignment.

We look forward to further engaging discussions and appreciate the opportunity to refine our work based on the reviewers' constructive feedback.

---

### Author Response · Authors · 2025-11-30
**Rebuttal Summary and Request for Consideration**

Dear Handling Area Chair,

We sincerely thank the anonymous reviewers for their time, thoughtful feedback, and voluntary service, and we greatly appreciate your coordination and support throughout the review process.

This paper was reviewed by four expert reviewers. During the rebuttal phase, all reviewers responded positively to our revisions and clarifications. Notably, Reviewer TPZH raised their rating from 6 to 8, and Reviewer 69y4 increased theirs from 4 to 6. **All updates were completed by 06:13 UTC on 27 November 2025, which predates the public disclosure of the recent information leak (widely reported between 14:00 and 16:00 UTC on 27 November 2025).**

For your reference, we summarize our rebuttal efforts, including our responses to the reviewers’ concerns, the additional clarifications and revisions we provided, and the subsequent exchanges during the discussion phase. In the revision, our key revisions include:

+ **More comparison benchmarks.** We added more baseline methods, including diffusion-based models (Marigold-DC and DepthLab) and geometry foundation models with additional priors (Pow3R, MapAnything, WorldMirror).
+ **Relative geometry evaluation.** As suggested by Reviewer 69y4, we compared the relative geometry performance of our proposed LDCM against dedicated relative geometry estimators; results are shown in Tables 3 and 24–29.
+ **Poisson alignment ablation.** We evaluated the effectiveness of our proposed Poisson alignment strategy on monocular depth estimators, demonstrating significant improvements in metric accuracy.
+ **Ablation study on training data.** As suggested by Reviewer LhSg, we analyzed the impact of incorporating additional real-world data; results are shown in Table 9 and Figure 5.
+ **Albation study on the global affine item.** As suggested by Reviewer 69y4, we evaluated the necessity of the global alignment correction.
+ **Added more qualitative results.** We included a noise robustness example and further visualization comparisons in Figures 5 and 7–9.
+ **Added inference time analysis.** We added a detailed analysis of inference time (see Line 1111) and a comparative summary in Table 11.
+ **Enhanced methodological details.** We expanded the related work section and provided additional details regarding the evaluation protocol for monocular depth estimators.

Thank you very much for your time and consideration. We would be grateful if you could take into account the substantial progress made during the rebuttal, particularly our comprehensive responses to all reviewers and their subsequent positive assessments, when evaluating our submission.

---

### Meta-Review · Area_Chair_StTD · 2026-01-05

**Summary:**

The authors propose a method for dense depth completion that utilizes a monocular (relative) depth foundational model (DepthAnythingV2) that is fitted by a Poisson-based solver to produce an initial depth map. The output is fed to a subsequent network that regresses a metric 3D point map. The method is evaluated zero-shot across several benchmarks, including KITTI, ETH3D, DIODE.

The initial recommendations for this manuscript were 1 marginally below the acceptance threshold (69y4), 2 marginally above the acceptance threshold(LhSg, TPZH), 1 accept (KZnL). All reviewers noted that the proposed Poisson-based preprocessing is effective in obtaining initial geometric structure in metric scale. All reviewers also noted performance of the method. Nonetheless, the reviewers raised several points including: (1) the lack of experiments on methods with and without global alignment strategies, e.g., affine correction, as well as the proposed Poisson alignment strategy (69y4, LhSg, KZnL); (2) experiments and analysis on the sensitivity of the method to noise (69y4); (3) comparisons with additional recent methods including point map models such as VGGT or DUSt3 and diffusion based models (69y4, LhSg, TPZH); (4) computational costs, comparisons, and analysis on inference time, particularly the Poisson reconstruction stage of the method (69y4, KZnL, TPZH); (5) lack of visual comparisons cross methods, particularly in challenging areas such as object boundaries or low-sampling areas. (LhSg, KZnL, TPZH); (6) experiments on more real-world data (LhSg). Overall, the authors were able to address the concerns raised by the reviewers and recommends the authors to include the materials discussed in the rebuttal phase into the next revision of the manuscript.

The AC would like to note that the **reviews focused largely on experiments and visualizations, and did not provide technical feedback on the methodology**.

**Reviewer Concerns:**

The authors posted a rebuttal to address the following points:

(1) the lack of experiments on methods with and without global alignment strategies, e.g., affine correction, as well as the proposed Poisson alignment strategy (69y4, LhSg, KZnL):

The authors added experiments on global alignment correct in Table 3 of revised main paper.

(2) experiments and analysis on the sensitivity of the method to noise (69y4):

The authors added a visualization of noisy sparse depth input on a single example in Figure 10 of the Appendix, and included comparison in Table 13 on 3, 5, and 10% noise.

(3) comparisons with additional recent methods including point map models such as VGGT or DUSt3 and diffusion based models (69y4, LhSg, TPZH):

The authors added comparisons with WorldMirror, MayAnything, Pow3R in Table 1, 3 and with MarigoldDC and DepthLab in Table 8.

(4) computational costs, comparisons, and analysis on inference time, particularly the Poisson reconstruction stage of the method (69y4, KZnL, TPZH):

The authors included inference time of each component and across methods in Table 11 in Appendix.

(5) lack of visual comparisons cross methods, particularly in challenging areas such as object boundaries or low-sampling areas. (LhSg, KZnL, TPZH):

The authors provided additional examples in Figure 7 in the Appendix, but the examples do not highlight the requested areas, all examples selected are largely of planar scenes. The authors were unable to address this point. **The AC recommends that the authors follow the reviewer suggestions and choose examples that properly highlight challenging areas.**

(6) experiments on more real-world data (LhSg): The authors added additional results in Table 9 of the Appendix.

**Reviewer Scores:**

The AC has read the reviews and the rebuttal. Given the responses, the AC expects TPZH, LhSg, and KZnL to maintain their positive rating. 69y4 mentioned in the discussion that they will raise their score to a 6.

---

### Decision · Program_Chairs · 2026-01-26

Accept (Poster)